# FedEBA+: Towards Fair and Effective Federated Learning via Entropy-based Model

## Abstract

Ensuring fairness is a crucial aspect of Federated Learning (FL), which enables the model to perform consistently across all clients. However, designing an FL algorithm that simultaneously improves global model performance and promotes fairness remains a formidable challenge, as achieving the latter often necessitates a trade-off with the former. To address this challenge, we propose a new FL algorithm, FedEBA+, which enhances fairness while simultaneously improving global model performance. FedEBA+ incorporates a fair aggregation scheme that assigns higher weights to underperforming clients and an alignment update method. In addition, we provide theoretical convergence analysis and show the fairness of FedEBA+. Extensive experiments demonstrate that FedEBA+ outperforms other SOTA fairness FL methods in terms of both fairness and global model performance.

## 1 Introduction

Federated Learning (FL) is a distributed learning paradigm that allows clients to collaborate with a central server to train a model (McMahan et al., 2017). To learn models without transferring data, clients process data locally and only periodically transmit model updates to the server, aggregating these updates into a global model. A major challenge for FL is to treat each client fairly to achieve *performance fairness* (Shi et al., 2021), where the global model's performance is uniformly distributed among all clients. Achieving fairness is vital to prevent problems like performance discrimination, client disengagement, and legal and ethical concerns (Caton & Haas, 2020). However, due to data heterogeneity, intermittent client participation, and system heterogeneity, the model is prone to be unfair (Mohri et al., 2019; Papadaki et al., 2022), which decreases FL's generalization ability and hurts clients' willingness to participate (Nishio & Yonetani, 2019).

While there are some encouraging results of addressing fairness challenges in FL, such as objective-based approaches (Mohri et al., 2019; Li et al., 2019a; 2021) and gradient-based methods (Wang et al., 2021; Hu et al., 2022), these methods typically result in a compromise of the performance of global model. However, training an effective global model is the fundamental goal of FL (Kairouz et al., 2019). This raises the question:

*Can we design an algorithm for FL that promotes fairness while improving the performance of the global model?*

To this end, we propose the **FedEBA+** algorithm, which uses **E**ntropy-**B**ased **A**ggregation **plus** alignment update to improve **Fed**erated learning. FedEBA+ introduces a new objective function for FL via a maximum entropy model and comprises a fair aggregation method and an alignment update strategy. Specifically, FedEBA+ algorithm:

1. Implement an entropy-based aggregation strategy called *EBA*, which gives underperforming clients with relatively high aggregation probability, thus making the performance of all clients more consistent.

2. Introduce a novel update process, comprising model alignment and gradient alignment, to enhance the overall accuracy and fairness of the global model.

Maximum entropy models have been successfully adopted to promote fairness in data preprocessing (Singh & Vishnoi, 2014) and resource allocation (Johansson & Sternad, 2005). The

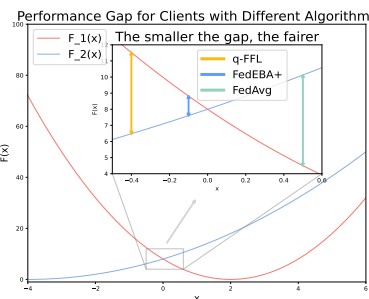

Figure 1: **Illustration of fairness improvement of FedEBA+ over q-FFL and FedAvg.** A smaller performance gap implies a smaller variance, resulting in a fairer method. For clients $F_1(x) = 2(x-2)^2$ and $F_2(x) = \frac{1}{2}(x+4)^2$ with global model $x^t = 0$ at round $t$, q-FFL, FedEBA+, and FedAvg produce $x^{t+1}$ of $-0.4$, $-0.1$, and $0.5$, respectively. The yellow, blue, and green double-arrow lines indicate the performance gap between the clients using different methods. FedEBA+ is the fairest method with the smallest loss gap, thus the smallest performance variance. Computational details are outlined in Appendix 12.1.

principle of constrained maximum entropy ensures a fair selection of probability distribution by being maximally noncommittal to unobserved information, eliminating inherent bias, thereby promoting fairness (Hubbard et al., 1990; Sampat & Zavala, 2019).

However, introducing entropy for fairness in FL is challenging due to its departure from traditional fairness objectives like resource allocation. Fair FL aims to ensure equitable model performance across diverse clients, addressing data heterogeneity and performance bias mitigation with specialized techniques such as model sharing (Donahue & Kleinberg, 2021) and fairness metrics like accuracy variance (Shi et al., 2021). In contrast, traditional fairness in resource allocation focuses on efficiently distributing homogeneous resources based on predefined criteria (Lan et al., 2010). The challenge lies in modeling FL aggregation as entropy and incorporating that using the aggregation method to make the performance distribution more uniform.

To address this challenge, FedEBA+ formulates the FL aggregation as a problem of maximizing entropy with the proposed FL constraints to achieve fair aggregation (Section 4.1). To the best of our knowledge, we are the first to analyze FL aggregation for fairness using entropy. Compared to existing methods such as uniform aggregation in FedAvg (McMahan et al., 2017) and reweight aggregation in q-FFL (Li et al., 2019a), FedEBA+ provides a more uniform performance among clients, as shown in Figure 1. Furthermore, we propose a new objective function that incorporates maximum entropy as a Lagrange constraint in the standard FL objective, resulting in a bi-variable objective (Section 4.2). This enables optimization of the objective over both aggregation probability for a fair aggregation strategy and model parameters for a novel FL model alignment update process, improving both global model accuracy (Section 4.2.1) and enhancing fairness (Section 4.2.2).

**Our major contributions can be summarized as below:**

- We propose FedEBA+, a novel FL algorithm for advocating fairness while improving the global model performance, compromising a fair aggregation method EBA and an innovative model update strategy.

- We provide the convergence analysis for FedEBA+, which is provable to converge to the nearby of the optimal solution in rate $\mathcal{O}\left(\frac{1}{\sqrt{nKT}} + \frac{1}{T}\right)$. In addition, we show the fairness of FedEBA+ via variance using the generalized linear regression model.

- Empirical results on Fashion-MNIST, CIFAR-10, CIFAR-100, and Tiny-ImageNet demonstrate that FedEBA+ outperforms existing fairness FL algorithms, excelling in both fairness and global model performance. In addition, we conduct extensive ablation experiments to assess the impact of hyperparameters on FedEBA+.

## 2 Related Work

In the realm of fairness-aware Federated Learning, various approaches have been explored, addressing concepts such as performance fairness (Li et al., 2019a), group fairness (Du et al., 2021), selection fairness (Zhou et al., 2021), and contribution fairness (Cong et al., 2020), among others (Shi et al., 2021); however, this paper uniquely focuses on performance fairness, aiming to serve client interests and enhance model performance. A more comprehensive discussion of the related work can be found in Appendix 8.

## 3 PRELIMINARIES AND METRICS

**Notations:** Let $m$ be the number of clients and $|S_t| = n$ be the number of selected clients for round $t$. We denote $K$ as the number of local steps and $T$ as the total number of communication rounds. We use $F_i(x)$ and $f(x)$ to represent the local and global loss of client $i$ with model $x$, respectively. Specifically, $x_{t,k}^i$ and $g_{t,k}^i = \nabla F_i(x_{t,k}^i, \xi_{t,k}^i)$ represents the parameter and local gradient of the $k$-th local step in the $i$-th worker after the $t$-th communication, respectively. The global model update is denoted as $\Delta_t = 1/\eta(x_{t+1} - x_t)$, while the local model update is represented as $\Delta_t^i = x_{t,k}^i - x_{t,0}^i$. Here, $\eta$ and $\eta_L$ correspond to the global and local learning rates, respectively.

**Problem formulation.** In this work, we consider the following optimization model:

$$\min_x f(x) = \sum_{i=1}^m p_i F_i(x), \tag{1}$$

where $x \in \mathbb{R}^d$ represents the parameters of a statistical model we aim to find, $m$ is the total number of clients, and $p_i$ denotes the aggregation weight of $i$-th client such that $p_i \geq 0$ and $\sum_{i=1}^m p_i = 1$. Suppose the $i$-th client holds the training data distribution $D_i$, then the local loss function is calculated by $F_i(x) \triangleq \mathbb{E}_{\xi_i \sim D_i}[F_i(x, \xi_i)]$.

**Metrics.** This article aims to *1) promote fairness* in federated learning while *2) enhance the global model's performance*. Regarding the fairness metric, we adhere to the definition proposed by (Li et al., 2019a), which employs the variance of clients' performance as the fairness metric:

**Definition 3.1** (Fairness via variance). *A model $x_1$ is more fair than $x_2$ if the test performance distribution of $x_1$ across the network with $m$ clients is more uniform than that of $x_2$, i.e.* $\mathrm{var}\{F_i(x_1)\}_{i \in [m]} < \mathrm{var}\{F_i(x_2)\}_{i \in [m]}$, *where $F_i(\cdot)$ denotes the test loss of client $i \in [m]$ and* $\mathrm{var}\{F_i(x)\} = \frac{1}{m} \sum_{i=1}^m \left[F_i(x) - \frac{1}{m} \sum_{i=1}^m F_i(x)\right]^2$ *denotes the variance.*

Ensuring the global model's performance is the fundamental goal of FL. However, fairness-targeted algorithms may compromise high-performing clients to mitigate variance (Shi et al., 2021). In addition to global accuracy, we evaluate fairness algorithms by analyzing the accuracy of the best 5% and worst 5% clients to assess potential compromises.

## 4 FEDEBA+: AGGREGATION AND ALIGNMENT UPDATE METHOD

In this section, we introduce a fair aggregation strategy using maximum entropy to ensure FL's fairness constraints and derive a unique aggregation expression (Sec 4.1). We define a novel FL optimization objective in Eq.(5) by treating maximum entropy as the Lagrangian constraint. This dual-objective framework enhances the global model's performance through model alignment when using $\tilde{f}(x)$ as the ideal global objective (Sec 4.2.1) and promotes fairness through gradient alignment when $\tilde{f}(x)$ is considered as the fair objective (Sec 4.2.2). The complete algorithm, encompassing entropy-based aggregation, model alignment, and gradient alignment is presented in Algorithm 1.

### 4.1 FAIR AGGREGATION: EBA

Inspired by the Shannon entropy approach to fairness (Jaynes, 1957), an optimization problem with unique constraints on the fair aggregation for FL is formulated as:

$$\max_p \mathbb{H}(p) := -\sum_{i=1}^m p_i \log(p_i) \qquad s.t. \sum_{i=1}^m p_i = 1, p_i \geq 0, \sum_{i=1}^m p_i F_i(x) = \tilde{f}(x), \tag{2}$$

where $\mathbb{H}(p)$ denotes the aggregation probability's entropy, employed to ensure fair aggregation, and $\tilde{f}(x)$ represents the ideal loss. The ideal loss $\tilde{f}(x)$ serves as the target or objective for the aggregated losses. Its specific expression depends on the goal of the FL system. It acts as a robust constraint for ensuring that the aggregated clients' losses are as close to the desired target loss as possible. For example, when the ideal loss acts as ideal fair loss, the gradient of ideal loss should be a reweight aggregation of clients' unbiased local update as

---

**Algorithm 1** FedEBA+

---

**Require:** initial weights $x_0$, global learning rate $\eta$, local learning rate $\eta_l$, number of rounds $T$.
**Ensure:** trained weights $x_T$
1: **for** round $t = 1, \ldots, T$ **do**
2:    Server selects a set of clients $|S_t|$ and send them model $x_t$.
3:    **if** # Do fairness alignment: **then**               ▷ Consider communication ability
4:      Server collects selected clients' loss $\mathbf{L} = [F_1(x_t), \ldots, F_{|S_t|}(x_t)]$.     ▷ Upload scaler with negligible communication overheads
5:      **if** $arccos(\frac{\mathbf{L},\mathbf{r}}{\|\mathbf{L}\| \cdot \|\mathbf{r}\|}) > \theta$ **then** ▷ Assess the need for fairness alignment by computing the arccosine between the loss vector $\mathbf{L} = [F_1(x_t), \ldots, F_{|S_t|(x_t)}]$ and $\mathbf{r} = [1, \cdots, 1]$
6:        Sever collects $\nabla F_i(x_t)$, calculates gradient $\tilde{g}^{b,t}$ by (9) and sends $\tilde{g}^{b,t}$ to selected clients
7:        **for** each worker $i \in S_t$, in parallel **do**
8:          **for** $k = 0, \cdots, K-1$ **do**
9:            $h_{t,k}^i \leftarrow (1-\alpha)g_{t,k}^i + \alpha\tilde{g}^t$
10:          $\Delta_t^i = x_{t,K}^i - x_{t,0}^i = -\eta_L \sum_{k=0}^{K-1} h_{t,k}^i$
11:        Aggregation: $\Delta_t = \sum_{i \in S_t} p_i \Delta_t^i$, where $p_i$ follows (3) by substituting $x = x_{t,K}^i$
12:    **else**
13:      **for** each worker $i \in S_t$, in parallel **do**
14:        **for** $k = 0, \cdots, K-1$ **do**
15:          $x_{t,k+1}^i = x_{t,k}^i - \eta_L g_{t,k}^i$.
16:        Let $\Delta_t^i = x_{t,K}^i - x_{t,0}^i = -\eta_L \sum_{k=0}^{K-1} g_{t,k}^i$ and $\tilde{\Delta}_t^i = x_{t,1}^i - x_{t,0}^i$
17:      Server aggregates model update by (7)
18:    Server update: $x_{t+1} = x_t + \eta\Delta_t$

---

shown in Section 4.2.2. Introducing $\tilde{f}(x)$ as a constraint reduces bias in model updating and aggregation, benefiting global accuracy and fairness. The toy case in Appendix 12.1, comparing the fairness behavior between FedAvg, q-FedAvg, and FedEBA+, illustrates that higher entropy with constraints equates to greater fairness. Notably, the constrained entropy model (2) can be applied to other tasks by replacing the ideal loss constraint with task-specific constraints. Furthermore, by integrating the maximum entropy model (2) into the original FL objective function (1) as a Lagrangian constraint, we develop a new FL objective function (5), detailed in Section 4.2.

**Proposition 4.1** (EBA: entropy-based aggregation method). *Solving problem* (2)*, we propose an aggregation strategy for prompting fairness performance in FL:*

$$p_i = \frac{\exp[F_i(x)/\tau]}{\sum_{i=1}^N \exp[F_i(x)/\tau]}, \tag{3}$$

*where $\tau > 0$ is the temperature.*

Proof for Proposition 4.1 is in Appendix 9.1, and the uniqueness of the solution (3) to optimization problem (2) is in Appendix 11.

**Remark 4.2** (The effectiveness of $\tau$ on fairness). *$\tau$ controls the fairness level as it decides the spreading of weights assigned to each client. A higher $\tau$ results in uniform weights for aggregation, while a lower $\tau$ yields concentrated weights. This aggregation algorithm degenerates to FedAvg(McMahan et al., 2017) or AFL (Mohri et al., 2019) when $\tau$ is extremely large or small.*

**Remark 4.3** (The annealing manner for $\tau$). *The linear annealing schedule is defined below:*

$$\tau^T = \tau^0/(1 + \kappa(T-1)), \tag{4}$$

*where $T$ is the total communication rounds and hyperparameter $\kappa$ controls the decay rate. There are also concave schedule $\tau^k = \tau^0/(1 + \beta(k-1))^{\frac{1}{2}}$ and convex schedule $\tau^k = \tau^0/(1 + \kappa(k-1))^3$. We show the performance of annealing manners for $\tau$ in Figure 8 of Section 6.*

Proposition 4.1 shows that assigning higher aggregation weights to underperforming clients directs the aggregated global model's focus toward these users, enhancing their performance and reducing the gap with top performers, ultimately promoting fairness.

When taking into account the prior distribution of aggregation probability, which is typically expressed as the relative data ratio $q_i = n_i/\sum_{i \in S_t} n_i$, the expression of EBA becomes $p_i = q_i \exp[F_i(x)/\tau]/\sum_{j=1}^{N} q_j \exp[F_j(x)/\tau]$. The derivation is given in Appendix 9.1.

## 4.2 ALIGNMENT UPDATE

Proposition 4.1 presents a fair aggregation strategy utilizing the maximum entropy model in FL. Building upon this, we enhance FL optimization by incorporating (2) as a Lagrangian constraint in the original FL objective (1), resulting in a bi-variable optimization objective. This objective improves fairness by exploiting the model update process in addition to satisfying aggregation fairness. The new objective function is defined as follows:

$$
L\left(x, p_i; \lambda_0, \lambda_1\right) := \sum_{i=1}^{N} p_i F_i(x) + \beta \left[ \sum_{i=1}^{N} p_i \log p_i + \lambda_0 \left( \sum_{i=1}^{N} p_i - 1 \right) + \lambda_1 \left( \tilde{f}(x) - \sum_{i=1}^{N} p_i F_i(x) \right) \right],
\tag{5}
$$

where $\beta > 0$ is the penalty coefficient of the entropy constraint for the new objective.

The new optimization function (5) is bi-variate, and its optimal solution w.r.t $p_i$ remains the same (equation (3)). By optimizing the function (5) with respect to the variable $x$, we introduce the following model update formula:

$$
\frac{\partial L\left(x, p_i, \lambda_0, \lambda_1\right)}{\partial x} = (1 - \alpha) \sum_{i=1}^{m} p_i \nabla F_i(x) + \alpha \nabla \tilde{f}(x),
\tag{6}
$$

where $\alpha = \lambda_1 \beta \geq 0$ is a constant. The above new update formulation combines the traditional FL update with the *ideal gradient* $\nabla \tilde{f}(x)$ to align model updates. $\nabla \tilde{f}(x)$ represents the *ideal global gradient* $\nabla \tilde{f}^a(x_t)$ for improved global model performance (Section 4.2.1) and represents the *ideal fair gradient* $\nabla \tilde{f}^b(x_t)$ for fairness (Section 4.2.2). The global model is updated by $\Delta_t = -\frac{\partial L(x, p_i, \lambda_0, \lambda_1)}{\partial x} = -(1 - \alpha) \sum_{i=1}^{m} p_i \nabla F_i(x) - \alpha \nabla \tilde{f}(x)$.

### 4.2.1 MODEL ALIGNMENT FOR IMPROVING GLOBAL ACCURACY

Based on equation (6), we propose a new approach to update the server-side model to improve the global model performance. The ideal global gradient $\nabla \tilde{f}^a(x_t)$ aligns the aggregated model with better representing the model update under global data. Although directly obtaining it is unfeasible, we estimate it by averaging local one-step gradients. Utilizing local SGD with $x_{t+1} = x_t - \eta \frac{\partial L(x)}{\partial x}$ and $x_{t+1} = x_t - \eta \Delta_t$, we have

$$
\Delta_t = (1 - \alpha) \sum_{i \in S_t} p_i \Delta_t^i + \alpha \tilde{\Delta}_t^a,
\tag{7}
$$

where $p_i$ follows the aggregation probability (3) by substituting $x = x_{t,K}^i$. Here,$\tilde{\Delta}_t^a$ denotes the aggregation of one-step local updates, defined as follows:

$$
\tilde{\Delta}_t^a = \frac{1}{|S_t|} \sum_{i \in S_t} \tilde{\Delta}_t^{a,i} = \frac{1}{|S_t|} \sum_{i \in S_t} (x_{t,1}^i - x_{t,0}^i).
\tag{8}
$$

We can estimate the ideal global model update using (8) because: 1) A single local update corresponds to an unshifted update on local data, whereas multiple local updates introduce model bias in heterogeneous FL (Karimireddy et al., 2020b). 2) The expectation of sampled clients' data over rounds represents the global data due to unbiased random sampling (Wang et al., 2022). The model alignment update is presented in Algorithm 1 (Steps 11-15). Utilizing $\tilde{\Delta}_t^a$ as the ideal global gradient, the global model update derived from (6) is expressed as (7), as depicted in Algorithm 1.

### 4.2.2 GRADIENT ALIGNMENT FOR FAIRNESS

To enhance fairness, we define $\tilde{f}(x_t)$ as the ideal fair gradient to align the local model updates. We use an arccos-based fairness assessment to determine if gradient alignment should be performed to reduce the communication burden. If the arccos value of clients' performance

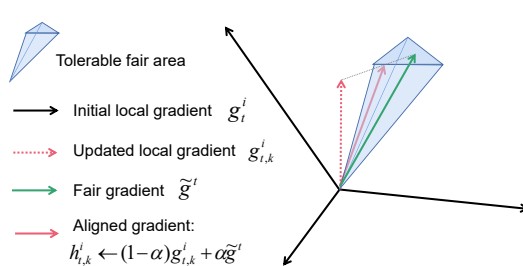

Figure 2: **Gradient Alignment improves fairness.** Gradient alignment ensures that each local step's gradient stays on track and does not deviate too far from the fair direction. It achieves this by constraining the aligned gradient, denoted by $h_{k,t}^i$, to fall within the tolerable fair area. The gradient $g_t^i$ represents the gradient of model for each client in round $t$, while $\tilde{g}^t$ denotes the ideal fair gradient for model $x_t$. The gradient $g_{k,t}^i$ is the gradient of client $i$ at round $t$ and local epoch $k$.

$\mathbf{L} = [F_1(x_t), \ldots, F_{|S_t|}(x_t)]$ and guidance vector $\mathbf{r} = [\mathbf{1}]^{\mathbf{m}}$ exceeds the threshold *fair angle* $\theta$, it is considered unfair; otherwise it is within the tolerable fair area, as shown in Figure 2.

To align gradients, the server receives $g_i^t = \nabla F_i(x_t)$ and $F_i(x_t)$ from clients, employing (3) to assess client importance. Subsequently, the ideal fair gradient $\nabla \tilde{f}^b(x_t)$ is estimated by

$$\nabla \tilde{f}^b(x_t) = \tilde{g}^{b,t} = \sum\nolimits_{i \in S_t} \tilde{p}_i g_i^t \,, \tag{9}$$

where $\tilde{p}_i = \exp[F_i(x_t)/\tau] / \sum_{i \in S_t} \exp[F_i(x_t)/\tau]$. $\tilde{g}^{b,t}$ represents the fair gradient of the selected clients, obtained using the global model's performance on these clients without local shift (i.e., one local update). Therefore, the aligned gradient of model $x_{t,k}^i$ can be expressed as:

$$h_{t,k}^i \leftarrow (1-\alpha)g_{t,k}^i + \alpha\tilde{g}^{b,t} \,. \tag{10}$$

Utilizing $h_{t,k}^i$ for the local update, the global model update, derived from (6), transforms into $\Delta_t = -(1-\alpha)\sum_{i \in S_t} p_i \eta_L \sum_{k=0}^{K-1} g_{t,k}^i - \alpha K \eta_L \sum_{i \in S_t} \tilde{p}_i g_i^t$, as depicted in Algorithm 1.

## 5 ANALYSIS OF CONVERGENCE AND FAIRNESS

In this section, we analyze the convergence and fairness of FedEBA+.

### 5.1 CONVERGENCE ANALYSIS OF FEDEBA+

In this subsection, we provide the convergence rate of FedEBA+ with two-side learning rates. To ease the theoretical analysis of our work, we use the following widely used assumptions:

**Assumption 1** (L-Smooth). *There exists a constant $L > 0$, such that $\|\nabla F_i(x) - \nabla F_i(y)\| \le L\|x - y\|, \forall x, y \in \mathbb{R}^d$, and $i = 1, 2, \ldots, m$.*

**Assumption 2** (Unbiased Local Gradient Estimator and Local Variance). *Let $\xi_t^i$ be a random local data sample in the round $t$ at client $i$: $\mathbb{E}\left[\nabla F_i(x_t, \xi_t^i)\right] = \nabla F_i(x_t), \forall i \in [m]$. There exists a constant bound $\sigma_L > 0$, satisfying $\mathbb{E}\left[\left\|\nabla F_i(x_t, \xi_t^i) - \nabla F_i(x_t)\right\|^2\right] \le \sigma_L^2$.*

**Assumption 3** (Bound Gradient Dissimilarity). *For any set of weights $\{w_i \ge 0\}_{i=1}^m$ with $\sum_{i=1}^m w_i = 1$, there exist constants $\sigma_G^2 \ge 0$ and $A \ge 0$ such that $\sum_{i=1}^m w_i \|\nabla F_i(x)\|^2 \le (A^2 + 1)\|\sum_{i=1}^m w_i \nabla F_i(x)\|^2 + \sigma_G^2$.*

The above assumptions are commonly used in both non-convex optimization and FL literature, see e.g. (Karimireddy et al., 2020b; Yang et al., 2021; Wang et al., 2020). For Assumption 3, if all local loss functions are identical, then we have $A = 0$ and $\sigma_G = 0$.

**Theorem 5.1.** *Under Assumption 1–3 and the aggregation strategy (3), and let constant local and global learning rate $\eta_L$ and $\eta$ be chosen such that $\eta_L < \min(1/(8LK), C)$, where $C$ is obtained from the condition that $\frac{1}{2} - 10L^2 \frac{1}{m}\sum_{i-1}^m K^2 \eta_L^2(A^2 + 1)(\chi_{p\|w}^2 A^2 + 1) > C > 0$, and $\eta \le 1/(\eta_L L)$. In particular, let $\eta_L = \mathcal{O}\left(\frac{1}{\sqrt{T}KL}\right)$ and $\eta = \mathcal{O}\left(\sqrt{Km}\right)$, the convergence rate of FedEBA+ is:*

$$\min_{t \in [T]} \|\nabla f(\boldsymbol{x}_t)\|^2 \le \mathcal{O}\left(\frac{2(f^0 - f^*) + m\sigma_L^2}{2\sqrt{mKT}} + \frac{5(\sigma_L^2 + 4K\sigma_G^2)}{2KT} + \frac{20(A^2 + 1)\chi_{\boldsymbol{w}\|\boldsymbol{p}}^2 \sigma_G^2}{T}\right) + 2\chi_{\boldsymbol{w}\|\boldsymbol{p}}^2 \sigma_G^2 \,. \tag{11}$$

where $\chi^2_{\boldsymbol{w}\|\boldsymbol{p}} = \sum_{i=1}^m (w_i - p_i)^2 / p_i$ represents the chi-square divergence between vectors $\boldsymbol{w} = \left[\frac{1}{m}, \ldots, \frac{1}{m}\right]$ and $\boldsymbol{p} = [p_1, \ldots, p_m]$. Observe that when all clients have uniform loss distribution, we have $\boldsymbol{p} = \boldsymbol{w}$ such that $\chi^2 = 0$.

From Theorem 5.1, we note FedEBA+ will converge to a nearby of the optimality in a rate of $\mathcal{O}\left(\frac{1}{T} + \frac{1}{nKT}\right)$, the same order as that of the SOTA FedAvg (Yang et al., 2021; Li et al., 2022). The proof of Theorem 5.1 in two cases of $\alpha = 0$ and $\alpha \neq 0$ is given in Appendix 10.

**Remark 5.2** (Effect of $\chi^2_{\boldsymbol{w}\|\boldsymbol{p}}\sigma_G^2$). *The non-vanishing term $\chi^2_{\boldsymbol{w}\|\boldsymbol{p}}\sigma_G^2$ in equation (11) means the aggregation error from unbiased aggregation distribution. That is, an error term always exists in the convergence rate as long as the aggregation algorithm is biased. This conclusion is consistent with previous works of FL (Wang et al., 2020; Cho et al., 2022).*

## 5.2 FAIRNESS ANALYSIS

In this section, we analyze fairness via variance and Pareto-optimality for FedEBA+.

**Variance analysis.** In this section, we analyze the variance of clients of FedEBA+ using the generalized linear regression model (Wainwright et al., 2008), following the setting in Li et al. (2020a), which is formulated by $f(\mathbf{x}; \xi) = T(\xi)^\top \mathbf{x} - A(\xi)$, where $T(\xi)$ is the generalized regression coefficient, and $A(\xi)$ is the noise term follows gaussian distribution. Details are shown in Appendix 12.2. Notably, this regression model is a more general version of $y_{k,i} = \tilde{\xi}_{k,i}^\top \mathbf{x}_k + z_{k,i}$, which was employed in (Lin et al., 2022) for fairness analysis.

**Theorem 5.3** (Fairness via variance). *According to Algorithm 1, the variance of test losses of FedEBA+ can be derived as:*

$$V^{EBA+} = \mathrm{var}\left(F_i^{test}\left(\mathbf{x}_{EBA+}\right)\right) = \frac{\tilde{b}^2}{4} \mathrm{var}\left(\|\tilde{\mathbf{w}} - \mathbf{w}_i\|_2^2\right), \tag{12}$$

*where $\tilde{\mathbf{w}} = \sum_{i=1}^m p_i \mathbf{w}_i$, $\mathbf{w}_i$ represents the true parameter on client $i$, , and $\tilde{b}$ is a constant that approximates $b_i$ in $\boldsymbol{\Xi}_i^\top \boldsymbol{\Xi}_i = mb_i\mathbf{I}_d$, where $\Xi_i = [T(\xi_{i,1}), \ldots, T(\xi_{i,n})]$. The data heterogeneity lies in $\mathbf{w}_i$, details shown in Appendix 12.2. By using the aggregation probability (3), we demonstrate that under the same non-iid degree, $\mathrm{var}\{F_i(x_{EBA+})\}_{i\in m} \leq \mathrm{var}\{F_i(x_{Avg})\}_{i\in m}$, i.e., FedEBA+ achieves a smaller variance than FedAvg with uniform aggregation. The proof details are shown in Appendix 12.2.*

Extending the convergence analysis to a broader scenario with smooth and strongly convex loss functions, we demonstrate that $Var_{EBA+} \leq Var_{AVG}$. Refer to the settings, assumptions, and proof details in Appendix 12.3.

**Pareto-optimality analysis.** In addition to variance, *Pareto-optimality* can serve as another metric to assess fairness, as suggested by several studies (Wei & Niethammer, 2022; Hu et al., 2022). This metric achieves equilibrium by reaching each client's optimal performance without hindering others (Guardieiro et al., 2023). We prove that FedEBA+ achieves Pareto optimality through the entropy-based aggregation strategy.

**Definition 5.4** (Pareto optimality). *Suppose we have a group of $m$ clients in FL, and each client $i$ has a performance score $f_i$. Pareto optimality happens when we can't improve one client's performance without making someone else's worse: $\forall i \in [1, m], \exists j \in [1, m], j \neq i$ such that $f_i \leq f_i'$ and $f_j > f_j'$, where $f_i'$ and $f_j'$ represent the improved performance measures of participants $i$ and $j$, respectively.*

In the following proposition, we show that FedEBA+ satisfies Pareto optimality.

**Proposition 5.5** (Pareto optimality.). *The proposed maximum entropy model $\mathbb{H}(p)$ (2) is proven to be monotonically increasing under the given constraints, ensuring that the aggregation strategy $\varphi(p) = \arg\max_{p \in \mathcal{P}} h(p(f))$ is Pareto optimal. Here, $p(f)$ is the aggregation weights $p = [p_1, p_2, \ldots, p_m]$ of the loss function $f = [f_1, f_2, \ldots, f_m]$, and $h(\cdot)$ represents the entropy function. The proof can be found in Appendix 13.*

## 6 NUMERICAL RESULTS

In this section, we demonstrate FedEBA+'s superior performance over other baselines.

Table 1: **Performance of algorithms on FashionMNIST and CIFAR-10.** We report the accuracy of global model, variance fairness, worst 5%, and best 5% accuracy. The data is divided into 100 clients, with 10 clients sampled in each round. All experiments are running over 2000 rounds for a single local epoch ($K = 10$) with local batch size = 50, and learning rate $\eta = 0.1$. The reported results are averaged over 5 runs with different random seeds. We highlight the best and the second-best results by using **bold font** and blue text.

| Algorithm | FashionMNIST (MLP) | | | | CIFAR-10 (CNN) | | | |
|---|---|---|---|---|---|---|---|---|
| | Global Acc. ↑ | Var. ↓ | Worst 5% ↑ | Best 5% ↑ | Global Acc. ↑ | Var. ↓ | Worst 5% ↑ | Best 5% ↑ |
| FedAvg | 86.49 ± 0.09 | 62.44±4.55 | 71.27±1.14 | 95.84± 0.35 | 67.79±0.35 | 103.83±10.46 | 45.00±2.83 | 85.13±0.82 |
| FedSGD | 83.79 ±0.28 | 81.72 ±0.26 | 61.19 ±±0.30 | 96.60 ± ±0.20 | 67.48 ±0.37 | 95.79 ±4.03 | 48.70 ±±0.9 | 84.20 ±±0.40 |
| q-FFL$\|_{q=0.001}$ | 87.05± 0.25 | 66.67± 1.39 | 72.11± 0.03 | 95.09± 0.71 | 68.53± 0.18 | 97.42± 0.79 | 48.40± 0.60 | 84.70± 1.31 |
| q-FFL$\|_{q=15.0}$ | 75.77±0.42 | 46.58±0.75 | 61.63±0.46 | 89.60±0.42 | 36.89±0.14 | 79.65±5.17 | 19.30±0.70 | 51.30±0.09 |
| FedMGDA+$\|_{\epsilon=0.0}$ | 86.01±0.31 | 58.87±3.23 | 71.49±0.16 | 95.45±0.43 | 67.16±0.33 | 97.33±1.68 | 46.00±0.79 | 83.30±0.10 |
| FedMGDA+$\|_{\epsilon=0.03}$ | 84.64±0.25 | 57.89±6.21 | **73.49±1.17** | 93.22±0.20 | 65.19±0.87 | 89.78±5.87 | 48.84±1.12 | 81.94±0.67 |
| AFL$\|_{\lambda=0.7}$ | 85.14±0.18 | 57.39±6.13 | 70.09±0.69 | 95.94±0.09 | 66.21±1.21 | 79.75±1.25 | 47.54±0.61 | 82.08±0.77 |
| AFL$\|_{\lambda=0.5}$ | 84.14±0.18 | 90.76±3.33 | 60.11±0.58 | 96.00±0.09 | 65.11±2.44 | 86.19±9.46 | 44.73±3.90 | 82.10±0.62 |
| PropFair$\|_{M=0.2,thres=0.2}$ | 85.51±0.28 | 75.27±5.38 | 63.60±0.53 | 97.60±0.19 | 65.79±0.53 | 79.67±5.71 | 49.88±0.93 | 82.40±0.40 |
| PropFair$\|_{M=5.0,thres=0.2}$ | 84.59±1.01 | 85.31±8.62 | 61.40±0.55 | 96.40±0.29 | 66.91±1.43 | 78.90±6.48 | 50.16±0.56 | 85.40±0.34 |
| TERM$\|_{T=0.1}$ | 84.31±0.38 | 73.46±2.06 | 68.23±0.10 | 94.16±0.16 | 65.41±0.37 | 91.99±2.69 | 49.08±0.66 | 81.98±0.19 |
| TERM$\|_{T=0.5}$ | 82.19±1.41 | 87.82±2.62 | 62.11±0.71 | 93.25±0.39 | 61.04±1.96 | 96.78±7.67 | 42.45±1.73 | 80.06±0.62 |
| FedFV$\|_{\alpha=0.1,\tau_{fv}=1}$ | 86.51±0.28 | 49.73±2.26 | 71.33±1.16 | 95.89±0.23 | 68.94±0.27 | 90.84±2.67 | 50.53±4.33 | 86.00±1.23 |
| FedFV$\|_{\alpha=0.1,\tau_{fv}=10}$ | 86.98±0.45 | 56.63±1.85 | 66.40±0.57 | **98.80±0.12** | 71.10±0.44 | 86.50±7.36 | 49.80±0.72 | **88.42±0.25** |
| FedEBA+$\|_{\alpha=0,\tau=0.1}$ | 86.70±0.11 | 50.27±5.60 | 71.13±0.69 | 95.47±0.27 | 69.38±0.52 | 89.49±10.95 | 50.40±1.72 | 86.07±0.90 |
| FedEBA+$\|_{\alpha=0.5,\tau=0.1}$ | 87.21±0.06 | **40.02±1.58** | 73.07±1.03 | 95.81±0.14 | 72.39±0.47 | 70.60±3.19 | 55.27±1.18 | 86.27±1.16 |
| FedEBA+$\|_{\alpha=0.9,\tau=0.1}$ | **87.50±0.19** | 43.41±4.34 | 72.07±1.47 | 95.91±0.19 | **72.75±0.25** | 68.71±4.39 | **55.80±1.28** | 86.93±0.52 |

**Evaluation and datasets.** We conduct two metrics (1) *fairness*, including variance of accuracy, worst 5% accuracy and best 5% accuracy, and (2) *gloabl accuracy*, the test accuracy of the global model. We split MNIST, FashionMNIST, CIFAR-10, CIFAR-100, and Tiny-ImageNet into non-iid datasets following the setting of (Wang et al., 2021) and Latent Dirichlet Allocation (LDA), details shown in Appendix 14.

**Baseline algorithms.** We compare FedEBA+ with classical method FedAvg (McMahan et al., 2017) and FedSGD (McMahan et al., 2016), and fairness FL algorithms,including AFL (Mohri et al., 2019),q-FFL (Li et al., 2019a),FedMGDA+(Hu et al., 2022),Prop-Fair (Zhang et al., 2022), TERM (Li et al., 2020a) and FedFV (Wang et al., 2021). All algorithms are evaluated under the same settings, such as batch size and learning rate. Hyperparameters for baselines are listed in Table 3 in Appendix 14. We report results for the baselines using their best parameters, while complete results with various hyperparameter choices can be found in Table 4 in Appendix 14. More details about hyperparameters and experiment settings are available in Appendix 14.

**Table 1 and Table 2 illustrate FedEBA+ outperforms other baselines in terms of both fairness and global accuracy.** In detail,

- ***Variance of FedBEA+ is much smaller than others:*** FedEBA+ consistently achieves lower variance than others, indicating greater fairness. The fairness improvement is **3×** on FashionMNIST and **1.5×** on CIFAR-10 compared to the best-performing baseline.

- ***FedEBA+ addresses the accuracy-variance trade-off issue faced by other algorithms:*** FedEBA+ notably improves global accuracy, with a **4%** improvement on CIFAR-10 and **3%** on CIFAR-100 and Tiny-Imagenet. Other baselines either have lower global accuracy or show limited improvement compared to FedAvg. Additionally, the performance improvement of best 5% demonstrates that FedEBA+ reduces variance without compromising the performance of good-performing clients. To showcase the advantage of FedEBA+ when considering fairness and accuracy simultaneously, we use the coefficient of variation ($C_V = \frac{std}{acc}$) (Jain et al., 1984) to measure the relative fairness level, the performance of algorithms on $C_V$ is shown in Table 13 in Appendix 15.

- ***Ablation study of FedEBA+:*** In Table 1, FedEBA+$\|_{\alpha=0}$ differs solely in aggregation from FedAvg, highlighting the proposed aggregation approach's advantages. In contrast, FedEBA+$\|_{\alpha>0}$ incorporates the aligned update in addition to aggregation, showing its effectiveness through improved performance compared to FedEBA+$\|_{\alpha=0}$. Complete results of ablation study for FedEBA+ on four datasets are provided in Table 11 Appendix 15

**Figure 3 showcases the effectiveness of FedEBA+ in terms of accuracy, variance, and convergence performance on MNIST and CIFAR-10.** 1)Figure 3(a) illustrates FedEBA+'s dual improvement in global model performance and fairness. The algorithm's positioning near the lower right corner of the figure signifies superior performance in both

Table 2: **Performance of algorithms on CIFAR-100 and Tiny-ImageNet.** We exclude incompatible algorithms (FedMGDA+, PropFair, and TERM) under our experimental settings on these two datasets. Instead, we include SCAFFOLD (Karimireddy et al., 2020b) and FedProx (Li et al., 2020b) to compare their performance.

| Algorithm | CIFAR-100 (ResNet-18) | | | | Tiny-ImageNet (MobileNet-v2) | | | |
|---|---|---|---|---|---|---|---|---|
| | Global Acc. ↑ | Var. ↓ | Worst 5% ↑ | Best 5% ↑ | Global Acc.↑ | Var. ↓ | Worst 5% ↑ | Best 5% ↑ |
| FedAvg | 30.94±0.04 | 17.24±0.08 | 0.20±0.00 | 65.90±1.48 | 61.99±0.17 | 19.62±1.12 | 53.60±0.06 | 71.18±0.13 |
| q-FFL$\|_{q=0.01}$ | 24.97±0.46 | 14.54±0.21 | 0.00±0.00 | 45.04±0.53 | 62.42±0.46 | 15.44±1.89 | 54.13±0.11 | 70.01±0.09 |
| AFL$\|_{\lambda=0.01}$ | 20.84±0.43 | 11.32±0.20 | 4.03±0.14 | 50.83±0.30 | 62.09±0.53 | 16.47±0.88 | 54.65±0.64 | 68.83±1.30 |
| FedProx$\|_{\mu=0.1}$ | 31.50±0.04 | 17.50±0.09 | 0.41±0.00 | 64.50±0.11 | 62.05±0.04 | 16.21±1.13 | 54.41±0.47 | 69.92±0.26 |
| SCAFFOLD$\|_{\eta=1.0}$ | 31.81±0.02 | 17.52±0.20 | 1.59±0.01 | 68.36±0.23 | 63.62±0.02 | 15.52±1.49 | 54.76±0.71 | 70.47±0.12 |
| FedFV$\|_{\alpha=0.1,\tau_{fv}=1}$ | 31.23±0.04 | 17.50±0.02 | 0.20±0.00 | 66.05±0.11 | 62.13±0.08 | 15.69±0.58 | 53.92±0.30 | 69.60±0.31 |
| FedEBA+$\|_{\alpha=0.1,\tau=0.5}$ | **33.39±0.22** | 16.92±0.04 | 0.95±0.15 | **68.51±0.21** | **64.05±0.09** | 14.91±1.85 | 54.32±0.09 | **71.27±0.04** |
| FedEBA+$\|_{\alpha=0.9,\tau=0.1}$ | 31.98±0.30 | 13.75±0.16 | 1.12±0.05 | 67.94±0.54 | 63.75±0.09 | **13.89±0.72** | **55.64±0.18** | 70.93±0.22 |

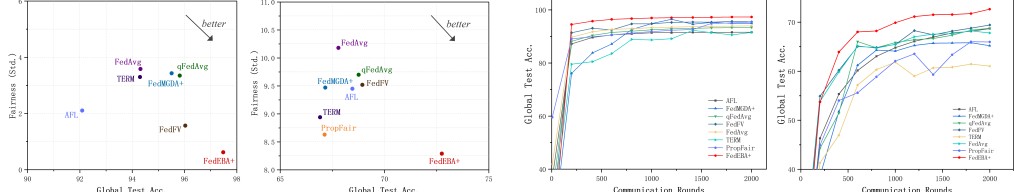

(a) **Performance of variance and accuracy**     (b) **Performance of convergence**

Figure 3: Performance of algorithms on (a) left: variance and accuracy on MNIST, (a) right: variance and accuracy on CIFAR-10, (b) left: convergence on MNIST, (b) right: convergence on CIFAR-10.

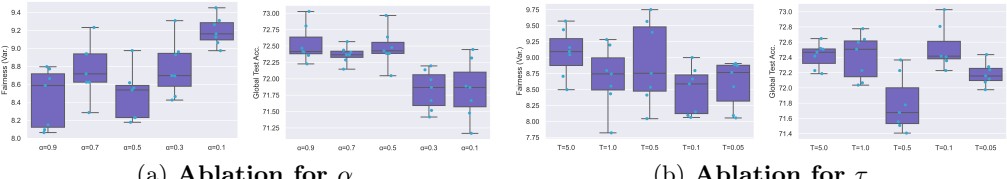

(a) **Ablation for $\alpha$**     (b) **Ablation for $\tau$**

Figure 4: Ablation study for hyperparameters

global accuracy and fairness. 2) Figure 3(b) illustrates the superior convergence performance of FedEBA+ compared to other fairness algorithms.

**Figure 4 demonstrates the monotonic stability of our algorithm to hyperparameters $\alpha$, while highlighting how parameter $\tau$ controls the balance between fairness and accuracy.** 1) Figure 4(a) shows fairness consistently improves as $\alpha$ increases, while accuracy steadily decreases. 2) Figure 4(b) shows the relationship between decreasing $\tau$ and improved fairness, while indicating that $\tau > 1$ generally leads to superior global accuracy.

**Additional ablation studies.** 1) Ablation study for communication cost (ablation for $\theta$) in Appendix 15 (Table 5 and Figures 9-12) demonstrates that FedEBA+ outperforms baselines at $\theta = 90°$, with the same communication cost as vanilla FedAvg, and exhibiting potential for further performance enhancement with $\theta < 90°$. 2) Ablation studies for Dirichlet parameter and annealing strategies are provided in Table 9, Figure 8 in Appendix 15.

**Additional results in Appendix 15 consistently demonstrate the superiority of FedEBA+,** including: 1) Performance table with full hyperparameter choices for algorithms (Table 4 for baselines and Table 12 for FedEBA+). 2) Performance comparison of FedEBA+ and baselines under local noisy label scenario (Table 10). 3) Performance of fairness algorithms integrated with advanced optimization methods like momentum (Table 7) and VARP (Table 8). 4) Performance results under cosine similarity and entropy metrics(Table 14).

## 7 CONCLUSIONS AND FUTURE WORKS

This paper introduces FedEBA+, a fair FL algorithm that enhances fairness and global model performance through innovative entropy-based aggregation and update alignment approaches. Theoretical analysis and experiments validate its superiority over SOTA baselines. Though experiments demonstrate that FedEBA+ outperforms baselines with the same communication costs as FedAvg, further performance improvements can be obtained by decreasing $\theta$. Therefore, integrating communication compression into FedEBA+ represents a valuable direction.

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

CONTENTS OF APPENDIX

## 8   AN EXPANDED VERSION OF THE RELATED WORK

**Fairness-Aware Federated Learning.** Various fairness concepts have been proposed in FL, including performance fairness (Li et al., 2019a; 2021; Wang et al., 2021; Zhao & Joshi, 2022; Kanaparthy et al., 2022; Huang et al., 2022), group fairness (Du et al., 2021; Ray Chaudhury et al., 2022), selection fairness (Zhou et al., 2021), and contribution fairness (Cong et al., 2020), among others (Shi et al., 2021; Wu et al., 2022; Chen et al., 2023). These concepts address specific aspects and stakeholder interests, making direct comparisons inappropriate. This paper specifically focuses on performance fairness, the most commonly used metric in FL, which serves client interests while improving model performance. We list and compare the commonly used fairness metrics of FL in Section 16.

To enhance performance fairness for FL, some works propose objective function-based approaches. In (Li et al., 2019a), q-FFL uses $\alpha$-fair allocation for balancing fairness and efficiency, but specific $\alpha$ choices may introduce bias. In contrast, FedEBA+ employs maximum entropy aggregation to accommodate diverse preferences. Additionally, FedEBA+ introduces a novel fair FL objective with dual-variable optimization, enhancing global model performance and variance. Besides, Deng et al. (2020) achieves fairness by defining a min-max optimization problem in FL. In the gradient-based approach, FedFV (Wang et al., 2021) mitigates gradient conflicts among FL clients to promote fairness. Efforts have been made to connect fairness and personalized FL to enhance robustness (Li et al., 2021; Lin et al., 2022). Recently, reweighting methods encourage a uniform performance by up-reweighting the importance of underperforming clients (Zhao & Joshi, 2022). However, these methods enhance fairness at the expense of the performance of the global model (Kanaparthy et al., 2022; Huang et al., 2022). In contrast, we propose FedEBA+ as a solution that significantly promotes fairness while improving the global model performance. Notably, FedEBA+ is orthogonal to existing optimization methods like momentum (Karimireddy et al., 2020a) and VARP (Jhunjhunwala et al., 2022), allowing seamless integration.

**Aggregation in Federated Optimization.** FL employs aggregation algorithms to combine decentralized data for training a global model (Kairouz et al., 2019). Approaches include federated averaging (FedAvg) McMahan et al. (2017), robust federated weighted averaging Pillutla et al. (2019); Laguel et al. (2021); Pillutla et al. (2023), importance aggregation Wang

et al. (2022), and federated dropout Zheng et al. (2022). However, these algorithms can be sensitive to the number and quality of participating clients, causing fairness issues (Li et al., 2019b; Balakrishnan et al., 2021; Shi et al., 2021). To the best of our knowledge, we are the first to analyze the aggregation from the view of entropy. Unlike heuristics that assign weights proportional to client loss (Zhao & Joshi, 2022; Kanaparthy et al., 2022), our method has physical meanings, i.e., the aggregation probability ensures that known constraints are as certain as possible while retaining maximum uncertainty for unknowns. By selecting the maximum entropy solution with constraints, we actually choose the solution that fits our information with the least deviation (Jaynes, 1957), thus achieving fairness.

## 9 ENTROPY ANALYSIS

### 9.1 DERIVATION OF PROPOSITION 4.1

In this section, we present the derivation of the maximum entropy distribution for the aggregation strategy of FedEBA+.

One may wonder why we propose an exponential formula treatment of loss function $p_i \propto e^{F_i(x)/\tau}$ instead of formulating the aggregation probability as something else, say, $p_i \propto F_i(x)$. In one word, because our aggregation strategy is the maximum entropy distribution.

Being a maximum entropy means minimizing the amount of prior information built to the distribution and guarantee that the selected probability distribution is devoid of subjective influences, thereby eradicating any inherent bias (Bian et al., 2021; Sampat & Zavala, 2019). Meanwhile, many physical systems tend to move towards maximal entropy configurations over time (Jaynes, 1957).

In the following we will give a derivation to show that $p_i \propto e^{F_i(x)/\tau}$ is indeed the maximum entropy distribution for FL. The derivation bellow is closely following (Jaynes, 1957) for statistical mechanics. Suppose the loss function of the user corresponding to the aggregation probability $p_i$ is $F_i(x)$. We would like to maximize the entropy $\mathbb{H}(p) = -\sum_{i=1}^{m} p_i \log p_i$, subject to FL constrains that $\sum_{i=1}^{m} p_i = 1, p_i \geq 0$, $\sum_i p_i F_i(x) = \tilde{f}(x)$, which means we constrain the average performance to be a constant, such as ideal global model performance or the ideal most fair average performance, independent of $p_i$.

*Proof.*

$$L\left(p, \lambda_0, \lambda_1\right) := -\left[\sum_{i=1}^{N} p_i \log p_i + \lambda_0 \left(\sum_{i=1}^{N} p_i - 1\right) + \lambda_1 \left(\mu - \sum_{i=1}^{N} p_i F_i(x)\right)\right],$$

where $\mu = f(x)$.

By setting

$$\frac{\partial L\left(p, \lambda_0, \lambda_1\right)}{\partial p_i} = -\left[\log p_i + 1 + \lambda_0 - \lambda_1 F_i(x)\right] = 0, \tag{13}$$

we get:

$$p_i = \exp\left[-\left(\lambda_0 + 1 - \lambda_1 F_i(x)\right)\right]. \tag{14}$$

According to $\sum_i p_i = 1$, we have:

$$\lambda_0 + 1 = \log \sum_{i=1}^{N} \exp\left(\lambda_1 F_i(x)\right) =: \log Z, \tag{15}$$

which is the log-partition function.

Thus,

$$p_i = \frac{\exp\left[\lambda_1 F_i(x)\right]}{Z}. \tag{16}$$

Note that the maximum value of the entropy is

$$H_{\max} = -\sum_{i=1}^{N} p_i \log p_i$$

$$= \lambda_0 + 1 - \lambda_1 \sum_{i=1}^{N} p_i F_i(x) \tag{17}$$

$$= \lambda_0 + 1 - \lambda_1 \mu \,.$$

So one can get,

$$\lambda_1 = -\frac{\partial H_{\max}}{\partial \mu} =: \frac{1}{\tau}\,, \tag{18}$$

which defines the inverse temperature. So we reach the exponential form of $p_i$ as:

$$p_i = \frac{\exp[F_i(x)/\tau]}{\sum_{i=1}^{N} \exp[F_i(x)/\tau]}\,. \tag{19}$$

$\square$

When taking into account the prior distribution of aggregation probability (Li et al., 2020b; Balakrishnan et al., 2021), which is typically expressed as $q_i = n_i/\sum_{i \in S_t} n_i$, the original entropy formula can be extended to include the prior distribution as follows:

$$\mathbb{H}(p) = \sum_{i=1}^{m} p_i \log(\frac{q_i}{p_i})\,. \tag{20}$$

Thus, the solution of the original problem under this prior distribution becomes:

$$p_i = \frac{q_i \exp[F_i(x)/\tau]}{\sum_{j=1}^{N} q_j \exp[F_j(x)/\tau]}\,. \tag{21}$$

*Proof.*

$$L\left(p, \lambda_0, \lambda_1\right) := -\sum_{i=1}^{N} p_i \log \frac{q_i}{p_i} + \lambda_0 \left(\sum_{i=1}^{N} p_i - 1\right) + \lambda_1 \left(\mu - \sum_{i=1}^{N} p_i F_i(x)\right)\,. \tag{22}$$

Following similar derivation steps, let

$$\frac{\partial L\left(p, \lambda_0, \lambda_1\right)}{\partial p_i} = -\log(q_i) + \log(p_i) + 1 + \lambda_0 - \lambda_1 F_i(x) = 0\,, \tag{23}$$

we get:

$$p_i = \exp\left[-\left(\lambda_0 + 1 - \log(q_i) - \lambda_1 F_i(x)\right)\right]\,. \tag{24}$$

According to $\sum_i p_i = 1$, we have:

$$\sum_i p_i = \sum_i \exp\left[-\left(\lambda_0 + 1 - \log(q_i) - \lambda_1 F_i(x)\right)\right] = 1\,. \tag{25}$$

Therefore, we get:

$$\lambda_0 + 1 = \log \sum_{i=1}^{N} q_i \exp\left(\lambda_1 F_i(x)\right) =: \log(Z)\,. \tag{26}$$

Then substituting $\lambda_0 + 1 = \log(Z)$ back to $p_i = \exp\left[-\left(\lambda_0 + 1 - \log(q_i) - \lambda_1 F_i(x)\right)\right]$, we get $p_i = \frac{q_i \exp[\lambda_1 F_i(x)]}{Z}$.

By setting $\lambda_1 =: \frac{1}{\tau}$, we obtain (21):

$$p_i = \frac{q_i \exp[F_i(x)/\tau]}{\sum_{j=1}^{N} q_j \exp[F_j(x)/\tau]}\,. \tag{27}$$

$\square$

## 10 Convergence Analysis of FedEBA+

In this section, we give the proof of Theorem 5.1.

Before going to the details of our convergence analysis, we first state the key lemmas used in our proof, which helps us to obtain the advanced convergence result.

**Lemma 10.1.** *To make this paper self-contained, we restate the Lemma 3 in (Wang et al., 2020):*

*For any model parameter $\boldsymbol{x}$, the difference between the gradients of $f(\boldsymbol{x})$ and $\tilde{f}(\boldsymbol{x})$ can be bounded as follows:*

$$\|\nabla f(\boldsymbol{x}) - \nabla \tilde{f}(\boldsymbol{x})\|^2 \leq \chi^2_{\boldsymbol{w}\|\boldsymbol{p}} \left[ A^2 \|\nabla \tilde{f}(\boldsymbol{x})\|^2 + \kappa^2 \right] , \tag{28}$$

*where $\chi^2_{\boldsymbol{w}\|\boldsymbol{p}}$ denotes the chi-square distance between $\boldsymbol{w}$ and $\boldsymbol{p}$, i.e., $\chi^2_{\boldsymbol{w}\|\boldsymbol{p}} = \sum_{i=1}^{m} (w_i - p_i)^2 / p_i$. $f(x)$ is the global objective with $f(x) = \sum_{i=1}^{m} w_i f_i(x)$ where $\boldsymbol{w}$ is usually average of all clients, i.e., $\boldsymbol{w} = [\frac{1}{m}, \cdots, \frac{1}{m}]$. $\tilde{f}(x) = \sum_{i=1}^{m} p_i f_i(x)$ is the surrogate objective with the reweight aggregation probability $\boldsymbol{p}$.*

*Proof.*

$$\begin{aligned}
\nabla f(x) - \nabla \tilde{f}(\boldsymbol{x}) &= \sum_{i=1}^{m} (w_i - p_i) \nabla f_i(\boldsymbol{x}) \\
&= \sum_{i=1}^{m} (w_i - p_i) \left( \nabla f_i(\boldsymbol{x}) - \nabla \tilde{f}(\boldsymbol{x}) \right) \\
&= \sum_{i=1}^{m} \frac{w_i - p_i}{\sqrt{p_i}} \cdot \sqrt{p_i} \left( \nabla f_i(\boldsymbol{x}) - \nabla \tilde{f}(\boldsymbol{x}) \right) .
\end{aligned} \tag{29}$$

Applying Cauchy-Schwarz inequality, it follows that

$$\begin{aligned}
\|\nabla f(x) - \nabla \tilde{f}(\boldsymbol{x})\|^2 &\leq \left[ \sum_{i=1}^{m} \frac{(w_i - p_i)^2}{p_i} \right] \left[ \sum_{i=1}^{m} p_i \left\| \nabla f_i(x) - \nabla \tilde{f}(\boldsymbol{x}) \right\|^2 \right] \\
&\leq \chi^2_{\boldsymbol{w}\|\boldsymbol{p}} \left[ A^2 \|\nabla \tilde{f}(\boldsymbol{x})\|^2 + \sigma_G^2 \right] ,
\end{aligned} \tag{30}$$

where the last inequality uses Assumption 3. Note that

$$\begin{aligned}
\|\nabla f(\boldsymbol{x})\|^2 &\leq 2\|\nabla f(\boldsymbol{x}) - \nabla \tilde{f}(\boldsymbol{x})\|^2 + 2\|\nabla \tilde{f}(\boldsymbol{x})\|^2 \\
&\leq 2 \left[ \chi^2_{\boldsymbol{w}\|\boldsymbol{p}} A^2 + 1 \right] \|\nabla \tilde{f}(\boldsymbol{x})\|^2 + 2\chi^2_{\boldsymbol{p}\|\boldsymbol{w}^2} \sigma_G^2 .
\end{aligned} \tag{31}$$

As a result, we obtain

$$\begin{aligned}
\min_{t \in [T]} \|\nabla f(\boldsymbol{x}_t)\|^2 &\leq \frac{1}{T} \sum_{t=0}^{T-1} \|\nabla f(\boldsymbol{x}_t)\|^2 \\
&\leq 2 \left[ \chi^2_{\boldsymbol{w}\|\boldsymbol{p}} A^2 + 1 \right] \frac{1}{T} \sum_{t=0}^{T-1} \left\| \nabla \tilde{f}(\boldsymbol{x}_t) \right\|^2 + 2\chi^2_{\boldsymbol{w}\|\boldsymbol{p}} \sigma_G^2 \\
&\leq 2 \left[ \chi^2_{\boldsymbol{w}\|\boldsymbol{p}} A^2 + 1 \right] \epsilon_{\text{opt}} + 2\chi^2_{\boldsymbol{w}\|\boldsymbol{p}} \sigma_G^2 ,
\end{aligned} \tag{32}$$

where $\epsilon_{\text{opt}} = \frac{1}{T} \sum_{t=0}^{T-1} \left\| \nabla \tilde{f}(\boldsymbol{x}_t) \right\|^2$ denotes the optimization error.

$\square$

10.1 ANALYSIS WITH $\alpha = 0$.

**Lemma 10.2** (Local updates bound.). *For any step-size satisfying $\eta_L \leq \frac{1}{8LK}$, we can have the following results:*

$$\mathbb{E}\|x_{i,k}^t - x_t\|^2 \leq 5K(\eta_L^2\sigma_L^2 + 4K\eta_L^2\sigma_G^2) + 20K^2(A^2 + 1)\eta_L^2\|\nabla f(x_t)\|^2. \tag{33}$$

*Proof.*

$$\mathbb{E}_t\|x_{t,k}^i - x_t\|^2$$
$$= \mathbb{E}_t\|x_{t,k-1}^i - x_t - \eta_L g_{t,k-1}^t\|^2$$
$$= \mathbb{E}_t\|x_{t,k-1}^i - x_t - \eta_L(g_{t,k-1}^t - \nabla F_i(x_{t,k-1}^i) + \nabla F_i(x_{t,k-1}^i) - \nabla F_i(x_t) + \nabla F_i(x_t))\|^2$$
$$\leq (1 + \frac{1}{2K-1})\mathbb{E}_t\|x_{t,k-1}^i - x_t\|^2 + \mathbb{E}_t\|\eta_L(g_{t,k-1}^t - \nabla F_i(x_{t,k}^i))\|^2$$
$$+ 4K\mathbb{E}_t[\|\eta_L(\nabla F_i(x_{t,K-1}^i) - \nabla F_i(x_t))\|^2] + 4K\eta_L^2\mathbb{E}_t\|\nabla F_i(x_t)\|^2$$
$$\leq (1 + \frac{1}{2K-1})\mathbb{E}_t\|x_{t,k-1}^i - x_t\|^2 + \eta_L^2\sigma_L^2 + 4K\eta_L^2L^2\mathbb{E}_t\|x_{t,k-1}^i - x_t\|^2$$
$$+ 4K\eta_L^2\sigma_G^2 + 4K\eta_L^2(A^2 + 1)\|\nabla\tilde{f}(x_t)\|^2$$
$$\leq (1 + \frac{1}{K-1})\mathbb{E}\|x_{t,k-1}^i - x_t\|^2 + \eta_L^2\sigma_L^2 + 4K\eta_L^2\sigma_G^2 + 4K(A^2 + 1)\|\eta_L\nabla\tilde{f}(x_t)\|^2. \tag{34}$$

Unrolling the recursion, we obtain:

$$\mathbb{E}_t\|x_{t,k}^i - x_t\|^2 \leq \sum_{p=0}^{k-1}(1 + \frac{1}{K-1})^p\left[\eta_L^2\sigma_L^2 + 4K\eta_L^2\sigma_G^2 + 4K(A^2 + 1)\|\eta_L\nabla\tilde{f}(x_t)\|^2\right]$$
$$\leq (K-1)\left[(1 + \frac{1}{K-1})^K - 1\right]\left[\eta_L^2\sigma_L^2 + 4K\eta_L^2\sigma_G^2 + 4K(A^2 + 1)\|\eta_L\nabla\tilde{f}(x_t)\|^2\right]$$
$$\leq 5K(\eta_L^2\sigma_L^2 + 4K\eta_L^2\sigma_G^2) + 20K^2(A^2 + 1)\eta_L^2\|\nabla\tilde{f}(x_t)\|^2. \tag{35}$$

$\square$

Thus, we can have the following convergence rate of FedEBA+:

**Theorem 10.3.** *Under Assumption 1–3, and let constant local and global learning rate $\eta_L$ and $\eta$ be chosen such that $\eta_L < min\left(1/(8LK), C\right)$, where $C$ is obtained from the condition that $\frac{1}{2} - 10L^2\frac{1}{m}\sum_{i-1}^m K^2\eta_L^2(A^2 + 1)(\chi_{\boldsymbol{w}\|\boldsymbol{p}}^2 A^2 + 1) > c > 0$ ,and $\eta \leq 1/(\eta_L L)$, the expected gradient norm of FedEBA+ with $\alpha = 0$, i.e., only using aggregation strategy 3, is bounded as follows:*

$$\min_{t\in[T]} \mathbb{E}\|\nabla f(x_t)\|^2 \leq 2\left[\chi_{\boldsymbol{w}\|\boldsymbol{p}}^2 A^2 + 1\right](\frac{f_0 - f_*}{c\eta\eta_L KT} + \Phi) + 2\chi_{\boldsymbol{w}\|\boldsymbol{p}}^2\sigma_G^2, \tag{36}$$

*where*

$$\Phi = \frac{1}{c}[\frac{5\eta_L^2 KL^2}{2}(\sigma_L^2 + 4K\sigma_G^2) + \frac{\eta\eta_L L}{2}\sigma_L^2 + 20L^2K^2(A^2 + 1)\eta_L^2\chi_{\boldsymbol{w}\|\boldsymbol{p}}^2\sigma_G^2]. \tag{37}$$

*where $c$ is a constant, $\chi_{\boldsymbol{w}\|\boldsymbol{p}}^2 = \sum_{i=1}^m (w_i - p_i)^2/p_i$ represents the chi-square divergence between vectors $\boldsymbol{p} = [p_1, \ldots, p_m]$ and $\boldsymbol{w} = [w_1, \ldots, w_m]$. For common FL algorithms with uniform aggregation or with data ratio as aggregation probability, $w_i = \frac{1}{m}$ or $w_i = \frac{n_i}{N}$.*

*Proof.* Based on Lemma 10.1, we first focus on analyzing the optimization error $\epsilon_{opt}$:

$$\mathbb{E}_t[\tilde{f}(x_{t+1})] \overset{(a1)}{\leq} \tilde{f}(x_t) + \left\langle \nabla\tilde{f}(x_t), \mathbb{E}_t[x_{t+1} - x_t] \right\rangle + \frac{L}{2}\mathbb{E}_t[\|x_{t+1} - x_t\|^2]$$

$$= \tilde{f}(x_t) + \left\langle \nabla\tilde{f}(x_t), \mathbb{E}_t[\eta\Delta_t + \eta\eta_L K\nabla\tilde{f}(x_t) - \eta\eta_L K\nabla\tilde{f}(x_t)] \right\rangle + \frac{L}{2}\eta^2\mathbb{E}_t[\|\Delta_t\|^2]$$

$$= \tilde{f}(x_t) - \eta\eta_L K\left\|\nabla\tilde{f}(x_t)\right\|^2 + \eta\underbrace{\left\langle \nabla\tilde{f}(x_t), \mathbb{E}_t[\Delta_t + \eta_L K\nabla\tilde{f}(x_t)] \right\rangle}_{A_1} + \frac{L}{2}\eta^2\underbrace{\mathbb{E}_t\|\Delta_t\|^2}_{A_2},$$

$$\tag{38}$$

where (a1) follows from the Lipschitz continuity condition. Here, the expectation is over the local data SGD and the filtration of $x_t$. However, in the next analysis, the expectation is over all randomness, including client sampling. This is achieved by taking expectation on both sides of the above equation over client sampling.

To begin with, we consider $A_1$:

$$A_1 = \left\langle \nabla\tilde{f}(x_t), \mathbb{E}_t[\Delta_t + \eta_L K\nabla\tilde{f}(x_t)] \right\rangle$$

$$= \left\langle \nabla\tilde{f}(x_t), \mathbb{E}_t[-\sum_{i=1}^{m} w_i \sum_{k=0}^{K-1} \eta_L g_{t,k}^i + \eta_L K\nabla\tilde{f}(x_t)] \right\rangle$$

$$\overset{(a2)}{=} \left\langle \nabla\tilde{f}(x_t), \mathbb{E}_t[-\sum_{i=1}^{m} w_i \sum_{k=0}^{K-1} \eta_L \nabla F_i(x_{t,k}^i) + \eta_L K\nabla\tilde{f}(x_t)] \right\rangle$$

$$= \left\langle \sqrt{\eta_L K}\nabla\tilde{f}(x_t), -\frac{\sqrt{\eta_L}}{\sqrt{K}}\mathbb{E}_t[\sum_{i=1}^{m} w_i \sum_{k=0}^{K-1} (\nabla F_i(x_{t,k}^i) - \nabla F_i(x_t))] \right\rangle$$

$$\overset{(a3)}{=} \frac{\eta_L K}{2}\|\nabla\tilde{f}(x_t)\|^2 + \frac{\eta_L}{2K}\mathbb{E}_t\left\|\sum_{i=1}^{m} w_i \sum_{k=0}^{K-1} (\nabla F_i(x_{t,k}^i) - \nabla F_i(x_t))\right\|^2$$

$$- \frac{\eta_L}{2K}\mathbb{E}_t\|\sum_{i=1}^{m} w_i \sum_{k=0}^{K-1} \nabla F_i(x_{t,k}^i)\|^2$$

$$\overset{(a4)}{\leq} \frac{\eta_L K}{2}\|\nabla\tilde{f}(x_t)\|^2 + \frac{\eta_L}{2}\sum_{k=0}^{K-1}\sum_{i=1}^{m} w_i \mathbb{E}_t\left\|\nabla F_i(x_{t,k}^i) - \nabla F_i(x_t)\right\|^2$$

$$- \frac{\eta_L}{2K}\mathbb{E}_t\|\sum_{i=1}^{m} w_i \sum_{k=0}^{K-1} \nabla F_i(x_{t,k}^i)\|^2$$

$$\overset{(a5)}{\leq} \frac{\eta_L K}{2}\|\nabla\tilde{f}(x_t)\|^2 + \frac{\eta_L L^2}{2m}\sum_{i=1}^{m}\sum_{k=0}^{K-1} \mathbb{E}_t\left\|x_{t,k}^i - x_t\right\|^2 - \frac{\eta_L}{2K}\mathbb{E}_t\|\sum_{i=1}^{m} w_i \sum_{k=0}^{K-1} \nabla F_i(x_{t,k}^i)\|^2$$

$$\leq \left(\frac{\eta_L K}{2} + 10K^3 L^2 \eta_L^3 (A^2 + 1)\right)\|\nabla\tilde{f}(x_t)\|^2 + \frac{5L^2\eta_L^3}{2}K^2\sigma_L^2 + 10\eta_L^3 L^2 K^3 \sigma_G^2$$

$$- \frac{\eta_L}{2K}\mathbb{E}_t\|\sum_{i=1}^{m} w_i \sum_{k=0}^{K-1} \nabla F_i(x_{t,k}^i)\|^2, \tag{39}$$

where (a2) follows from Assumption 2. (a3) is due to $\langle x, y \rangle = \frac{1}{2}\left[\|x\|^2 + \|y\|^2 - \|x - y\|^2\right]$ and (a4) uses Jensen's Inequality: $\|\sum_{i=1}^{m} w_i z_i\|^2 \leq \sum_{i=1}^{m} w_i \|z_i\|^2$, (a5) comes from Assumption 1.

Then we consider $A_2$:

$$A_2 = \mathbb{E}_t \|\Delta_t\|^2$$

$$= \mathbb{E}_t \left\| \eta_L \sum_{i=1}^{m} w_i \sum_{k=0}^{K-1} g_{t,k}^i \right\|^2$$

$$= \eta_L^2 \mathbb{E}_t \left\| \sum_{i=1}^{m} w_i \sum_{k=0}^{K-1} g_{t,k}^i - \sum_{i=1}^{m} w_i \sum_{k=0}^{K-1} \nabla F_i(x_{t,k}^i) \right\|^2 + \eta_L^2 \mathbb{E}_t \left\| \sum_{i=1}^{m} w_i \sum_{k=0}^{K-1} \nabla F_i(x_{t,k}^i) \right\|^2$$

$$\leq \eta_L^2 \sum_{i=1}^{m} w_i \sum_{k=0}^{K-1} \mathbb{E}\|g_i(x_{t,k}^i) - \nabla F_i(x_{t,k}^i)\|^2 + \eta_L^2 \mathbb{E}_t \| \sum_{i=1}^{m} w_i \sum_{k=0}^{K-1} \nabla F_i(x_{t,k}^i)\|^2$$

$$\overset{(a6)}{\leq} \eta_L^2 K \sigma_L^2 + \eta_L^2 \mathbb{E}_t \| \sum_{i=1}^{m} w_i \sum_{k=0}^{K-1} \nabla F_i(x_{t,k}^i)\|^2 \,. \tag{40}$$

where (a6) follows from Assumption 2.

Now we substitute the expressions for $A_1$ and $A_2$ and take the expectation over the client aggregation distribution on both sides to make $\tilde{f}(x)$ to $f(x)$. It should be noted that the derivation of $A_1$ and $A_2$ above is based on considering the expectation over the sampling distribution:

$$f(x_{t+1}) \leq f(x_t) - \eta \eta_L K \mathbb{E}_t \left\| \nabla \tilde{f}(x_t) \right\|^2 + \eta \mathbb{E}_t \left\langle \nabla \tilde{f}(x_t), \Delta_t + \eta_L K \nabla \tilde{f}(x_t) \right\rangle + \frac{L}{2} \eta^2 \mathbb{E}_t \|\Delta_t\|^2$$

$$\overset{(a7)}{\leq} f(x_t) - \eta \eta_L K \left( \frac{1}{2} - 20 L^2 K^2 \eta_L^2 (A^2 + 1)(\chi_{\boldsymbol{w}\|\boldsymbol{p}}^2 A^2 + 1) \right) \mathbb{E}_t \left\| \nabla \tilde{f}(x_t) \right\|^2$$

$$+ \frac{5 \eta \eta_L^3 L^2 K^2}{2} (\sigma_L^2 + 4K \sigma_G^2) + \frac{\eta^2 \eta_L^2 K L}{2} \sigma_L^2 + 20 L^2 K^3 (A^2 + 1) \eta \eta_L^3 \chi_{\boldsymbol{w}\|\boldsymbol{p}}^2 \sigma_G^2$$

$$- \left( \frac{\eta \eta_L}{2K} - \frac{L \eta^2 \eta_L^2}{2} \right) \mathbb{E}_t \left\| \frac{1}{m} \sum_{i=1}^{m} \sum_{k=0}^{K-1} \nabla F_i(x_{t,k}^i) \right\|^2$$

$$\overset{(a8)}{\leq} f(x_t) - c \eta \eta_L K \mathbb{E} \left\| \nabla \tilde{f}(x_t) \right\|^2 + \frac{5 \eta \eta_L^3 L^2 K^2}{2} (\sigma_L^2 + 4K \sigma_G^2)$$

$$+ \frac{\eta^2 \eta_L^2 K L}{2} \sigma_L^2 + 20 L^2 K^3 (A^2 + 1) \eta \eta_L^3 \chi_{\boldsymbol{w}\|\boldsymbol{p}}^2 \sigma_G^2 - \left( \frac{\eta \eta_L}{2K} - \frac{L \eta^2 \eta_L^2}{2} \right) \mathbb{E}_t \left\| \frac{1}{m} \sum_{i=1}^{m} \sum_{k=0}^{K-1} \nabla F_i(x_{t,k}^i) \right\|^2$$

$$\overset{(a9)}{\leq} f(x_t) - c \eta \eta_L K \mathbb{E}_t \|\nabla \tilde{f}(x_t)\|^2 + \frac{5 \eta \eta_L^3 L^2 K^2}{2} (\sigma_L^2 + 4K \sigma_G^2)$$

$$+ \frac{\eta^2 \eta_L^2 K L}{2} \sigma_L^2 + 20 L^2 K^3 (A^2 + 1) \eta \eta_L^3 \chi_{\boldsymbol{w}\|\boldsymbol{p}}^2 \sigma_G^2 \,, \tag{41}$$

where (a7) is due to Lemma 10.1, (a8) holds because there exists a constant $c > 0$ (for some $\eta_L$) satisfying $\frac{1}{2} - 10 L^2 \frac{1}{m} \sum_{i-1}^{m} K^2 \eta_L^2 (A^2 + 1)(\chi_{\boldsymbol{w}\|\boldsymbol{p}}^2 A^2 + 1) > c > 0$, and the (a9) follows from $\left( \frac{\eta \eta_L}{2K} - \frac{L \eta^2 \eta_L^2}{2} \right) \geq 0$ if $\eta \eta_l \leq \frac{1}{KL}$.

Rearranging and summing from $t = 0, \ldots, T - 1$, we have:

$$\sum_{t=1}^{T-1} c \eta \eta_L K \mathbb{E} \|\nabla \tilde{f}(x_t)\|^2 \leq f(x_0) - f(x_T) + T(\eta \eta_L K)\Phi \,. \tag{42}$$

Which implies:

$$\frac{1}{T} \sum_{t=1}^{T-1} \mathbb{E} \|\nabla \tilde{f}(x_t)\|^2 \leq \frac{f_0 - f_*}{c \eta \eta_L K T} + \Phi \,, \tag{43}$$

where

$$\Phi = \frac{1}{c}[\frac{5\eta_L^2 K L^2}{2}(\sigma_L^2 + 4K\sigma_G^2) + \frac{\eta\eta_L L}{2}\sigma_L^2 + 20L^2 K^2(A^2 + 1)\eta_L^2\chi_{\boldsymbol{w}\|\boldsymbol{p}}^2\sigma_G^2]. \tag{44}$$

Then, given the result of $\epsilon_{opt}$, we can derive the convergence rate of $\|\nabla f(x_t)\|$ by substitute $\epsilon_{opt}$ back to (32):

$$\min_{t\in[T]} \|\nabla f(\boldsymbol{x}_t)\|^2 \le 2\left[\chi_{\boldsymbol{w}\|\boldsymbol{p}}^2 A^2 + 1\right]\epsilon_{\text{opt}} + 2\chi_{\boldsymbol{w}\|\boldsymbol{p}}^2\sigma_G^2 \tag{45}$$

$$\le 2\left[\chi_{\boldsymbol{w}\|\boldsymbol{p}}^2 A^2 + 1\right](\frac{f_0 - f_*}{c\eta\eta_L KT} + \Phi) + 2\chi_{\boldsymbol{w}\|\boldsymbol{p}}^2\sigma_G^2. \tag{46}$$

**Corollary 10.4.** *Suppose $\eta_L$ and $\eta$ are $\eta_L = \mathcal{O}\left(\frac{1}{\sqrt{TKL}}\right)$ and $\eta = \mathcal{O}\left(\sqrt{Km}\right)$ such that the conditions mentioned above are satisfied. Then for sufficiently large $T$, the iterates of FedEBA+ with $\alpha = 0$ satisfy:*

$$\min_{t\in[T]} \|\nabla f(\boldsymbol{x}_t)\|^2 \le \mathcal{O}\left(\frac{(f^0 - f^*)}{\sqrt{mKT}}\right) + \mathcal{O}\left(\frac{\sqrt{m}\sigma_L^2}{2\sqrt{KT}}\right) + \mathcal{O}\left(\frac{5(\sigma_L^2 + 4K\sigma_G^2)}{2KT}\right)$$
$$+ \mathcal{O}\left(\frac{20(A^2 + 1)\chi_{\boldsymbol{w}\|\boldsymbol{p}}^2\sigma_G^2}{T}\right) + 2\chi_{\boldsymbol{w}\|\boldsymbol{p}}^2\sigma_G^2. \tag{47}$$

$\square$

## 10.2 ANALYSIS WITH $\alpha \ne 0$

To derivate the convergence rate of FedEBA+ with $\alpha \ne 0$, we need the following assumption:

**Assumption 4** (Distance bound between practical aggregated gradient and ideal fair gradient). *In each round, we assume the aggregated gradient $\nabla\overline{f}(x_t) = \sum_{i\in S_t} p_i F_i(x_t)$ and the fair gradient $\nabla\tilde{f}(x_t)$ is bounded: $\mathbb{E}\|\nabla\overline{f}(x_t) - \nabla\tilde{f}(x_t)\|^2 \le \rho^2, \forall i, t.$*

To simplify the notation, we define $h_{t,k}^i = (1 - \alpha)\nabla F_i(x_{t,k}^i) + \alpha\nabla\overline{f}(x_t)$.

**Lemma 10.5.** *For any step-size satisfying $\eta_L \le \frac{1}{8LK}$, we can have the following results:*

$$\mathbb{E}\|x_{i,k}^t - x_t\|^2 \le 5K(1 - \alpha)^2(\eta_L^2\sigma_L^2 + 6K\eta_L^2\sigma_G^2) + +30K^2\eta_L^2\alpha^2\rho^2$$
$$+ 30K^2\eta_L^2(1 + A^2(1 - \alpha)^2)\|\nabla\tilde{f}(x_t)\|^2. \tag{48}$$

*Proof.*

$$\mathbb{E}_t\|x_{t,k}^i - x_t\|^2$$
$$= \mathbb{E}_t\|x_{t,k-1}^i - x_t - \eta_L h_{t,k-1}^t\|^2$$
$$= \mathbb{E}_t\|x_{t,k-1}^i - x_t - \eta_L((1 - \alpha)g_{t,k-1}^t + \alpha\nabla\overline{f}(x_t) - (1 - \alpha)\nabla F_i(x_{t,k-1}^i)$$
$$+ (1 - \alpha)\nabla F_i(x_{t,k-1}^i) - (1 - \alpha)\nabla F_i(x_t) + (1 - \alpha)\nabla F_i(x_t) + \nabla\tilde{f}(x_t) - \nabla\tilde{f}(x_t))\|^2$$
$$+ 4K\mathbb{E}_t[\|\eta_L(\nabla F_i(x_{t,K-1}^i) - \nabla F_i(x_t))\|^2] + 4K\eta_L^2\mathbb{E}_t\|\nabla F_i(x_t)\|^2$$
$$\le (1 + \frac{1}{2K - 1})\mathbb{E}_t\|x_{t,k-1}^i - x_t\|^2 + (1 - \alpha)^2\eta_L^2\sigma_L^2 + 6K\eta_L^2 L^2\mathbb{E}_t\|x_{t,k-1}^i - x_t\|^2$$
$$+ 6K\eta_L^2\alpha^2\mathbb{E}\|\nabla\overline{f}(x_t) - \nabla\tilde{f}(x_t)\|^2 + 6K\eta_L^2(1 - \alpha)^2(\sigma_G^2 + A^2\|\nabla\tilde{f}(x_t)\|^2) + 6K\eta_L^2\|\nabla\tilde{f}(x_t)\|^2$$
$$\le (1 + \frac{1}{K - 1})\mathbb{E}_t\|x_{t,k-1}^i - x_t\|^2 + (1 - \alpha)^2\eta_L^2\sigma_L^2$$
$$+ 6K\eta_L^2\alpha^2\rho^2 + 6K\eta_L^2(1 - \alpha)^2(\sigma_G^2 + A^2\|\nabla\tilde{f}(x_t)\|^2) + 6K\eta_L^2\|\nabla\tilde{f}(x_t)\|^2, \tag{49}$$

Unrolling the recursion, we obtain:

$$
\mathbb{E}_t \|x_{t,k}^i - x_t\|^2
$$

$$
\leq \sum_{p=0}^{k-1} (1 + \frac{1}{K-1})^p \left( (1-\alpha)^2 \eta_L^2 \sigma_L^2 + 6K(1-\alpha)^2 \eta_L^2 \sigma_G^2 + 6K\alpha^2 \eta_L^2 \rho^2 \right.
$$

$$
\left. + 6K\eta_L^2 (A^2(1-\alpha)^2 + 1)\|\nabla \tilde{f}(x_t)\|^2 \right)
$$

$$
\leq (K-1) \left[ (1 + \frac{1}{K-1})^K - 1 \right] \left[ (1-\alpha)^2 \eta_L^2 \sigma_L^2 \right.
$$

$$
\left. + 6K(1-\alpha)^2 \eta_L^2 \sigma_G^2 + 6K\alpha^2 \eta_L^2 \rho^2 + 6K\eta_L^2 (A^2(1-\alpha)^2 + 1)\|\nabla \tilde{f}(x_t)\|^2 \right]
$$

$$
\leq 5K\eta_L^2 (1-\alpha)^2 (\sigma_L^2 + 6K\sigma_G^2) + 30K^2 \eta_L^2 \alpha^2 \rho^2 + 30K^2 \eta_L^2 (A^2(1-\alpha)^2 + 1)\|\nabla \tilde{f}(x_t)\|^2 . \quad (50)
$$

Similarly, to get the convergence rate of objective $f(x_t)$, we first focus on $\tilde{f}(x_t)$:

$$
\mathbb{E}_t[\tilde{f}(x_{t+1})] \stackrel{(a1)}{\leq} \tilde{f}(x_t) + \left\langle \nabla \tilde{f}(x_t), \mathbb{E}_t[x_{t+1} - x_t] \right\rangle + \frac{L}{2} \mathbb{E}_t[\|x_{t+1} - x_t\|^2]
$$

$$
= \tilde{f}(x_t) + \left\langle \nabla \tilde{f}(x_t), \mathbb{E}_t[\eta \Delta_t + \eta \eta_L K \nabla \tilde{f}(x_t) - \eta \eta_L K \nabla \tilde{f}(x_t)] \right\rangle + \frac{L}{2} \eta^2 \mathbb{E}_t[\|\Delta_t\|^2]
$$

$$
= \tilde{f}(x_t) - \eta \eta_L K \left\| \nabla \tilde{f}(x_t) \right\|^2 + \eta \underbrace{\left\langle \nabla \tilde{f}(x_t), \mathbb{E}_t[\Delta_t + \eta_L K \nabla \tilde{f}(x_t)] \right\rangle}_{A_1} + \frac{L}{2} \eta^2 \underbrace{\mathbb{E}_t\|\Delta_t\|^2}_{A_2}, \quad (51)
$$

where (a1) follows from the Lipschitz continuity condition. Here, the expectation is over the local data SGD and the filtration of $x_t$. However, in the next analysis, the expectation is over all randomness, including client sampling. This is achieved by taking expectation on both sides of the above equation over client sampling.

To begin with, we consider $A_1$:

$$A_1 = \left\langle \nabla \tilde{f}(x_t), \mathbb{E}_t[\Delta_t + \eta_L K \nabla \tilde{f}(x_t)] \right\rangle$$

$$= \left\langle \nabla \tilde{f}(x_t), \mathbb{E}_t[-\sum_{i=1}^{m} w_i \sum_{k=0}^{K-1} \eta_L h_{t,k}^i + \eta_L K \nabla \tilde{f}(x_t)] \right\rangle$$

$$\overset{(a2)}{=} \left\langle \nabla \tilde{f}(x_t), \mathbb{E}_t[-\sum_{i=1}^{m} w_i \sum_{k=0}^{K-1} \eta_L [(1-\alpha)\nabla F_i(x_{t,k}^i) + \alpha \overline{f}(x_t)] + \eta_L K \nabla \tilde{f}(x_t)] \right\rangle$$

$$= \langle \sqrt{\eta_L K} \nabla \tilde{f}(x_t),$$

$$- \frac{\sqrt{\eta_L}}{\sqrt{K}} \mathbb{E}_t \left( \sum_{i=1}^{m} w_i \sum_{k=0}^{K-1} (1-\alpha)[\nabla F_i(x_{t,k}^i) - \nabla \tilde{f}(x_t)] + \sum_{i=1}^{m} w_i \sum_{k=0}^{K-1} \alpha[\nabla \overline{f}(x_t) - \nabla \tilde{f}(x_t)] \right) \rangle$$

$$\overset{(a3)}{=} \frac{\eta_L K}{2} \|\nabla \tilde{f}(x_t)\|^2 - \frac{\eta_L}{2K} \mathbb{E}_t \| \sum_{i=1}^{m} w_i \sum_{k=0}^{K-1} [(1-\alpha)\nabla F_i(x_{t,k}^i) + \alpha \nabla \overline{f}(x_t)]\|^2$$

$$+ \frac{\eta_L}{2K} \mathbb{E}_t \left\| \sum_{i=1}^{m} w_i \sum_{k=0}^{K-1} \left( (1-\alpha)[\nabla F_i(x_{t,k}^i) - \nabla \tilde{f}(x_t)] + \alpha[\nabla \overline{f}(x_t) - \nabla \tilde{f}(x_t)] \right) \right\|^2$$

$$\overset{(a4)}{\leq} \frac{\eta_L K}{2} \|\nabla \tilde{f}(x_t)\|^2 + \frac{\eta_L (1-\alpha)^2}{2} \sum_{k=0}^{K-1} \sum_{i=1}^{m} w_i \mathbb{E}_t \left\| \nabla F_i(x_{t,k}^i) - \nabla F_i(x_t) \right\|^2$$

$$+ \frac{\eta_L \alpha^2}{2} \sum_{k=0}^{K-1} \sum_{i=1}^{m} w_i \mathbb{E} \|\nabla \overline{f}(x_t) - \nabla \tilde{f}(x_t)\|^2 - \frac{\eta_L}{2K} \mathbb{E}_t \| \sum_{i=1}^{m} w_i \sum_{k=0}^{K-1} [(1-\alpha)\nabla F_i(x_{t,k}^i) + \alpha \nabla \overline{f}(x_t)]\|^2$$

$$\overset{(a5)}{\leq} \frac{\eta_L K}{2} \|\nabla \tilde{f}(x_t)\|^2 + \frac{\eta_L (1-\alpha)^2 L^2}{2m} \sum_{i=1}^{m} \sum_{k=0}^{K-1} \mathbb{E}_t \left\| x_{t,k}^i - x_t \right\|^2$$

$$+ \frac{\eta_L \alpha^2}{2m} \sum_{i=1}^{m} \sum_{k=0}^{K-1} \mathbb{E} \|\nabla \overline{f}(x_t) - \nabla \tilde{f}(x_t)\|^2 - \frac{\eta_L}{2K} \mathbb{E}_t \| \sum_{i=1}^{m} w_i \sum_{k=0}^{K-1} [(1-\alpha)\nabla F_i(x_{t,k}^i) + \alpha \nabla \overline{f}(x_t)]\|^2$$

$$\leq \frac{\eta_L K}{2} \|\nabla \tilde{f}(x_t)\|^2 + \frac{\eta_L (1-\alpha)^2}{2m} \sum_{i=1}^{m} \sum_{k=0}^{K-1} \left( 5K\eta_L (1-\alpha)^2 (\sigma_L^2 + 6K\sigma_G^2) + 30K^2 \eta_L^2 [\alpha^2 \rho^2 \right.$$

$$\left. + (1 + A^2(1-\alpha)^2)]\|\nabla \tilde{f}(x_t\|^2] \right) + \frac{\eta_L^2 \alpha^2}{2} K\rho^2 - \frac{\eta_L}{2K} \mathbb{E}\| \sum_{i=1}^{m} w_i \sum_{k=0}^{K-1} [(1-\alpha)\nabla F_i(x_{t,k}^i) + \alpha \nabla \overline{f}(x_t)]\|^2 ,$$

$$\tag{52}$$

where (a2) follows from Assumption 2. (a3) is due to $\langle x, y \rangle = \frac{1}{2}\left[\|x\|^2 + \|y\|^2 - \|x-y\|^2\right]$ and (a4) uses Jensen's Inequality: $\|\sum_{i=1}^{m} w_i z_i\|^2 \leq \sum_{i=1}^{m} w_i \|z_i\|^2$, (a5) comes from Assumption 1.

Then we consider $A_2$:

$$A_2 = \mathbb{E}_t \|\Delta_t\|^2$$

$$= \mathbb{E}_t \left\| \eta_L \sum_{i=1}^{m} w_i \sum_{k=0}^{K-1} h_{t,k}^i \right\|^2$$

$$= \eta_L^2 \mathbb{E}_t \left\| \sum_{i=1}^{m} w_i \sum_{k=0}^{K-1} \left[ (1-\alpha)\nabla F_i(x_{t,k}^i; \xi_t^i) + \alpha \overline{f}(x_t) \right] \right\|^2$$

$$\leq \eta_L^2 \mathbb{E}\| \sum_{i=1}^{m} w_i \sum_{k=0}^{K-1} \left[ (1-\alpha)\nabla F_i(x_{t,k}^i; \xi_t^i) + \alpha \overline{f}(x_t) \right]$$

$$- (1-\alpha)\nabla F_i(x_{t,k}^i) + (1-\alpha)\nabla F_i(x_{t,k}^i)\|^2$$

$$\overset{(a6)}{\leq} \eta_L^2 K(1-\alpha)^2 \sigma_L^2 + \eta_L^2 \mathbb{E}\| \sum_{i=1}^{m} w_i \sum_{k=0}^{K-1} [(1-\alpha)\nabla F_i(x_{t,k}^i) + \alpha \nabla \overline{f}(x_t)]\|^2 \tag{53}$$

where (a6) follows from Assumption 2.

Now we substitute the expressions for $A_1$ and $A_2$ and take the expectation over the client sampling distribution on both sides. It should be noted that the derivation of $A_1$ and $A_2$ above is based on considering the expectation over the sampling distribution:

$$
f(x_{t+1}) \leq f(x_t) - \eta\eta_L K \mathbb{E}_t \left\| \nabla \tilde{f}(x_t) \right\|^2 + \eta \mathbb{E}_t \left\langle \nabla \tilde{f}(x_t), \Delta_t + \eta_L K \nabla \tilde{f}(x_t) \right\rangle + \frac{L}{2}\eta^2 \mathbb{E}_t \|\Delta_t\|^2
$$

$$
\overset{(a7)}{\leq} f(x_t) - \eta\eta_L K \left( \frac{1}{2} - 30\alpha^2 L^2 K^2 \eta_L^2 ((1-\alpha)^2 A^2 + 1) \right) \mathbb{E} \left\| \nabla \tilde{f}(x_t) \right\|^2
$$

$$
+ \frac{5(1-\alpha)^2 \eta\eta_L^3 L^2 K^2}{2} \left[ 5(1-\alpha)^2(\sigma_L^2 + 6K\sigma_G^2) + 30K\alpha^2\rho^2 \right] + \frac{\eta\eta_L^2\alpha^2}{2} K\rho^2
$$

$$
+ \frac{L\eta^2\eta_L^2}{2}(1-\alpha)^2 K\sigma_L^2 - \left( \frac{\eta\eta_L}{2K} - \frac{\eta^2\eta_L^2 L}{2} \right) \mathbb{E} \| \sum_{i=1}^{m} w_i \sum_{k=0}^{K-1} [(1-\alpha)\nabla F_i(x_{t,k}^i) + \alpha\nabla\overline{f}(x_t)] \|^2
$$

$$
\tag{54}
$$

where $\frac{1}{2} - 15\alpha^2 L^2 K^2 \eta_L^2 ((1-\alpha)^2 A^2 + 1) > c > 0$ and $\frac{\eta\eta_L}{2K} - \frac{\eta\eta_L^2 L}{2} \geq 0$.

Rearranging and summing from $t = 0, \ldots, T-1$, we have:

$$
\sum_{t=1}^{T-1} c\eta\eta_L K \mathbb{E}\|\nabla\tilde{f}(x_t)\|^2 \leq f(x_0) - f(x_T) + T(\eta\eta_L K)\Phi . \tag{55}
$$

Which implies:

$$
\frac{1}{T} \sum_{t=1}^{T-1} \mathbb{E}\|\nabla\tilde{f}(x_t)\|^2 \leq \frac{f_0 - f_*}{c\eta\eta_L KT} + \tilde{\Phi} , \tag{56}
$$

where

$$
\tilde{\Phi} = \frac{1}{c} \left[ \frac{5\eta_L^2 KL^2(1-\alpha)^4}{2}(\sigma_L^2 + 6K\sigma_G^2) + 15K^2\eta_L^2(1-\alpha)^2\alpha^2\rho^2 + \frac{\eta\eta_L L(1-\alpha)^2}{2}\sigma_L^2 + \frac{\eta_L\alpha^2\rho^2}{2} \right] . \tag{57}
$$

Then, given the result of $\epsilon_{opt}$, we can derive the convergence rate of $\|\nabla f(x_t)\|$ by substitute $\epsilon_{opt}$ back to (32):

$$
\min_{t \in [T]} \|\nabla f(\boldsymbol{x}_t)\|^2 \leq 2 \left[ \chi_{\boldsymbol{w}\|\boldsymbol{p}}^2 A^2 + 1 \right] \epsilon_{\text{opt}} + 2\chi_{\boldsymbol{w}\|\boldsymbol{p}}^2 \sigma_G^2 \tag{58}
$$

$$
\leq 2 \left[ \chi_{\boldsymbol{w}\|\boldsymbol{p}}^2 A^2 + 1 \right] \left( \frac{f_0 - f_*}{c\eta\eta_L KT} + \tilde{\Phi} \right) + 2\chi_{\boldsymbol{w}\|\boldsymbol{p}}^2 \sigma_G^2 . \tag{59}
$$

$\square$

**Corollary 10.6.** *Suppose $\eta_L$ and $\eta$ are $\eta_L = \mathcal{O}\left(\frac{1}{\sqrt{T}KL}\right)$ and $\eta = \mathcal{O}\left(\sqrt{Km}\right)$ such that the conditions mentioned above are satisfied. Then for sufficiently large $T$, the iterates of FedEBA+ with $\alpha \neq 0$ satisfy:*

$$
\min_{t \in [T]} \|\nabla f(\boldsymbol{x}_t)\|^2 \leq \mathcal{O}\left( \frac{(f^0 - f^*)}{\sqrt{mKT}} \right) + \mathcal{O}\left( \frac{(1-\alpha)^2\sqrt{n}\sigma_L^2}{2\sqrt{KT}} \right) + \mathcal{O}\left( \frac{5(1-\alpha)^2(\sigma_L^2 + 6K\sigma_G^2)}{2KT} \right)
$$

$$
+ \mathcal{O}\left( \frac{15(1-\alpha)^2\alpha^2\rho^2}{T} \right) + \mathcal{O}\left( \frac{\alpha^2\rho^2}{2\sqrt{T}K} \right) + 2\chi_{\boldsymbol{w}\|\boldsymbol{p}}^2 \sigma_G^2 . \tag{60}
$$

## 11 UNIQUENESS OF OUR AGGREGATION STRATEGY

In this section, we prove the proposed entropy-based aggregation strategy is unique.

Recall our optimization objective of constrained maximum entropy:

$$
H(p(x)) = -\sum (p(x)\log(p(x))) , \tag{61}
$$

subject to certain contains, which is $\sum_i p_i = 1, p_i \geq 0, \sum_i p_i f_i = \tilde{f}$.

Based on equation 3, and writing the entropy in matrix form, we have:

$$H_{i,j}(p) = \begin{cases} p_i(\frac{f_i}{\tau} - \log \sum e^{f_i/\tau}) = -ap_i & \text{for } i = j \\ 0 & \text{otherwise} \end{cases}, \tag{62}$$

where $a$ is some positive constant.

For every non-zero vector $v$ we have that:

$$v^T H(p) v = \sum_{j \in \mathcal{N}} -ap_i v_j^2 < 0. \tag{63}$$

The Hessian is thus negative definite.

Furthermore, since the constraints are linear, both convex and concave, the constrained maximum entropy function is strictly concave and thus has a unique global maximum.

## 12 FAIRNESS ANALYSIS VIA VARIANCE

To demonstrate the ability of FedEBA+ to enhance fairness in federated learning, we first employ a two-user toy example to demonstrate how FedEBA+ can achieve a more balanced performance between users in comparison to FedAvg and q-FedAvg, thus ensuring fairness. Additionally, we use a general class of regression models to show how FedEBA+ reduces the variance among users and thus improves fairness. Similarly, to simplify the analysis, we consider the degenerate scenario of FedEBA+ where the parameter $\alpha = 0$.

### 12.1 TOY CASE FOR ILLUSTRATING FAIRNESS

In this section, we examine the performance fairness of our algorithm. In particular, we consider two clients participating in training, each with a regression model: $f_1(x_t) = 2(x-2)^2$, $f_1(x_t) = \frac{1}{2}(x+4)^2$. Corresponding,

$$\nabla f_1(x_t) = 4(x - 2), \tag{64}$$

$$\nabla f_2(x_t) = (x + 4). \tag{65}$$

When the global model parameter $x_t = 0$ is sent to each client, each client will update the model by running gradient decent, here w.l.o.g, we consider one single-step gradient decent, and stepsize $\lambda = \frac{1}{4}$:

$$x_1^{t+1} = x_t - \lambda \nabla f_1(x_t) = 2. \tag{66}$$
$$x_2^{t+1} = x_t - \lambda \nabla f_2(x_t) = -1. \tag{67}$$

Thus, for uniform aggregation:

$$x_{uniform}^{t+1} = \frac{1}{2}(x_1^{t+1} + x_2^{t+1}) = \frac{1}{2}. \tag{68}$$

While for FedEBA+:

$$x_{EBA+}^{t+1} = \frac{e^{f_1(x_1^{t+1})}}{e^{f_1(x_1^{t+1})} + e^{f_2(x_2^{t+1})}} x_1^{t+1} + \frac{e^{f_2(x_2^{t+1})}}{e^{f_1(x_1^{t+1})} + e^{f_2(x_2^{t+1})}} x_2^{t+1} \approx -0.1. \tag{69}$$

Therefore,

$$\mathbf{Var}_{uniform} = \frac{1}{2} \sum_{i=1}^{2} \left( f_i(x_{uniform}^{t+1}) - \frac{1}{2} \sum_{i=1}^{2} (f_i(x_{uniform}^{t+1})) \right) = 2 * (2.81)^2. \tag{70}$$

$$\mathbf{Var}_{EBA+} = \frac{1}{2} \sum_{i=1}^{2} \left( f_i(x_{EBA+}^{t+1}) - \frac{1}{2} \sum_{i=1}^{2} (f_i(x_{EBA+}^{t+1})) \right) = 2 * (0.6)^2. \tag{71}$$

Thus, we prove that FedEBA+ achieves a much smaller variance than uniform aggregation.

Furthermore, for q-FedAvg, we consider $q = 2$ that is also used in the proof of (Li et al., 2019a):

$$\nabla x_1^t = L(x^t - x_1^{t+1}) = -2 \,. \tag{72}$$

$$\nabla x_2^t = L(x^t - x_2^{t+1}) = 1 \,. \tag{73}$$

Thus, we have:

$$\Delta_1^t = f_1^q(x_t)\nabla x_1^t = 8 * (-2) = -16 \,. \tag{74}$$

$$h_1^t = q f_1^{q-1}(x_t)\|\nabla x_1^t\|^2 + L f_1^q(x_t) = 1 \times 1 \times 2^2 + 8 = 12 \,. \tag{75}$$

$$\Delta_2^t = f_2^q(x_t)\nabla x_2^t = 8 * (1) = 8 \,. \tag{76}$$

$$h_2^t = q f_2^{q-1}(x_t)\|\nabla x_2^t\|^2 + L f_2^q(x_t) = 1 \times 1 \times 1^2 + 8 = 9 \,. \tag{77}$$

Finally, we can update the global parameter as:

$$x_{qfedavg}^{t+1} = x^t - \frac{\sum_i \Delta_i^t}{\sum_i h_i^t} \approx -0.4 \,. \tag{78}$$

Then we can easily get:

$$\mathbf{Var}_{qfedavg} = \frac{1}{2}\sum_{i=1}^{2}\left(f_i(x_{qfedavg}^{t+1}) - \frac{1}{2}\sum_{i=1}^{m}(f_i(x_{qfedavg}^{t+1}))\right) = 2 * (2.52)^2$$

In conclusion, we prove that

$$\mathbf{Var}_{EBA+} \leq \mathbf{Var}_{qfedavg} \leq \mathbf{Var}_{uniform} \,. \tag{79}$$

## 12.2 Analysis Fairness by Generalized Linear Regression Model

**Our setting.** In this section, we consider a generalized linear regression setting, which follows from that in (Lin et al., 2022).

Suppose that the true parameter on client $i$ is $\mathbf{w}_i$, and there are $n$ samples on each client. The observations are generated by $\hat{y}_{i,k}(\mathbf{w_i}, \xi_k) = T(\xi_{i,k})^\top \mathbf{w}_i - A(\xi_{i,k})$, where the $A(\xi_{i,k})$ are i.i.d and distributed as $\mathcal{N}\left(0, \sigma_1^2\right)$. Then the loss on client $i$ is $F_i(\mathbf{x}_i) = \frac{1}{2n}\sum_{k=1}^{n}\left(T(\xi_{i,k})^\top \mathbf{x}_i - A(\xi_{i,k}) - \hat{y}_{i,k}\right)^2$.

We compare the performance of fairness of different aggregation methods. Recall Defination 3.1. We measure performance fairness in terms of the variance of the test accuracy/losses.

**Solutions of different methods** First, we derive the solutions of different methods. Let $\boldsymbol{\Xi}_i = (T(\xi_{i,1}), T(\xi_{i,2}), \ldots, T(\xi_{i,n}))^\top$, $\mathbf{A}_i = (A(\xi_{i,1}), A(\xi_{i,2}), \ldots, A(\xi_{i,n}))^\top$ and $\mathbf{y}_i = (y_{i,1}, y_{i,2}, \ldots, y_{i,n})^\top$. Then the loss on client $i$ can be rewritten as $F_i(\mathbf{x}_i) = \frac{1}{2n}\|\boldsymbol{\Xi}_i \mathbf{x}_i - \mathbf{A}_i - \mathbf{y}_i\|_2^2$, where $\text{rank}(\boldsymbol{\Xi}_i) = d$. The least-square estimator of $\mathbf{w}_i$ is

$$\left(\boldsymbol{\Xi}_i^\top \boldsymbol{\Xi}_i\right)^{-1}\boldsymbol{\Xi}_i^\top(\mathbf{y}_i + \mathbf{A}_i) \,. \tag{80}$$

*FedAvg:* For FedAvg, the solution is defined as $\mathbf{w}^{\text{Avg}} = \text{argmin}_{\mathbf{w}\in\mathbb{R}^d}\frac{1}{m}\sum_{i=1}^{m}F_i(\mathbf{w})$. One can check that $\mathbf{w}^{\text{Avg}} = \left(\sum_{i=1}^{m}\boldsymbol{\Xi}_i^\top\boldsymbol{\Xi}_i\right)^{-1}\sum_{i=1}^{m}\boldsymbol{\Xi}_i^\top(\mathbf{y}_i + \mathbf{A}_i) = \left(\sum_{i=1}^{m}\boldsymbol{\Xi}_i^\top\boldsymbol{\Xi}_i\right)^{-1}\sum_{i=1}^{m}\boldsymbol{\Xi}_i^\top\boldsymbol{\Xi}_i\hat{\mathbf{w}}_i + \Lambda$, where $\Lambda = \left(\sum_{i=1}^{m}\boldsymbol{\Xi}_i^\top\boldsymbol{\Xi}_i\right)^{-1}\sum_{i=1}^{m}\boldsymbol{\Xi}_i^\top A_i$ and $\hat{\mathbf{w}}_i = \text{argmin}_{\mathbf{x}\in\mathbb{R}^d}f_i(x_i)$ is the solution on client $i$.

*FedEBA+:* For our method FedEBA+, the solution of the global model is $\mathbf{w}^{\text{EBA+}} = \text{argmin}_{\mathbf{w}\in\mathbb{R}^d}\sum_{i=1}^{m}p_i F_i(\mathbf{w}) = \left(\sum_{i=1}^{m}p_i\boldsymbol{\Xi}_i^\top\boldsymbol{\Xi}_i\right)^{-1}\sum_{i=1}^{m}p_i\boldsymbol{\Xi}_i^\top\boldsymbol{\Xi}_i\hat{\mathbf{w}}_i + \hat{\Lambda}$, where $p_i \propto e^{F_i}(\mathbf{w_i})$, and $\hat{\Lambda} = \left(\sum_{i=1}^{m}p_i\boldsymbol{\Xi}_i^\top\boldsymbol{\Xi}_i\right)^{-1}\sum_{i=1}^{m}p_i\boldsymbol{\Xi}_i^\top A_i$

Following the setting of (Lin et al., 2022), to make the calculations clean, we assume $\boldsymbol{\Xi}_i^\top \boldsymbol{\Xi}_i = nb_i \boldsymbol{I}_d$. Then the solutions of different methods can be simplified as

- FedAvg: $\mathbf{w}^{\text{Avg}} = \frac{\sum_{i=1}^m b_i(\hat{\mathbf{w}}_i + A_i)}{\sum_{i=1}^m b_i}$.
- FedEBA+: $\mathbf{w}^{\text{Avg}} = \frac{\sum_{i=1}^m b_i p_i(\hat{\mathbf{w}}_i + A_i)}{\sum_{i=1}^m b_i p_i}$.

**Test Loss** We compute the test losses of different methods. In this part, we assume $b_i = b$ to make calculations clean. This is reasonable since we often normalize the data.

Recall that the dataset on client $i$ is $(\boldsymbol{\Xi}_i, \mathbf{y}_i)$, where $\boldsymbol{\Xi}_i$ is fixed and $\mathbf{y}_i$ follows Gaussian distribution $\mathcal{N}\left(\boldsymbol{\Xi}_i \mathbf{w}_i, \sigma_2^2 \boldsymbol{I}_n\right)$. Then the data heterogeneity across clients only lies in the heterogeneity of $\mathbf{w}_i$. Besides, since distribution of $\Lambda$ also follows gaussian distribution $\mathcal{N}\left(0, \sigma_1^2 \boldsymbol{I}_n\right)$, thus $w_i + A_i$ follows from $\mathcal{N}\left(\boldsymbol{\Xi}_i \mathbf{w}_i, \sigma^2 \boldsymbol{I}_n\right)$, where $\sigma^2 = \sigma_1^2 + \sigma_2^2$. Then, we can obtain the distribution of the solutions of different methods. Let $\overline{\mathbf{w}} = \frac{\sum_{i=1}^N \mathbf{w}_i}{N}$. We have

- FedAvg: $\mathbf{w}^{\text{Avg}} \sim \mathcal{N}\left(\overline{\mathbf{w}}, \frac{\sigma^2}{bNn} \boldsymbol{I}_d\right)$.

- FedEBA+: $\mathbf{w}^{\text{EBA+}} \sim \mathcal{N}\left(\tilde{\mathbf{w}}, \sum_{i=1}^N p_i^2 \frac{\sigma^2}{bn} \boldsymbol{I}_d\right)$, where $\tilde{\mathbf{w}} = \sum_{i=1}^N p_i w_i$.

Since $\boldsymbol{\Xi}_i$ is fixed, we assume the test data is $(\boldsymbol{\Xi}_i, \mathbf{y}_i')$ where $\mathbf{y}_i' = \boldsymbol{\Xi}_i \mathbf{w}_i + \mathbf{z}_i'$ with $\mathbf{z}_i' \sim \mathcal{N}\left(\mathbf{0}_n, \sigma_z^2 \boldsymbol{I}_n\right)$ independent of $\mathbf{z}_i$. Then the test loss on client $k$ is defined as:

$$
\begin{aligned}
F_i^{\text{te}}(\mathbf{x}_i) &= \frac{1}{2n} \mathbb{E} \left\| \boldsymbol{\Xi}_i \mathbf{x}_i + A_i - \mathbf{y}_i' \right\|_2^2 \\
&= \frac{1}{2n} \mathbb{E} \left\| \boldsymbol{\Xi}_i \mathbf{x}_i + A_i - (\boldsymbol{\Xi}_i \mathbf{w}_i + \mathbf{z}_i') \right\|_2^2 \\
&= \frac{\tilde{\sigma}^2}{2} + \frac{1}{2n} \mathbb{E} \left\| \boldsymbol{\Xi}_i (\mathbf{x}_i - \mathbf{w}_i) \right\|_2^2 \\
&= \frac{\tilde{\sigma}^2}{2} + \frac{b}{2} \mathbb{E} \left\| \mathbf{x}_i - \mathbf{w}_i \right\|_2^2 \\
&= \frac{\tilde{\sigma}^2}{2} + \frac{b}{2} \operatorname{tr}\left(\operatorname{var}(\mathbf{x}_i)\right) + \frac{b}{2} \left\| \mathbb{E} \mathbf{x}_i - \mathbf{w}_i \right\|_2^2 .
\end{aligned}
\tag{81}
$$

where $\tilde{\sigma}$ is a Gaussian variance, which comes from the fact that both $A_i$ and $z_i'$ follow Gaussian distribution with mean 0.

Therefore, for different methods, we can compute that

$$
f_i^{\text{te}}\left(\mathbf{w}^{\text{Avg}}\right) = \frac{\tilde{\sigma}^2}{2} + \frac{\tilde{\sigma}^2 d}{2Nn} + \frac{b}{2} \left\| \overline{\mathbf{w}} - \mathbf{w}_i \right\|_2^2
\tag{82}
$$

$$
f_i^{\text{te}}\left(\mathbf{w}^{\text{EBA+}}\right) = \frac{\tilde{\sigma}^2}{2} + \sum_{k=1}^N p_i^2 \frac{\tilde{\sigma}^2 d}{2n} + \frac{b}{2} \left\| \tilde{\mathbf{w}} - \mathbf{w}_i \right\|_2^2 .
\tag{83}
$$

Define var as the variance operator. Then we give the formal version of Theorem 5.3.

The variance of test losses on different clients of different aggregation methods are as follows:

$$
V^{\text{Avg}} = \operatorname{var}\left(f_i^{te}\left(\mathbf{w}^{Avg}\right)\right) = \frac{b^2}{4} \operatorname{var}\left(\left\| \overline{\mathbf{w}} - \mathbf{w}_i \right\|_2^2\right) .
\tag{84}
$$

$$
V^{\text{EBA+}} = \operatorname{var}\left(f_i^{te}\left(\mathbf{w}^{EBA+}\right)\right) = \frac{b^2}{4} \operatorname{var}\left(\left\| \tilde{\mathbf{w}} - \mathbf{w}_i \right\|_2^2\right) .
\tag{85}
$$

Based on a simple fact: assign larger weights to smaller values and smaller weights to larger values, and give a detailed mathematical proof to show that the variance of such a distribution is smaller than the variance of a uniform distribution. Which means $V^{\text{EBA+}} \le V^{\text{Avg}}$.

Formally, let $\|\tilde{\mathbf{w}} - \mathbf{w}_i\|^2 = A_i$. From equation (83), we know that $f_i^{te}(\mathbf{w}^{\mathbf{EBA+}}) \propto A_i$, and $p_i \propto f_i$. Thus, we know $p_i \propto A_i$.

Then, we consider the expression of $V^{\mathrm{EBA+}} = \frac{b^2}{4} var_i(A_i)$. Assume $A_i = [A_1 > A_2 > \cdots > A_m]$, then the corresponding aggregation probability distribution is $[p_1 > p_2 > \cdots > p_m]$.

We show the analysis of variance with set size 2, while the analysis can be easily extended to the number $K$. For FedEBA+, we have

$$var_i(A_i) = \sum_{i=1}^{m} p_i \left( A_i - \sum_i p_i A_i \right)^2 \tag{86}$$

$$= p_1(A_1 - (p_1 A_1 + p_2 A_2))^2 + p_2(A_2 - (p_1 A_1 + p_2 A_2))^2 \tag{87}$$

$$= p_1(1 - p_1)^2 A_1^2 - 2(1 - p_1)p_1 p_2 A_1 A_2 + p_1 p_2^2 A_2^2$$
$$+ p_2(1 - p_2)^2 A_2^2 - 2(1 - p_2)p_1 p_2 A_1 A_2 + p_1^2 p_2 A_1^2 \tag{88}$$

$$= (p_1 p_2^2 + p_1^2 p_2)A_1^2 - 2p_1 p_2(2 - p_1 - p_2)A_1 A_2 + (p_1 p_2^2 + p_1^2 p_2)A_2^2 \tag{89}$$

$$\overset{(a1)}{=} p_1 p_2(A_1^2 + A_2^2) - 2p_1 p_2 A_1 A_2 \tag{90}$$

$$= p_1 p_2(A_1 - A_2)^2 \,, \tag{91}$$

where $(a1)$ follows from the fact $\sum_i p_i = 1$.

According to our previous analysis, $p_1 > p_2$ while $A_1 > A_2$. According to Cauchy-Schwarz inequality, one can easily prove that $p_1 p_2 \leq \frac{1}{4}$, where $\frac{1}{4}$ comes from uniform aggregation.

Therefore, we prove that $V^{\mathrm{EBA+}} \leq V^{\mathrm{Avg}}$.

### 12.3 FAIRNESS ANALYSIS BY SMOOTH AND STRONGLY CONVEX LOSS FUNCTIONS.

In this section, we define the test loss on client $i$ as $L(x_i)$, to distinguish it from the training loss $F_i(x_i)$.

To extend the analysis to a more general case, we first introduce the following assumptions:

**Assumption 5** (Smooth and strongly convex loss functions). *The loss function $L_i(x)$ for each client is $L$-smooth,*

$$\|\nabla L_i(x)\|_2 \leq L \,, \tag{92}$$

*and $\mu$-strongly convex:*

$$L(y) \geq L(x) + <\nabla L(x), y - x> + \frac{1}{2}\mu\|y - x\|^2 \,. \tag{93}$$

The variance of FedAvg with $N$ clients loss can be formulated as:

$$V_N^{Avg} = \frac{1}{N}\sum_{i=1}^{N} L_i^2(x) - (\frac{1}{N}\sum_{i=1}^{N} L_i(x))^2. \tag{94}$$

For FedEBA+, the variance can be formulated with a similar form, only different in client's loss $L_i(\tilde{x})$, abbreviated as $\tilde{L}_i$. Then, the variance of FedEBA+ with $N$ clients can be formulated as:

$$V_N^{EBA+} = \frac{1}{N}\sum_{i=1}^{N} \tilde{L}_i^2 - (\frac{1}{N}\sum_{i=1}^{N} \tilde{L}_i)^2. \tag{95}$$

When client number is $N+1$, abbreviate FedAvg's loss $L_i(x)$ as $L_i$, we conclude

$$
\begin{aligned}
V_N^{Avg} &= \frac{1}{N+1}\sum_{i=1}^{N+1} L_i^2 - \left(\frac{1}{N+1}\sum_{i=1}^{N+1} L_i\right)^2 \\
&= \frac{N}{N+1}\frac{1}{N}\left(L_1^2 + L_2^2 + \cdots + L_{N+1}^2\right) - \left[\frac{N}{N+1}\frac{1}{N}\left(L_1 + L_2 + \cdots + L_{N+1}\right)\right]^2 \\
&= \frac{N}{N+1}\frac{1}{N}\left[\left(L_1^2 + L_2^2 + \cdots + L_N^2\right) + L_{N+1}^2\right] - \left[\frac{N}{N+1}\left(\frac{L_1 + L_2 + \cdots + L_N}{N} + \frac{L_{N+1}}{N}\right)\right]^2 \\
&= \left(\frac{N}{N+1}\right)^2\left[\frac{N+1}{N}\frac{\sum_{i=1}^{N} L_i^2}{N} - \left(\frac{1}{N}\sum_{i=1}^{N} L_i\right)^2\right] \\
&\quad + \frac{1}{N+1}L_{N+1}^2 - \frac{L_{N+1}^2}{(N+1)^2} - 2(\frac{N}{N+1})^2\frac{\sum_{i=1}^{N} L_i}{N}\frac{L_{N+1}}{N} \\
&= (\frac{N}{N+1})^2\frac{1}{N}\frac{\sum_{i=1}^{N} L_i^2}{N} + \frac{N}{N+1}V_N + \frac{1}{N+1}L_{N+1}^2 \\
&\quad - \frac{1}{(N+1)^2}L_{N+1}^2 - 2(\frac{N}{N+1})^2\frac{\sum_{i=1}^{N} L_i}{N}\frac{L_{N+1}}{N} \\
&= \frac{N}{N+1}V_N + \frac{L_1^2 + \cdots + L_N^2}{(N+1)^2} + + \frac{NL_{N+1}^2}{(N+1)^2} - \frac{2(L_1 + \cdots + _N)L_{N+1}}{(N+1)^2} \\
&= \frac{N}{N+1}V_N + \frac{\sum_{i=1}^{N}(L_i - L_{N+1})^2}{(N+1)^2} .
\end{aligned}
\tag{96}
$$

We start proving $V_N^{Avg} \geq V_N^{EBA+}, \forall N$ by considering a special case with two clients: There are two clients, Client 1 and Client 2, each with local model $x_1, x_2$ and training loss $F_1(x_1)$ and $F_2(x_2)$.

In this analysis, we assume Client 2 to be the *outlier*, which means the client's optimal parameter and model parameter distribution is far away from Client 1. In particular, $\mu_2 >> L_{smooth}^1$.

The global model starts with $x = 0$, and after enough local training updates, the model $x_1, x_2$ will converge to their personal optimum $x_1^*, x_2^*$. W.l.o.g, we let Client 1 with $F_1(x_1^*) = 0$, Client 2 with $F_2(x_2^*) = a > 0$. Let $x_1^* < x_2^*$ (relative position, which does not affect the analysis).

Based on the proposed aggregation $p_i \propto \exp\frac{F_i(x)}{\tau}$, we can derive the aggregated global model $\tilde{x}$ of FedEBA+ to be:

$$
\tilde{x} = p_1 x_1^* + p_2 x_2^* = \frac{x_1^* + e^a x_2^*}{e^a + 1} .
\tag{97}
$$

While for FedAvg, the aggregated global model $\overline{x}$ is:

$$
\overline{x} = \frac{x_1^* + x_2^*}{2} .
\tag{98}
$$

For FedEBA+, the test loss of Client 1 and Client 2 are $\tilde{L}_1 = L_1(\tilde{x}), \tilde{L}_2 = L_2(\tilde{x})$ respectively. The corresponding variance is $V_2^{EBA+} = \frac{1}{2}(\tilde{L}_1 - \tilde{L}_2)^2$.

For FedAvg, the test loss of Client 1 and Client 2 is $\overline{L}_1 = L_1(\overline{x}), \overline{L}_2 = L_2(\overline{x})$ respectively. The corresponding variance is $V_2^{AVG} = \frac{1}{2}(\overline{L}_1 - \overline{L}_2)^2$.

Since Client 2 is a outlier with $F_2(x_2^*) > 0$ and $x_1^* < x_2^*$, we can easily conclude $F_2(x)$ is monotonically decreasing on $(x_1^*, x_2^*)$, $F_1(x)$ is monotonically increasing on $(x_1^*, x_2^*)$. Besides, w.l.o.g, since $\nabla F_1(x) \leq L_{smooth} << \mu_2$, we can let $\mu = \frac{a}{x_2^* - x_1^*}$.

Thus, we promise $\frac{a}{x_2^* - x_1^*} > \nabla F_1(x_2^*)$. According to the property of calculus, we can easily check that $F_2(x) - F_1(x) > 0$ is monotonically decreasing on $(x_1^*, x_2^*)$.

Since

$$x_2^* - \tilde{x} = \frac{x_2^* - x_1^*}{e^a + 1} \le x_2^* - \overline{x} = \frac{x_2^* - x_1^*}{2}, \tag{99}$$

thus we have $(F_2(\tilde{x}) - F_1(\tilde{x}))^2 \le (F_2(\overline{x}) - F_1(\overline{x}))^2$ .

So far, we have prove $V_2^{EBA+} \le V_2^{AVG}$.

To extend the analysis to arbitrary $N$, we utilize the mathematical induction:

Assume $V_N^{EBA+} \le V_N^{AVG}$, we need to derive $V_{N+1}^{EBA+} \le V_{N+1}^{AVG}$.

Consider a similar scenario as we analyze with two clients. We assume Client N+1 to be an outlier, which means the client's optimal value and parameter distribution are far away from other clients. In particular, $\mu_{N+1>>L_{smooth}^{others}}$. W.l.o.g, let the optimal value $F(x_{N+1}^*)$ for Client N+1 be $a$, others to be zero.

Again, the global model starts with $x = 0$, and after enough local training updates, the models will converge to their personal optimum $x_1^*, x_2^*, \ldots, x_{N+1}^*$ and $x_{N+1}^* > x_{others}^*$.

By (96), we have:

$$V_{N+1}^{Avg} = \frac{N}{N+1} V_N^{AVG} + \frac{\sum_{i=1}^N (\overline{L}_i - \overline{L}_{N+1})^2}{(N+1)^2}, \tag{100}$$

where $\overline{L}_i$ is the test loss of client $i$ after average and

$$V_{N+1}^{EBA+} = \frac{N}{N+1} V_N^{EBA+} + \frac{\sum_{i=1}^N (\tilde{L}_i - \tilde{L}_{N+1})^2}{(N+1)^2}. \tag{101}$$

Since we know $V_N^{EBA+} \le V_N^{AVG}$, thus as long as we promise $\frac{\sum_{i=1}^N (\tilde{L}_i - \tilde{L}_{N+1})^2}{(N+1)^2} \le \frac{\sum_{i=1}^N (\overline{L}_i - \overline{L}_{N+1})^2}{(N+1)^2}$, we can finish the proof.

Consider an arbitrary client $i \in [1, N]$, since we already know $F_{N+1}(x_{N+1}^*) = a > F_i(x_i^*) = 0$, the expression for $\tilde{x}$ is

$$\tilde{x} = \sum_{i=1}^{N+1} p_i x_i^* = \frac{1}{N + e^a} \sum_{i=1}^N x_i^* + \frac{e^a}{N + e^a} x_{N+1}^*, \tag{102}$$

While for FedAvg,

$$\overline{x} = \sum_{i=1}^{N+1} \frac{1}{N+1} x_i^*. \tag{103}$$

Following the exact analysis on Client $i$ and Client $N + 1$, we can conclude that $F_{N+1}(x) - F_i(x) > 0$ is monotonically decreasing on $(x_i^*, x_{N+1}^*)$.

Since

$$x_{N+1}^* - \tilde{x} = \frac{N x_{N+1}^* - \sum_{i=1}^N x_i^*}{e^a + N} \le x_{N+1}^* - \overline{x} = \frac{N x_{N+1}^* - \sum_{i=1}^N x_i^*}{e^a + 1}, \tag{104}$$

thus we have $(F_{N+1}(\tilde{x}) - F_i(\tilde{x}))^2 \le (F_{N+1}(\overline{x}) - F_i(\overline{x}))^2 \ \forall i \in [1, \ldots, N]$.

Therefore, we promise $\frac{\sum_{i=1}^N (\tilde{L}_i - \tilde{L}_{N+1})^2}{(N+1)^2} \le \frac{\sum_{i=1}^N (\overline{L}_i - \overline{L}_{N+1})^2}{(N+1)^2}$.

So far, we have prove $V_{N+1}^{EBA+} \le V_{N+1}^{AVG}$.

According to the mathematical induction, we prove $V_N^{EBA+} \le V_N^{AVG}$ for arbitrary client number $N$ under smooth and strongly convex setting.

## 13 Pareto-optimality Analysis

In this section, we demonstrate the Proposition 5.5. In particular, we consider the degenerate setting of FedEBA+ where the parameter $\alpha = 0$. We first provide the following lemma that illustrates the correlation between Pareto optimality and monotonicity.

**Lemma 13.1** (Property 1 in (Sampat & Zavala, 2019).)**.** *The allocation strategy* $\varphi(p) = \arg\max_{p \in \mathcal{P}} h(p(f))$ *is Pareto optimal if* $h$ *is a strictly monotonically increasing function.*

In order for this paper to be self-contained, we restate the proof of Property 1 in (Sampat & Zavala, 2019) here:

**Proof Sketch:** We prove the result by contradiction. Consider that $p^* = \varphi(\mathcal{P})$ is not Pareto optimal; thus, there exists an alternative $p \in \mathcal{P}$ such that

$$\sum_i p_i f_i = \frac{\sum_i p_i \log p_i}{Z} \geq \sum_i p_i^* f_i = \frac{\sum_i p_i^* \log p_i^*}{Z}, \tag{105}$$

where $Z > 0$ is a constant. Since $h(p)$ is a strictly monotonically increasing function, we have $h(p) > h(p^*)$. This is a contradiction because $h^*$ maximizes $h(\cdot)$.

According to the above lemma, to show our algorithm achieves Pareto-optimal, we only need to show it is monotonically increasing.

Recall the objective of maximum entropy:

$$\mathbb{H}(p) = -\sum p(x) log(p(x)), \tag{106}$$

subject to certain constraints on the probabilities $p(x)$.

To show that the proposed aggregation strategy is monotonically increasing, we need to prove that if the constraints on the probabilities $p(x)$ are relaxed, then the maximum entropy of the aggregation probability increases.

One way to do this is to use the properties of the logarithm function. The logarithm function is strictly monotonically increasing. This means that for any positive real numbers a and b, if $a \leq b$, then $\log(a) \leq \log(b)$.

Now, suppose that we have two sets of constraints on the probabilities $p(x)$, and that the second set of constraints is a relaxation of the first set. This means that the second set of constraints allows for a larger set of probability distributions than the first set of constraints.

If we maximize the entropy subject to the first set of constraints, we get some probability distribution $p(x)$. If we then maximize the entropy subject to the second set of constraints, we get some probability distribution $q(x)$ such that $p(x) \leq q(x)$ for all x.

Using the properties of the logarithm function and the definition of the entropy, we have:

$$
\begin{aligned}
H(p(x)) &= -\sum (p(x) \log(p(x))) \\
&\leq -\sum (p(x) \log(q(x))) \\
&= -\sum ((p(x)/q(x))q(x) \log(q(x))) \\
&= H(q(x)) - \sum ((\frac{p(x)}{q(x)} q(x) \log(p(x)/q(x)))) \\
&\leq H(q(x)).
\end{aligned} \tag{107}
$$

This means that the entropy $H(q(x))$ is greater or equal to $H(p(x))$ when the second set of constraints is a relaxation of the first set of constraints. As the entropy increases when the constraints are relaxed, the maximum entropy-based aggregation strategy is monotonically increasing.

Up to this point, we proved that our proposed aggregation strategy is monotonically increasing. Combined with the Lemma 13.1, we can prove that equation (3) is Pareto optimal.

## 14 EXPERIMENT DETAILS

### 14.1 EXPERIMENTAL ENVIRONMENT

For all experiments, we use NVIDIA GeForce RTX 3090 GPUs. Each simulation trail with 2000 communication rounds and three random seeds.

**Federated datasets.** We tested the performance of FedEBA+ on five public datasets: MNIST, Fashion MNIST, CIFAR-10, CIFAR-100, and Tiny-ImageNet. We use two methods to split the real datasets into non-iid datasets: (1) following the setting of (Wang et al., 2021), where 100 clients participate in the federated system, and according to the labels, we divide all the data of MNIST, FashionMNIST, CIFAR-10, CIFAR-100 and Tiny-ImageNet into 200 shards separately, and each user randomly picks up 2 shards for local training. (2) we leverage Latent Dirichlet Allocation (LDA) to control the distribution drift with the Dirichlet parameter $\alpha = 0.1$. As for the model, we use an MLP model with 2 hidden layers on MNIST and Fashion-MNIST, and a CNN model with 2 convolution layers on CIFAR-10, ResNet-18 on CIFAR-100, and MobileNet-v2 on TinyImageNet.

**Baselines** We compared several advanced FL fairness algorithms with FedEBA+, including FedAvg (McMahan et al., 2017), q-FFL (Li et al., 2019a), FedFV (Wang et al., 2021), FedMGDA+ (Hu et al., 2022),PropFair (Zhang et al., 2022), TERM (Li et al., 2020a), and AFL (Mohri et al., 2019). We compare these algorithms on all four datasets.

**Hyper-parameters** As shown in Table 3, we tuned some hyper-parameters of baselines to ensure the performance in line with the previous studies and listed parameters used in FedEBA+. All experiments are running over 2000 rounds for a single local epoch ($K = 10$) with local batch size $B_{MNIST} = 200$, $B_{Fashion-MNIST} \in \{50, full\}$, $B_{CIFAR-10} \in \{50, full\}$ and $B_{CIFAR-100} = 64$. The learning rate remains the same for different methods, that is $\eta = 0.1$ on MNIST, Fasion-MNIST, CIFAR-10, $\eta = 0.05$ on Tiny-ImageNet and $\eta = 0.01$ on CIFAR-100 with decay rate $d = 0.999$.

Table 3: Hyperparameters of all experimental algorithms.

| Algorithm | Hyper Parameters |
|-----------|------------------|
| $q - FFL$ | $q \in \{0.001, 0.01, 0.1, 0.5, 10, 15\}$ |
| $PropFair$ | $M \in \{0.2, 5.0\}, \epsilon = 0.2$ |
| $AFL$ | $\lambda \in \{0.01, 0.1, 0.5, 0.7\}$ |
| $TERM$ | $T \in \{0.1, 0.5, 0.8\}$ |
| $FedMGDA+$ | $\epsilon \in \{0, 0.03, 0.08\}$ |
| $FedFV$ | $\alpha \in \{0.1, 0.2, 0.5\}, \tau \in \{0, 1, 10\}$ |
| $FedEBA+$ | $\tau \in \{0.5, 0.1\}, \alpha \in \{0.0, 0.5, 0.9\}$ |

## 15 ADDITIONAL EXPERIMENT RESULTS

**Fairness of FedEBA+** In this section, we provide additional experimental results to illustrate that FedEBA+ is superior over other baselines.

Figure 5 illustrates that, on the MNIST dataset, FedEBA+ demonstrates faster convergence, increased stability, and superior results in comparison to baselines. As for the CIFAR-10 dataset, its complexity causes some instability for all methods, however, FedEBA+ still concludes the training with the most favorable fairness results.

**Table 6 shows FedEBA+ outperforms other baselines on CIFAR-10 using MLP model.** The results in Table 6 demonstrate that 1) FedEBA+ consistently achieves a smaller variance of accuracy compared to other baselines, thus is fairer. 2) FedEBA+ significantly improves the performance of the worst 5% clients and 3) FedEBA+ performances steady in

Table 4: **Performance of algorithms on FashionMNIST and CIFAR-10.** We report the accuracy of global model, variance fairness, worst 5%, and best 5% accuracy. The data is divided into 100 clients, with 10 clients sampled in each round. All experiments are running over 2000 rounds for a single local epoch ($K = 10$) with local batch size $= 50$, and learning rate $\eta = 0.1$. The reported results are averaged over 5 runs with different random seeds. We highlight the best and the second-best results by using **bold font** and blue text.

| Algorithm | FashionMNIST (MLP) | | | | CIFAR-10 (CNN) | | | |
|---|---|---|---|---|---|---|---|---|
| | Global Acc. | Var. | Worst 5% | Best 5% | Global Acc. | Var. | Worst 5% | Best 5% |
| FedAvg | $86.49 \pm 0.09$ | $62.44\pm4.55$ | $71.27\pm1.14$ | $95.84\pm 0.35$ | $67.79\pm0.35$ | $103.83\pm10.46$ | $45.00\pm2.83$ | $85.13\pm0.82$ |
| $q-FFL\|_{q=0.001}$ | $87.05\pm 0.25$ | $66.67\pm 1.39$ | $72.11\pm 0.03$ | $95.09\pm 0.71$ | $68.53\pm 0.18$ | $97.42\pm 0.79$ | $48.40\pm 0.60$ | $84.70\pm 1.31$ |
| $q-FFL\|_{q=0.01}$ | $86.62\pm 0.03$ | $58.11\pm 3.21$ | $71.36\pm 1.98$ | $95.29\pm0.27$ | $68.85\pm 0.03$ | $95.17\pm 1.85$ | $48.20\pm0.80$ | $84.10\pm0.10$ |
| $q-FFL\|_{q=0.5}$ | $86.57\pm 0.19$ | $54.91\pm 2.82$ | $70.88\pm 0.98$ | $95.06\pm0.17$ | $68.76\pm 0.22$ | $97.81\pm 2.18$ | $48.33\pm0.84$ | $84.51\pm1.33$ |
| $q-FFL\|_{q=10.0}$ | $77.29\pm 0.20$ | $47.20\pm 0.82$ | $61.99\pm 0.48$ | $92.25\pm0.57$ | $40.78\pm 0.06$ | $85.93\pm 1.48$ | $22.70\pm0.10$ | $56.40\pm0.21$ |
| $q-FFL\|_{q=15.0}$ | $75.77\pm0.42$ | $46.58\pm0.75$ | $61.63\pm0.46$ | $89.60\pm0.42$ | $36.89\pm0.14$ | $79.65\pm5.17$ | $19.30\pm0.70$ | $51.30\pm0.09$ |
| $FedMGDA+\|_{\epsilon=0.0}$ | $86.01\pm0.31$ | $58.87\pm3.23$ | $71.49\pm0.16$ | $95.45\pm0.43$ | $67.16\pm0.33$ | $97.33\pm1.68$ | $46.00\pm0.79$ | $83.30\pm0.10$ |
| $FedMGDA+\|_{\epsilon=0.03}$ | $84.64\pm0.25$ | $57.89\pm6.21$ | $73.49\pm1.17$ | $93.22\pm0.20$ | $65.19\pm0.87$ | $89.78\pm5.87$ | $48.84\pm1.12$ | $81.94\pm0.67$ |
| $FedMGDA+\|_{\epsilon=0.08}$ | $84.90\pm0.34$ | $61.55\pm5.87$ | $\mathbf{73.64\pm0.85}$ | $92.78\pm0.12$ | $65.06\pm0.69$ | $93.70\pm14.10$ | $48.23\pm0.82$ | $82.01\pm0.09$ |
| $AFL\|_{\lambda=0.7}$ | $85.14\pm0.18$ | $57.39\pm6.13$ | $70.09\pm0.69$ | $95.94\pm0.09$ | $66.21\pm1.21$ | $79.75\pm1.25$ | $47.54\pm0.61$ | $82.08\pm0.77$ |
| $AFL\|_{\lambda=0.5}$ | $84.14\pm0.18$ | $90.76\pm3.33$ | $60.11\pm0.58$ | $96.00\pm0.09$ | $65.11\pm2.44$ | $86.19\pm9.46$ | $44.73\pm3.90$ | $82.10\pm0.62$ |
| $AFL\|_{\lambda=0.1}$ | $84.91\pm0.71$ | $69.39\pm6.50$ | $69.24\pm0.35$ | $95.39\pm0.72$ | $65.63\pm0.54$ | $88.74\pm3.39$ | $47.29\pm0.30$ | $82.33\pm0.41$ |
| $PropFair\|_{M=0.2,thres=0.2}$ | $85.51\pm0.28$ | $75.27\pm5.38$ | $63.60\pm0.53$ | $97.60\pm0.19$ | $65.79\pm0.53$ | $79.67\pm5.71$ | $49.88\pm0.93$ | $82.40\pm0.40$ |
| $PropFair\|_{M=5.0,thres=0.2}$ | $84.59\pm1.01$ | $85.31\pm8.62$ | $61.40\pm0.55$ | $96.40\pm0.29$ | $66.91\pm1.43$ | $78.90\pm6.48$ | $50.16\pm0.56$ | $85.40\pm0.34$ |
| $TERM\|_{T=0.1}$ | $84.31\pm0.38$ | $73.46\pm2.06$ | $68.23\pm0.10$ | $94.16\pm0.16$ | $65.41\pm0.37$ | $91.99\pm2.69$ | $49.08\pm0.66$ | $81.98\pm0.19$ |
| $TERM\|_{T=0.5}$ | $82.19\pm1.41$ | $87.82\pm2.62$ | $62.11\pm0.71$ | $93.25\pm0.39$ | $61.04\pm1.96$ | $96.78\pm7.67$ | $42.45\pm1.73$ | $80.06\pm0.62$ |
| $TERM\|_{T=0.8}$ | $81.33\pm1.21$ | $95.65\pm9.56$ | $56.41\pm0.56$ | $92.88\pm0.70$ | $59.21\pm1.45$ | $82.63\pm3.64$ | $41.33\pm0.68$ | $77.39\pm1.04$ |
| $FedFV\|_{\alpha=0.1,\tau_{fv}=1}$ | $86.51\pm0.28$ | $49.73\pm2.26$ | $71.33\pm1.16$ | $95.89\pm0.23$ | $68.94\pm0.27$ | $90.84\pm2.67$ | $50.53\pm4.33$ | $86.00\pm1.23$ |
| $FedFV\|_{\alpha=0.2,\tau_{fv}=0}$ | $86.42\pm0.38$ | $52.41\pm5.94$ | $71.22\pm1.35$ | $95.47\pm0.43$ | $68.89\pm0.15$ | $82.99\pm3.10$ | $50.08\pm0.40$ | $86.24\pm1.17$ |
| $FedFV\|_{\alpha=0.5,\tau_{fv}=10}$ | $86.88\pm0.26$ | $47.63\pm1.79$ | $71.49\pm0.39$ | $95.62\pm0.29$ | $69.42\pm0.60$ | $78.10\pm3.62$ | $52.80\pm0.34$ | $85.76\pm0.80$ |
| $FedFV\|_{\alpha=0.1,\tau_{fv}=10}$ | $86.98\pm0.45$ | $56.63\pm1.85$ | $66.40\pm0.57$ | $\mathbf{98.80\pm0.12}$ | $71.10\pm0.44$ | $86.50\pm7.36$ | $49.80\pm0.72$ | $\mathbf{88.42\pm0.25}$ |
| $FedEBA+\|_{\alpha=0,\tau=0.1}$ | $86.70\pm0.11$ | $50.27\pm5.60$ | $71.13\pm0.69$ | $95.47\pm0.27$ | $69.38\pm0.52$ | $89.49\pm10.95$ | $50.40\pm1.72$ | $86.07\pm0.90$ |
| $FedEBA+\|_{\alpha=0.5,\tau=0.1}$ | $87.21\pm0.06$ | $\mathbf{40.02\pm1.58}$ | $73.07\pm1.03$ | $95.81\pm0.14$ | $72.39\pm0.47$ | $70.60\pm3.19$ | $55.27\pm1.18$ | $86.27\pm1.16$ |
| $FedEBA+\|_{\alpha=0.9,\tau=0.1}$ | $\mathbf{87.50\pm0.19}$ | $43.41\pm4.34$ | $72.07\pm1.47$ | $95.91\pm0.19$ | $\mathbf{72.75\pm0.25}$ | $\mathbf{68.71\pm4.39}$ | $\mathbf{55.80\pm1.28}$ | $86.93\pm0.52$ |

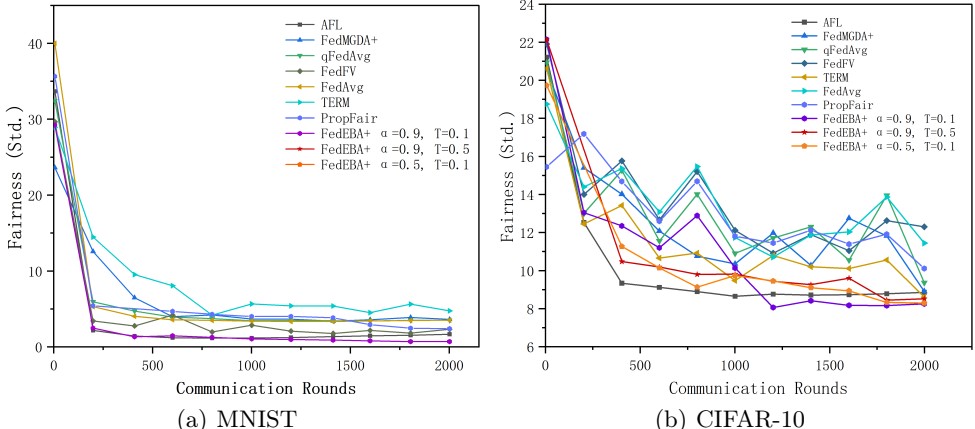

(a) MNIST    (b) CIFAR-10

Figure 5: **Performance of all the methods in terms of Fairness (Var.).**

terms of best 5% clients. A significant improvement in worst 5% is achieved with relatively no compromise in best 5 %, thus is fairer.

**Fairness in Different Non-i.i.d. Cases**   We adopt two kinds of data splitation strategies to change the degree of non-i.i.d., which are data devided by labels mentioned in the main text, and the data partitioning in deference to the Latent Dirichlet Allocation (LDA) with the Dirichlet parameter . Based on FedAvg, we have experimented with various data segmentation strategies for FedEBA+ to verify the performance of FedEBA+ for scenarios with different kinds of data held by clients.

**Global Accuracy of FedEBA+**   We run all methods on the CNN model, regarding the CIFAR-10 figure. Under different hyper-parameters, FedEBA+ can reach a stable

Table 5: **Ablation study for $\theta$ of FedEBA+.** This table shows our schedule of using the fair angle $\theta$ to control the gradient alignment times is effective, as it largely reduces the communication rounds with larger angles. In addition, compared with the results of baseline in Table 1, the results illustrate that our algorithm remains effective when we increase the fair angle. The additional cost is computed by Additional communication/total communications, the communication cost of communicating the MLP model is 7.8MB/round, the CNN model is 30.4MB/round.

| Algorithm | FashionMNIST (MLP) | | | CIFAR-10 (CNN) | | |
|---|---|---|---|---|---|---|
| | Global Acc. | Var. | Additional cost | Global Acc. | Var. | Additional cost |
| FedAvg | $86.49 \pm 0.09$ | $62.44 \pm 4.55$ | - | $67.79 \pm 0.35$ | $103.83 \pm 10.46$ | - |
| q-FFL | $87.05 \pm 0.25$ | $66.67 \pm 1.39$ | - | $68.53 \pm 0.18$ | $97.42 \pm 0.79$ | - |
| FedMGDA+ | $84.64 \pm 0.25$ | $57.89 \pm 6.21$ | - | $67.16 \pm 0.33$ | $97.33 \pm 1.68$ | - |
| AFL | $85.14 \pm 0.18$ | $57.39 \pm 6.13$ | - | $66.21 \pm 1.21$ | $79.75 \pm 1.25$ | - |
| PropFair | $85.51 \pm 0.28$ | $75.27 \pm 5.38$ | - | $65.79 \pm 0.53$ | $79.67 \pm 5.71$ | - |
| TERM | $84.31 \pm 0.38$ | $73.46 \pm 2.06$ | - | $65.41 \pm 0.37$ | $91.99 \pm 2.69$ | - |
| FedFV | $86.98 \pm 0.45$ | $56.63 \pm 1.85$ | - | $71.10 \pm 0.44$ | $86.50 \pm 7.36$ | - |
| FedEBA+ | | | | | | |
| $\theta = 0°$ | $87.50 \pm 0.19$ | $43.41 \pm 4.34$ | 50.0% | $72.75 \pm 0.25$ | $68.71 \pm 4.39$ | 50.0% |
| $\theta = 15°$ | $87.14 \pm 0.12$ | $43.95 \pm 5.12$ | 48.6% | $71.92 \pm 0.33$ | $75.95 \pm 4.72$ | 26.2% |
| $\theta = 30°$ | $86.96 \pm 0.06$ | $46.82 \pm 1.21$ | 37.7% | $70.91 \pm 0.46$ | $70.97 \pm 4.88$ | 12.7% |
| $\theta = 45°$ | $86.94 \pm 0.26$ | $46.63 \pm 4.38$ | 4.2% | $70.24 \pm 0.08$ | $79.51 \pm 2.88$ | 0.2% |
| $\theta = 90°$ | $86.78 \pm 0.47$ | $48.91 \pm 3.62$ | 0% | $70.14 \pm 0.27$ | $79.43 \pm 1.45$ | 0% |

Table 6: Performance of algorithms on CIFAR-10 using MLP. We report the global model's accuracy, fairness of accuracy, worst 5% and best 5% accuracy. All experiments are running over 2000 rounds for a single local epoch ($K = 10$) with local batch size $= 50$, and learning rate $\eta = 0.1$. The reported results are averaged over 5 runs with different random seeds. We highlight the best and the second-best results by using bold font and blue text.

| Method | Global Acc. | Std. | Worst 5% | Best 5% |
|---|---|---|---|---|
| FedAvg | $46.85 \pm 0.65$ | $12.57 \pm 1.50$ | $19.84 \pm 6.55$ | $69.28 \pm 1.17$ |
| $q-FFL\|_{q=0.1}$ | $47.02 \pm 0.89$ | $13.16 \pm 1.84$ | $18.72 \pm 6.94$ | $70.16 \pm 2.06$ |
| $q-FFL\|_{q=0.2}$ | $46.91 \pm 0.90$ | $13.09 \pm 1.84$ | $18.88 \pm 7.00$ | $70.16 \pm 2.10$ |
| $q-FFL\|_{q=1.0}$ | $46.79 \pm 0.73$ | $11.72 \pm 1.00$ | $22.80 \pm 3.39$ | $68.00 \pm 1.60$ |
| $q-FFL\|_{q=2.0}$ | $46.36 \pm 0.38$ | $10.85 \pm 0.76$ | $24.64 \pm 2.17$ | $66.80 \pm 2.02$ |
| $q-FFL\|_{q=5.0}$ | $45.25 \pm 0.42$ | $9.59 \pm 0.36$ | $26.56 \pm 1.03$ | $63.60 \pm 1.13$ |
| $Ditto\|_{\lambda=0.0}$ | $52.78 \pm 1.23$ | $10.17 \pm 0.24$ | $31.80 \pm 2.27$ | $71.47 \pm 1.20$ |
| $Ditto\|_{\lambda=0.5}$ | $53.77 \pm 1.02$ | $8.89 \pm 0.32$ | $36.27 \pm 2.81$ | $71.27 \pm 0.52$ |
| $AFL\|_{\lambda=0.01}$ | $52.69 \pm 0.19$ | $10.57 \pm 0.37$ | $34.00 \pm 1.30$ | $71.33 \pm 0.57$ |
| $AFL\|_{\lambda=0.1}$ | $52.68 \pm 0.46$ | $10.64 \pm 0.14$ | $33.27 \pm 1.75$ | $\mathbf{71.53 \pm 0.52}$ |
| $TERM\|_{T=1.0}$ | $45.14 \pm 2.25$ | $9.12 \pm 0.35$ | $27.07 \pm 3.49$ | $62.73 \pm 1.37$ |
| $FedMGDA+\|_{\epsilon=0.01}$ | $45.65 \pm 0.21$ | $10.94 \pm 0.87$ | $25.12 \pm 2.34$ | $67.44 \pm 1.20$ |
| $FedMGDA+\|_{\epsilon=0.05}$ | $45.58 \pm 0.21$ | $10.98 \pm 0.81$ | $25.12 \pm 1.87$ | $67.76 \pm 2.27$ |
| $FedMGDA+\|_{\epsilon=0.1}$ | $45.52 \pm 0.17$ | $11.32 \pm 0.86$ | $24.32 \pm 2.24$ | $68.48 \pm 2.68$ |
| $FedMGDA+\|_{\epsilon=0.5}$ | $45.34 \pm 0.21$ | $11.63 \pm 0.69$ | $24.00 \pm 1.93$ | $68.64 \pm 3.11$ |
| $FedMGDA+\|_{\epsilon=1.0}$ | $45.34 \pm 0.22$ | $11.64 \pm 0.66$ | $24.00 \pm 1.93$ | $68.64 \pm 3.11$ |
| $FedFV\|_{\alpha=0.1,\tau_{fv}=1}$ | $54.28 \pm 0.37$ | $9.25 \pm 0.42$ | $35.25 \pm 1.01$ | $71.13 \pm 1.37$ |
| $FedEBA\|_{\alpha=0.9,\tau=0.1}$ | $53.94 \pm 0.13$ | $9.25 \pm 0.95$ | $35.87 \pm 1.80$ | $69.93 \pm 1.00$ |
| $FedEBA+\|_{\alpha=0.5,\tau=0.1}$ | $53.14 \pm 0.05$ | $8.48 \pm 0.32$ | $36.03 \pm 2.08$ | $69.20 \pm 0.75$ |
| $FedEBA+\|_{\alpha=0.9,\tau=0.1}$ | $\mathbf{54.43 \pm 0.24}$ | $\mathbf{8.10 \pm 0.17}$ | $\mathbf{40.07 \pm 0.57}$ | $69.80 \pm 0.16$ |

high performance of worst 5% while guaranteeing best 5%, as shown in Figure 6. As for FashionMNIST using MLP model, the worst 5% and best 5% performance of FedEBA+ are similar to that of CIFAR-10. We can see that FedEBA+ has a more significant lead in worst 5% with almost no loss in best 5%, as shown in Figure 7.

**Incorporating noisy label scenario** The local noisy label follows the symmetric flipping approach introduced in Jiang et al. (2022); Fang & Ye (2022), with a noise ratio of $\epsilon$ set to

Table 7: Performance of algorithms+momentum on Fashion-MNIST to show that FedEBA+ is orthogonal to advance optimization methods like momentum (Karimireddy et al., 2020a), allowing seamless integration. All experiments are running over 2000 rounds on the MLP model for a single local epoch ($K = 10$) with local batch size = 50, global momentum = 0.9 and learning rate $\eta = 0.1$. The reported results are averaged over 5 runs with different random seeds. We highlight the best and the second-best results by using bold font and blue text.

| Method | Global Acc. | Var. | Worst 5% | Best 5% |
|---|---|---|---|---|
| FedAvg | $86.68\pm 0.37$ | $66.15\pm 3.23$ | $72.18\pm 0.22$ | $96.04\pm\pm 0.35$ |
| $AFL\|_{\lambda=0.05}$ | $79.68\pm 0.91$ | $55.00\pm 3.34$ | $66.67\pm 0.12$ | $94.00\pm 0.08$ |
| $AFL\|_{\lambda=0.7}$ | $85.41\pm 0.30$ | $63.42\pm\pm 1.55$ | $\mathbf{73.83\pm 0.37}$ | $96.46\pm 0.12$ |
| $q-FFL\|_{q=0.01}$ | $86.82\pm 0.20$ | $64.11\pm 2.17$ | $71.08\pm 0.16$ | $96.29\pm 0.08$ |
| $q-FFL\|_{q=15}$ | $79.59\pm 0.48$ | $62.26\pm 2.88$ | $66.33\pm 1.14$ | $90.07\pm 0.98$ |
| $FedMGDA+\|_{\epsilon=0.0}$ | $82.69\pm 0.52$ | $65.26\pm 3.81$ | $69.63\pm 1.20$ | $92.67\pm 0.54$ |
| $PropFair\|_{M=5,thres=0.2}$ | $85.67\pm 0.19$ | $73.44\pm 2.44$ | $64.59\pm 0.42$ | $97.47\pm 0.11$ |
| $FedProx\|_{\mu=0.1}$ | $86.76\pm 0.26$ | $60.69\pm 3.07$ | $72.67\pm 0.29$ | $95.96\pm 0.14$ |
| $TERM\|_{T=0.1}$ | $84.58\pm 0.28$ | $76.44\pm 2.50$ | $69.52\pm 0.36$ | $94.04\pm 0.50$ |
| $FedFV\|_{\alpha=0.1,\tau=10}$ | $87.46\pm 0.18$ | $58.35\pm 1.89$ | $67.71\pm 0.56$ | $\mathbf{97.79\pm 0.18}$ |
| $FedEBA+\|_{\alpha=0.9,T=0.1}$ | $\mathbf{87.67\pm 0.28}$ | $\mathbf{46.67\pm 1.09}$ | $71.90\pm 0.70$ | $96.26\pm 0.03$ |

Table 8: Performance of algorithms+VARP on Fashion-MNIST to show that FedEBA+ is orthogonal to advance optimization methods like VARP (Jhunjhunwala et al., 2022), allowing seamless integration. All experiments are running over 2000 rounds on the MLP model for a single local epoch ($K = 10$) with local batch size = 50, global learning rate = 1.0 and client learning rate = 0.1. The reported results are averaged over 5 runs with different random seeds. We highlight the best and the second-best results by using bold font and blue text.

| Method | Global Acc. | Var. | Worst 5% | Best 5% |
|---|---|---|---|---|
| FedAvg(FedVARP) | $87.12\pm 0.08$ | $59.96\pm 2.48$ | $72.45\pm 0.26$ | $96.09\pm\pm 0.27$ |
| $q-FFL\|_{q=0.01}$ | $86.73\pm 0.31$ | $62.89\pm 2.67$ | $73.55\pm 0.11$ | $95.54\pm 0.14$ |
| $q-FFL\|_{q=15}$ | $78.98\pm 0.63$ | $58.28\pm 1.95$ | $67.12\pm 0.97$ | $88.42\pm 0.67$ |
| $FedFV\|_{\alpha=0.1,\tau=10}$ | $87.28\pm 0.10$ | $57.90\pm 1.77$ | $67.41\pm 0.30$ | $\mathbf{97.66\pm 0.06}$ |
| $FedEBA+\|_{\alpha=0.9,T=0.1}$ | $\mathbf{87.45\pm 0.18}$ | $\mathbf{49.91\pm 2.38}$ | $71.44\pm 0.64$ | $95.94\pm 0.09$ |

Table 9: **Ablation study for Dirichlet parameter $\alpha$.** Performance comparison between FedAvg and FedEBA+ on CIFAR-100 using ResNet18 (devided by Dirichlet Distribution with $\alpha \in \{0.1, 0.5, 1.0\}$). We report the global model's accuracy, fairness of accuracy, worst 5% and best 5% accuracy. All experiments are running over 2000 rounds for a single local epoch ($K = 10$) with local batch size = 64, and learning rate $\eta = 0.01$. The reported results are averaged over 5 runs with different random seeds.

| Algorithm | Global Acc. | | | Var. | | | Worst 5% | | | Best 5% | | |
|---|---|---|---|---|---|---|---|---|---|---|---|---|
| | $\alpha=0.1$ | $\alpha=0.5$ | $\alpha=1.0$ | $\alpha=0.1$ | $\alpha=0.5$ | $\alpha=1.0$ | $\alpha=0.1$ | $\alpha=0.5$ | $\alpha=1.0$ | $\alpha=0.1$ | $\alpha=0.5$ | $\alpha=1.0$ |
| FedAvg | $30.94\pm0.04$ | $54.69\pm0.25$ | $64.91\pm0.02$ | $17.24\pm0.08$ | $7.92\pm0.03$ | $5.18\pm0.06$ | $0.20\pm0.00$ | $38.79\pm0.24$ | $54.36\pm0.11$ | $65.90\pm1.48$ | $70.10\pm0.25$ | $75.43\pm0.39$ |
| FedEBA+ | $33.39\pm0.22$ | $58.55\pm0.41$ | $65.98\pm0.04$ | $16.92\pm0.04$ | $7.71\pm0.08$ | $4.44\pm0.10$ | $0.95\pm0.15$ | $41.63\pm0.16$ | $58.20\pm0.17$ | $68.51\pm0.21$ | $74.03\pm0.07$ | $74.96\pm0.16$ |

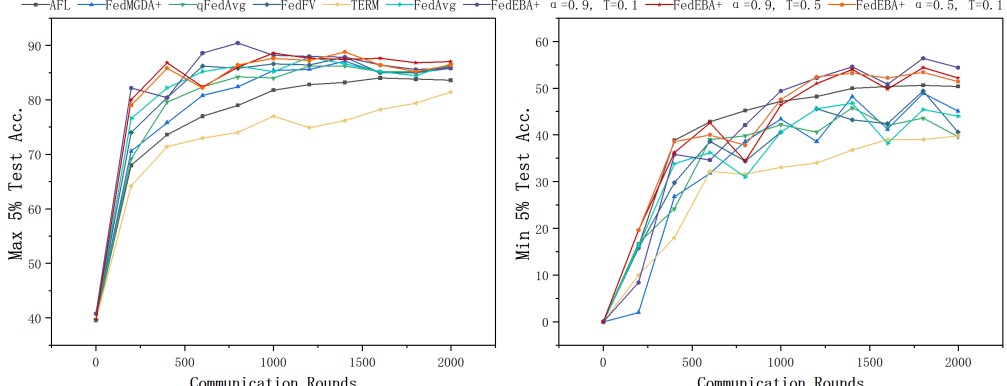

Figure 6: **The maximum and minimum 5% performance of all baselines and FedEBA+ on CIFAR-10.**

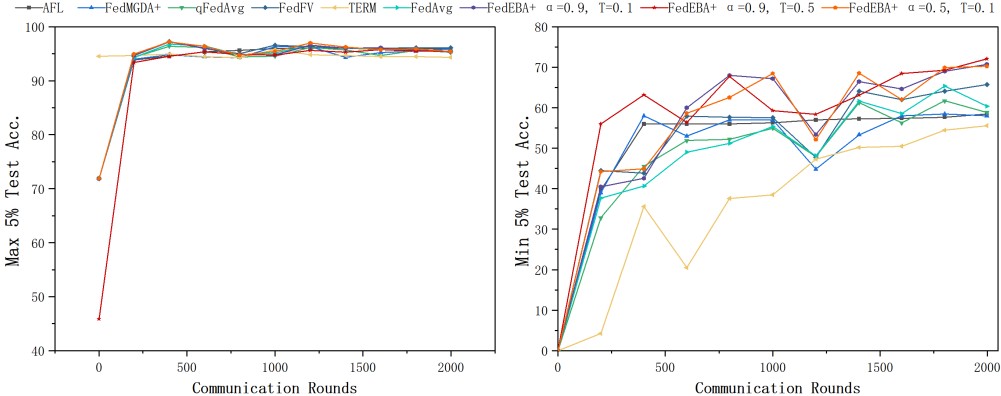

Figure 7: **The maximum and minimum 5% performance of all baselines and FedEBA+ on FashionMNSIT.**

Table 10: **Performance of algorithms on local noisy label scenario.** We test the effectiveness of FedEBA+ when incorporating the local noisy labels on both FashionMNIST and CIFAR-10 datasets, with noise ratio $\epsilon = 0.5$.

| Algorithm | FashionMNIST (MLP) | | | | CIFAR-10 (CNN) | | | |
|---|---|---|---|---|---|---|---|---|
| | Global Acc. ↑ | Var. ↓ | Worst 5% ↑ | Best 5% ↑ | Global Acc.↑ | Var. ↓ | Worst 5% ↑ | Best 5% ↑ |
| FedAvg | $66.10 \pm 1.61$ | $14.73 \pm 0.64$ | $34.51 \pm 1.61$ | $91.75 \pm 1.02$ | $38.03 \pm 0.69$ | $14.53 \pm 0.28$ | $13.21 \pm 0.39$ | $72.10 \pm 0.30$ |
| $q - FFL$ | $65.79 \pm 0.05$ | $16.52 \pm 0.24$ | $35.42 \pm 1.00$ | $90.84 \pm 0.74$ | $38.84 \pm 0.11$ | $14.40 \pm 0.20$ | $15.01 \pm 0.28$ | $71.05 \pm 0.53$ |
| $FedEBA+$ | $68.22 \pm 1.45$ | $13.94 \pm 0.27$ | $36.16 \pm 1.71$ | $91.49 \pm 1.60$ | $42.23 \pm 0.35$ | $9.25 \pm 0.24$ | $24.49 \pm 0.46$ | $65.63 \pm 0.83$ |
| FedAvg +LSR | $72.63 \pm 0.10$ | $21.12 \pm 1.00$ | $24.62 \pm 0.46$ | $95.21 \pm 0.21$ | $46.64 \pm 0.94$ | $16.08 \pm 1.03$ | $20.47 \pm 1.33$ | $75.02 \pm 0.18$ |
| $q - FFL$+LSR | $67.69 \pm 0.03$ | $22.26 \pm 0.16$ | $24.95 \pm 0.17$ | $95.50 \pm 0.09$ | $38.03 \pm 0.30$ | $14.02 \pm 0.21$ | $15.84 \pm 0.72$ | $69.99 \pm 0.21$ |
| $FedEBA+$ + LSR | $73.03 \pm 0.10$ | $20.88 \pm 0.88$ | $1.12 \pm 0.05$ | $13.89 \pm 0.72$ | $50.96 \pm 0.49$ | $15.83 \pm 0.34$ | $18.02 \pm 0.02$ | $74.50 \pm 0.10$ |

0.5. All the other settings like the learning rate keep the same. Specifically, we employ the MLP model for Fashion-MNIST and the CNN model for CIFAR-10.

The results of Table 10 reveal that (1) FedEBA+ maintains its superiority in accuracy and fairness even when there are local noisy labels; (2) FedEBA+ can be integrated with established approaches for addressing local noisy labels, consistently outperforming other algorithms combined with existing methods in terms of both fairness and accuracy.

**Ablation study.** For the annealing schedule of $\tau$ in Section 4.1, Figure 8 shows that the annealing schedule has advantages in reducing the variance compared with constant $\tau$. Besides, the global accuracy is robust to the annealing strategy, and the annealing strategy is robust to the initial temperature $T_0$.

We provide the ablation studies for $\theta$, the tolerable fair angle.

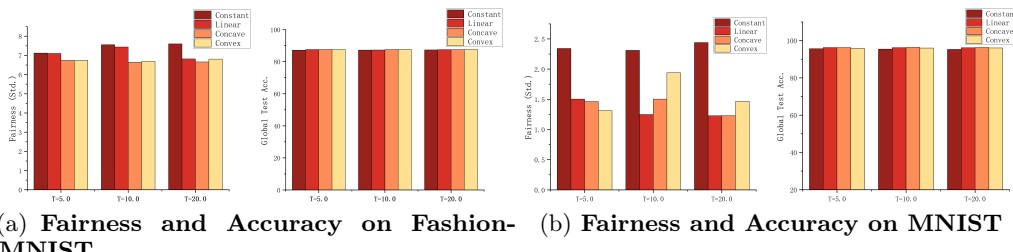

(a) **Fairness and Accuracy on Fashion-MNIST**
(b) **Fairness and Accuracy on MNIST**

Figure 8: Ablation study for Annealing schedule $\tau$

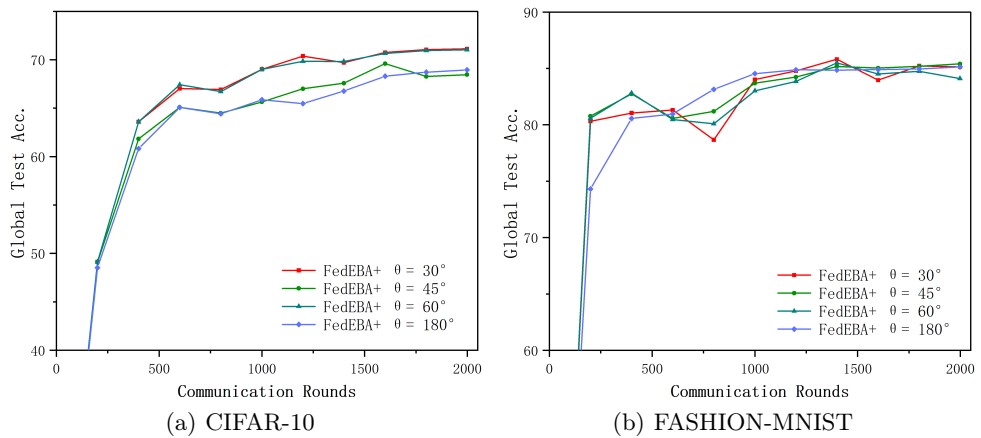

(a) CIFAR-10
(b) FASHION-MNIST

Figure 9: **Performance of $FedEBA+$ under different $\theta$ in terms of global accuracy.**

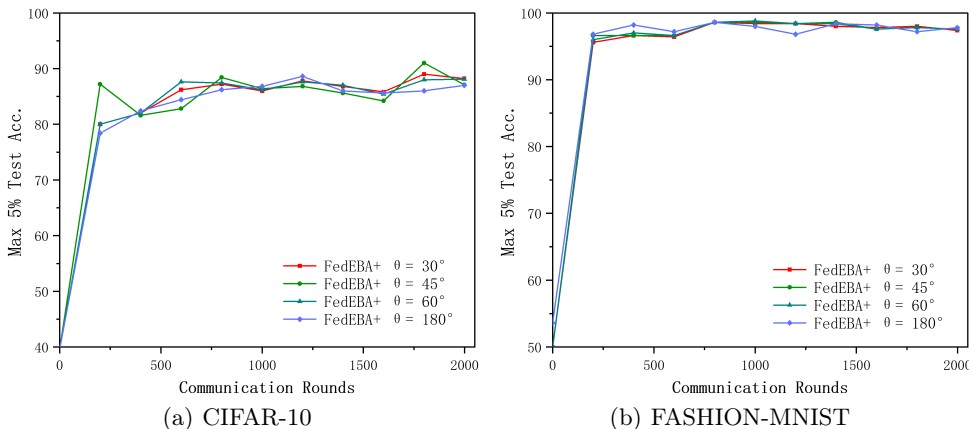

(a) CIFAR-10
(b) FASHION-MNIST

Figure 10: **Performance of $FedEBA+$ under different $\theta$ in terms of Max $5\%$ test accuracy.**

The results in Figure 9 10 11 show our algorithm is relatively robust to the tolerable fair angle $\theta$, though the choice of $\theta = 45$ may slow the performance slightly on global accuracy and min $5\%$ accuracy over CIFAR-10.

## 16  DISCUSSION OF FAIRNESS METRICS

In this section, we summarize the commonly used definitions of fairness metrics and comment on their advantages and disadvantages.

Euclidean Distance and person correlation coefficient are usually used for contribution fairness, and risk difference and Jain's fairness Index are usually used for group fairness, which is a different target from performance fairness in this paper. In particular, cosine

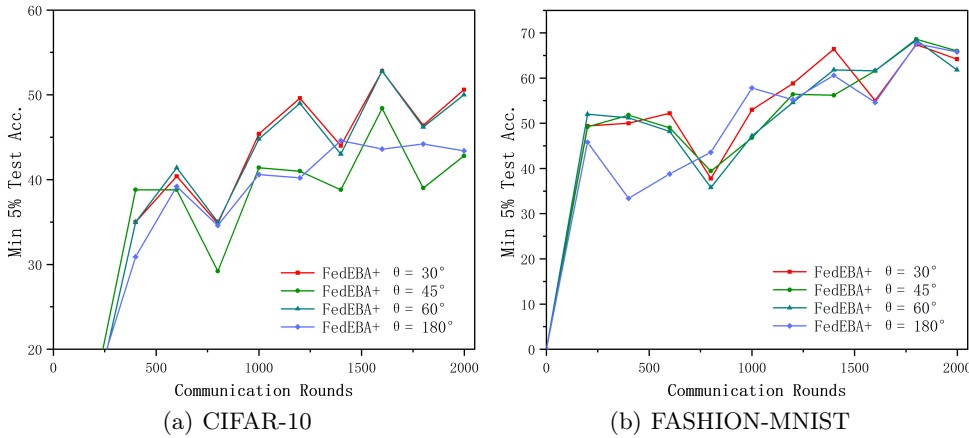

(a) CIFAR-10      (b) FASHION-MNIST

Figure 11: **Performance of $FedEBA+$ under different $\theta$ in terms of Min $5\%$ test accuracy.**

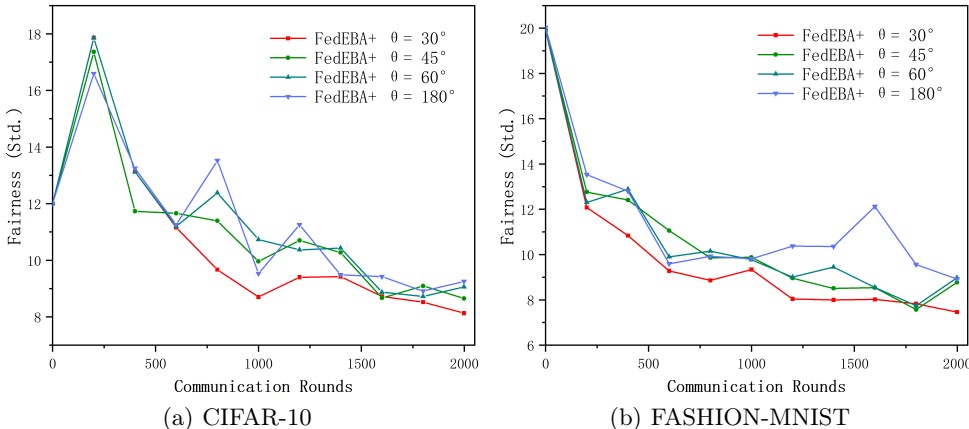

(a) CIFAR-10      (b) FASHION-MNIST

Figure 12: **Performance of $FedEBA+$ under different $\theta$ in terms of Fairness (Std).**

Table 11: **Ablation study for FedEBA+ on four datasets.** We test the effectiveness of FedEBA+ when decomposing each proposed step, i.e., entropy-based aggregation and alignment update, on different datasets. FedEBA differs from FedAvg only in the aggregation method, and FedEBA+ incorporates the alignment into FedEBA. FedAvg serves as the backbone, FedAvg+① is employed to demonstrate the individual effectiveness of our proposed aggregation step, FedAvg+②is utilized to showcase the individual effectiveness of our proposed alignment step, and FedAvg + ① + ② is used to show the effectiveness of our proposed algorithm, FedEBA+.

| Algorithm | CIFAR-10 (CNN) | | | | FashionMNIST (MLP) | | | |
|---|---|---|---|---|---|---|---|---|
| | Global Acc. ↑ | Var. ↓ | Worst 5% ↑ | Best 5% ↑ | Global Acc.↑ | Var. ↓ | Worst 5% ↑ | Best 5% ↑ |
| FedAvg | 67.79±0.35 | 103.83±10.46 | 45.00±2.83 | 85.13±0.82 | 86.49±0.09 | 62.44±4.55 | 71.27±1.14 | 95.84±0.35 |
| FedAvg+① | 69.38±0.52 | 89.49±10.95 | 50.40±1.72 | 86.07±0.90 | 86.70±0.11 | 50.27±5.60 | 71.13±0.69 | 95.47±0.27 |
| FedAvg+② | 72.04±0.51 | 75.73±4.27 | 53.45±1.25 | 87.33±0.23 | 87.42± 0.09 | 60.08±7.30 | 69.12±1.23 | 97.8±0.19 |
| FedAvg+①+② | 72.75±0.25 | 68.71±4.39 | 55.80±1.28 | 86.93±0.52 | 87.50±0.19 | 43.41±4.34 | 72.07±1.47 | 95.91±0.19 |

| Algorithm | CIFAR-100 (Resnet-18) | | | | Tiny-ImageNet (MobileNet-2) | | | |
|---|---|---|---|---|---|---|---|---|
| | Global Acc. ↑ | Var. ↓ | Worst 5% ↑ | Best 5% ↑ | Global Acc.↑ | Var. ↓ | Worst 5% ↑ | Best 5% ↑ |
| FedAvg | 30.94±0.04 | 17.24±0.08 | 0.20±0.00 | 65.90±1.48 | 61.99±0.17 | 19.62±1.12 | 53.60±0.06 | 71.18±0.13 |
| FedAvg+① | 32.38±0.13 | 17.09±0.06 | 0.75±0.22 | 66.40±0.47 | 63.34±0.25 | 15.29±1.36 | 54.17±0.04 | 70.98±0.10 |
| FedAvg+①+② | 33.39±0.22 | 16.92±0.04 | 0.95±0.15 | 68.51±0.21 | 64.05±0.09 | 14.91±1.85 | 54.32±0.09 | 71.27±0.04 |

similarity and entropy have similar roles to variance, used to measure the performance distribution among clients, the more uniform of the distribution, the smaller the variance, more similar to vector 1, the larger entropy of the normalized performance. Thus, we only need one of them for performance fairness, thus we use variance that is most widely used in related works as metric.

The detailed discussion of each metric is shown below:

Table 12: **Performance of FedEBA+ with different $\tau$ and $\alpha$ choices.** The performance of different hyper-parameter choices of FedEBA+ shows better performance than baselines.

| Algorithm | FashionMNIST (MLP) | | CIFAR-10 (CNN) | |
|---|---|---|---|---|
| | Global Acc. | Var. | Global Acc. | Var. |
| FedAvg | $86.49 \pm 0.09$ | $62.44 \pm 4.55$ | $67.79 \pm 0.35$ | $103.83 \pm 10.46$ |
| q-FFL$|_{q=0.001}$ | $87.05 \pm 0.25$ | $66.67 \pm 1.39$ | $68.53 \pm 0.18$ | $97.42 \pm 0.79$ |
| q-FFL$|_{q=0.5}$ | $86.57 \pm 0.19$ | $54.91 \pm 2.82$ | $68.76 \pm 0.22$ | $97.81 \pm 2.18$ |
| q-FFL$|_{q=10.0}$ | $77.29 \pm 0.20$ | $47.20 \pm 0.82$ | $40.78 \pm 0.06$ | $85.93 \pm 1.48$ |
| PropFair$|_{M=0.2,thres=0.2}$ | $85.51 \pm 0.28$ | $75.27 \pm 5.38$ | $65.79 \pm 0.53$ | $79.67 \pm 5.71$ |
| PropFair$|_{M=5.0,thres=0.2}$ | $84.59 \pm 1.01$ | $85.31 \pm 8.62$ | $66.91 \pm 1.43$ | $78.90 \pm 6.48$ |
| FedFV$|_{\alpha=0.1,\tau fv=10}$ | $86.98 \pm 0.45$ | $56.63 \pm 1.85$ | $71.10 \pm 0.44$ | $86.50 \pm 7.36$ |
| FedFV$|_{\alpha=0.2,\tau fv=0}$ | $86.42 \pm 0.38$ | $52.41 \pm 5.94$ | $68.89 \pm 0.15$ | $82.99 \pm 3.10$ |
| FedEBA+$|_{\alpha=0.1,\tau=0.1}$ | $86.98 \pm 0.10$ | $53.26 \pm 1.00$ | $71.82 \pm 0.54$ | $83.18 \pm 3.44$ |
| FedEBA+$|_{\alpha=0.3,\tau=0.1}$ | $87.01 \pm 0.06$ | $51.878 \pm 1.56$ | $71.79 \pm 0.35$ | $77.74 \pm 6.54$ |
| FedEBA+$|_{\alpha=0.7,\tau=0.1}$ | $87.23 \pm 0.07$ | $40.456 \pm 1.45$ | $72.36 \pm 0.15$ | $77.61 \pm 6.31$ |
| FedEBA+$|_{\alpha=0.9,\tau=0.05}$ | $87.42 \pm 0.10$ | $50.46 \pm 2.37$ | $72.19 \pm 0.16$ | $71.79 \pm 6.37$ |
| FedEBA+$|_{\alpha=0.9,\tau=0.5}$ | $87.26 \pm 0.06$ | $52.65 \pm 4.03$ | $71.89 \pm 0.39$ | $75.29 \pm 9.01$ |
| FedEBA+$|_{\alpha=0.9,\tau=1.0}$ | $87.14 \pm 0.07$ | $52.71 \pm 1.45$ | $72.30 \pm 0.26$ | $73.79 \pm 9.11$ |
| FedEBA+$|_{\alpha=0.9,\tau=5.0}$ | $87.10 \pm 0.14$ | $55.52 \pm 2.15$ | $72.43 \pm 0.11$ | $82.08 \pm 8.31$ |

Table 13: Comparison of Algorithms with metric *coefficient of variation* $(C_V)$ The $C_V$ improvement shows the improvement of algorithms over FedAvg. The result is calculated by global accuracy and variance of Table 1.

| Algorithm | FashionMNIST | | CIFAR-10 | |
|---|---|---|---|---|
| | $C_v = \frac{std}{acc}$ | $C_v$ improvement | $C_v = \frac{std}{acc}$ | $C_v$ improvement |
| FedAvg | 0.09136199 | 0% | 0.150312741 | 0% |
| q-FFL | 0.112432356 | -23% | 0.144026806 | 4.2% |
| FedMGDA+ | 0.089893051 | 1.3% | 0.146896915 | 2.4% |
| AFL | 0.088978374 | 2.6% | 0.134878199 | 10.1% |
| PropFair | 0.101459812 | -11.3% | 0.135671155 | 10.9% |
| TERM | 0.101659126 | -10.1% | 0.146631123 | 2.7% |
| FedFV | 0.086517483 | 4.8% | 0.130809249 | 13.3% |
| FedEBA+ | 0.072539115 | 21.8% | 0.1139402 | 27.8% |

- **Variance**, applied in accuracy parity and performance fairness scenarios, is valued for its simplicity and straightforward implementation, focusing on a common performance metric. However, it has a limitation as it only measures relative fairness, making it sensitive to outliers (Zafar et al., 2017; Li et al., 2019a; 2021; Hu et al., 2022; Shi et al., 2021).

- **Cosine similarity**, sharing applications with variance, is known for its similarity to variance and the ease with which it captures linear relationships (Li et al., 2019a). Nevertheless, it falls short when it comes to capturing magnitude differences and is sensitive to zero vectors (Selbst et al., 2019; Hardt et al., 2016).

- Also utilized in scenarios akin to variance, **entropy** offers simplicity but has dependencies on normalization and sensitivity to the number of clients involved in the computation, making it less robust in certain situations (Li et al., 2019a; Selbst et al., 2019; Hardt et al., 2016).

Table 14: **Performance of Algorithms with Various Metrics.** We provide the results under cosine similarity and entropy metrics, as used in (Li et al., 2019a), the geometric angle corresponds to cosine similarity metric, and KL divergence between the normalized accuracy vector **a** and uniform distribution **u** that can be directly translated to the entropy of **a**. We test the algorithms on the FashionMNIST dataset, with fine-tuned hyperparameters.

| Algorithm | Global Acc. | Var. | Angle ($\circ$) | KL ($a\|\|u$) |
|---|---|---|---|---|
| FedAvg | $86.49 \pm 0.09$ | $62.44\pm4.55$ | $8.70\pm1.71$ | $0.0145\pm0.002$ |
| q-FFL | $87.05\pm 0.25$ | $66.67\pm 1.39$ | $7.97\pm0.06$ | $0.0127\pm0.001$ |
| FedMGDA+ | $84.64\pm0.25$ | $57.89\pm6.21$ | $8.21\pm1.71$ | $0.0132\pm0.0004$ |
| AFL | $85.14\pm0.18$ | $57.39\pm6.13$ | $7.28\pm0.45$ | $0.0124\pm0.0002$ |
| PropFair | $85.51\pm0.28$ | $75.27\pm5.38$ | $8.61\pm2.29$ | $0.0139\pm0.002$ |
| TERM | $84.31\pm0.38$ | $73.46\pm2.06$ | $9.04\pm0.45$ | $0.0137\pm0.004$ |
| FedFV | $86.98\pm0.45$ | $56.63\pm1.85$ | $8.01\pm1.14$ | $0.0111\pm0.0002$ |
| FedEBA+ | $87.50\pm0.19$ | $43.41\pm4.34$ | $6.46\pm0.65$ | $0.0063\pm0.0009$ |

- Applied in contribution fairness, **Euclidean distance** provides a straightforward interpretation and is sensitive to magnitude differences. However, it lacks consideration for the direction of the differences, limiting its overall effectiveness.

- In contribution fairness scenarios, the **Pearson correlation coefficient** is appreciated for its scale invariance and ability to capture linear relationships (Jia et al., 2019). Yet, it may be sensitive to outliers and may not accurately capture magnitude differences, assuming a linear relationship between the data variables (Wang et al., 2019).

- Commonly used in group fairness contexts, **risk difference** is sensitive to group disparities and offers interpretability (Du et al., 2021). However, it lacks normalization, which can impact its effectiveness in certain scenarios (Dwork et al., 2012).

- **Jain's Fairness Index** finds application in various fairness aspects, including group fairness, selection fairness, performance fairness, and contribution fairness. It boasts normalization across groups and flexibility in handling various metrics. Nevertheless, it is sensitive to metric choice and introduces complexity in interpretability (Chiu, 1984; Liu et al., 2022).

