# OpenReview forum: "FedEBA+: Towards Fair and Effective Federated Learning via Entropy-based Model"
_ICLR.cc/2024/Conference — Submitted to ICLR 2024_

### Official Review · Reviewer_4hVe · 2023-10-30

**Soundness:** 3 good
**Presentation:** 4 excellent
**Contribution:** 3 good
**Rating:** 6
**Confidence:** 3

**Summary:**

This paper aims to formulate a concurrently fair and efficient protocol in federated learning. Even though fairness has been a widely studied topic in federated learning and many earlier works have discussed that, it remains a major challenge to design an FL algorithm that both improves the global (final) model performance and guarantees fairness. The authors tackled the challenge by modifying the aggregation rule via a method inspired by entropy, a central notion in information theory.

**Strengths:**

The source of inspiration is fundamental (information in the learning protocol).

The objective function is novel to the best of my knowledge, and many novel hyper-parameters (e.g. temperature) are intuitive and well-explained.

Analyses on convergence and fairness preservation are provided and comprehensive.

**Weaknesses:**

Section 5.2 carries significant importance as it assesses the fairness performance of the protocol. However, the analysis is based on regression only.

Although the authors' choice of definition of fairness (Definition 3.1, fairness via variance) is a common one, there are many more definitions available in the literature. Even in the cited original source of this definition (Li, et al 2019), they introduced a few other fairness metrics (cosine similarity and entropy). The theoretical analyses and experiments in this paper prove the FedEBA+'s efficiency for the chosen fairness definition for the regression task, but does not show the efficiency nor limitations for other definitions. It would also be beneficial if the authors could summarize some other commonly used definitions and comment on their advantages and disadvantages.

**Questions:**

As mentioned in the weakness section, there are some other fairness metrics, in the cited reference and some other ones. Would you provide any justifications to use this metric instead of others?

Did you try FedEBA+ on other tasks beyond regression?

---

> ### Author Response · Authors · 2023-11-16
> **Response to Reviewer 4hVe (1/3)**
>
> Dear reviewer 4hVe, thank you for providing constructive feedback. We have fully revised our manuscript and have addressed all of the comments, as well as added new experiments to further strengthen our work. Please find our responses to your raised questions below:
>
> > As mentioned in the weakness section, there are some other fairness metrics, in the cited reference (cosine similarity and entropy) and some other ones. Would you provide any justifications to use this metric instead of others? It would also be beneficial if the authors could summarize some other commonly used definitions and comment on their advantages and disadvantages.
>
> Thank you for your suggestion; we would like to clarify that:
>
> - Variance as fairness can be used to measure the difference between clients, designed to serve the interest of FL clients while aiming to enhance FL model performance during the process, ensuring each party has similar performance. Otherwise, this will hinder the clients’ participation interest. **In this paper, we use fairness because it is straightforward to follow, easy to compute, and is the most widely used in fairness FL works~[1,2,3,4,5], leading to a fair comparison.**
> - **We provide results under other fairness metrics, i.e., cosine similarity and entropy metrics**, as used in [1]. The geometric angle corresponds to the cosine similarity metric, and the KL divergence between the normalized accuracy vector $\mathbf{a}$ and the uniform distribution $\mathbf{u}$ can be directly translated to the entropy of $\mathbf{a}$. In the table below, it shows that FedEBA+ maintains its superiority over others on FashionMNIST in these fairness metrics.
>
> | Algorithms                       | Global Acc.  | Fairness Metric 1:Var. | Fairness Metric 2: Angle ($\circ$) - cosine similarity | Fairness Metric 3: KL ($a\|\|u$) -entropy |
> | -------------------------------- | ------------ | ---------------------- | ------------------------------------------------------ | --------------------------------------- |
> | FedAvg                           | 86.49 ± 0.09 | 62.44±4.55             | 8.70±1.71                                              | 0.0145±0.002                            |
> | q-FFL $q=0.001$                  | 87.05± 0.25  | 66.67± 1.39            | 7.97±0.06                                              | 0.0127±0.001                            |
> | FedMGDA+ $\epsilon=0.0$          | 84.64±0.25   | 57.89±6.21             | 8.21±1.71                                              | 0.0132±0.0004                           |
> | AFL $\lambda=0.7$                | 85.14±0.18   | 57.39±6.13             | 7.28±0.45                                              | 0.0124±0.0002                           |
> | PropFair $M=0.2,thres=0.2$       | 85.51±0.28   | 75.27±5.38             | 8.61±2.29                                              | 0.0139±0.002                            |
> | TERM $T=0.1$                     | 84.31±0.38   | 73.46±2.06             | 9.04±0.45                                              | 0.0137±0.004                            |
> | FedFV $\alpha=0.1, \tau_{FV}=10$ | 86.98±0.45   | 56.63±1.85             | 8.01±1.14                                              | 0.0111±0.0002                           |
> | FedEBA+ $\alpha=0.9 \tau=0.1$    | 87.50±0.19   | 43.41±4.34             | 6.46±0.65                                              | 0.0063±0.0009                           |
>
>
>
> - **We have summarized the commonly used definitions of fairness metrics and commented on their advantages and disadvantages in the table below.** Euclidean Distance and Pearson correlation coefficient are usually used for contribution fairness, and risk difference and Jain’s fairness Index are typically used for group fairness, which is a different target from performance fairness in this paper.
>
>   In particular, cosine similarity and entropy have similar roles to variance, used to measure the performance distribution among clients. The more uniform the distribution, the smaller the variance, the more similar to vector $1$, and the larger the entropy of the normalized performance. Thus, we only need one of them for performance fairness, and we use variance, which is the most widely used metric in related works.

---

> ### Author Response · Authors · 2023-11-16
> **Response to Reviewer 4hVe (2/3)**
>
> | Fairness metrics | Variance [1,2,5,6]                                           | cosine similarity [2,7,8]                                    | entropy [2,7,8]                                              | Euclidean Distance [9,10,11]                                 | Pearson Correlation Coefficient [12, 13]                     | Risk Difference [14,15,16]                         | Jain’s Fairness Index [17,18,19    ]                         |
> | ---------------- | ------------------------------------------------------------ | ------------------------------------------------------------ | ------------------------------------------------------------ | ------------------------------------------------------------ | ------------------------------------------------------------ | -------------------------------------------------- | ------------------------------------------------------------ |
> | Application      | Accuracy parity; performance fairness                        | same as variance                                             | same as variance                                             | Contribution fairness                                        | Contribution fairness                                        | Group fairness                                     | Group fairness; selection fairness; performance fairness; contribution fairness; |
> | Advantage        | Simplicity and straightforward to implement, as it focuses on a common performance metric. | similar to variance                                          | similar to variance                                          | Straightforward Interpretation; Sensitivity to Magnitude Differences | Scale Invariance; Captures Linear Relationships              | Sensitivity to Group Disparities; interpretability | Normalization Across Groups; Applicability to Various Metrics; Flexibility |
> | Disadvantage     | It only measures the relative fairness; Sensitive to outliers. | similar to variance; Does Not Capture Magnitude Differences; Sensitivity to Zero Vectors | similar to variance; Dependence on Normalization;Sensitivity to Client Number | Sensitivity to Scale; doesn’t account for direction          | Sensitive to Outliers; May Not Capture Magnitude Differences; the result may be inaccurate since it assumes that the two data variables have a linear relationship | Lack of Normalization                              | Sensitivity to Metric Choice; Complexity in interpretability |
>
> We have included the above discussion in Section 16 of our revised manuscript.
>
> [1]Zafar M B, Valera I, Gomez Rodriguez M, et al. Fairness beyond disparate treatment & disparate impact: Learning classification without disparate mistreatment[C] WWW 2017.
>
> [2]Li T, Sanjabi M, Beirami A, et al. Fair resource allocation in federated learning[J]. ICLR 2020.
>
> [3]Li T, Hu S, Beirami A, et al. Ditto: Fair and robust federated learning through personalization[C] ICML 2021.
>
> [4]Wang Z, Fan X, Qi J, et al. Federated learning with fair averaging[J]. Ijcai 2021
>
> [5]Hu Z, Shaloudegi K, Zhang G, et al. Federated learning meets multi-objective optimization[J]. TNSE 2022.
>
> [6]Shi Y, Yu H, Leung C. Towards fairness-aware federated learning[J]. TNNLS 2023.
>
> [7]Selbst A D, Boyd D, Friedler S A, et al. Fairness and abstraction in sociotechnical systems[C]//Proceedings of the conference on fairness, accountability, and transparency. 2019: 59-68.
>
> [8]Hardt M, Price E, Srebro N. Equality of opportunity in supervised learning[J]. NeurIPS 2016.
>
> [9]Aggarwal C C, Hinneburg A, Keim D A. On the surprising behavior of distance metrics in high dimensional space[C] Database Theory—ICDT 2001
>
> [10]Wei S, Tong Y, Zhou Z, et al. Efficient and fair data valuation for horizontal federated learning[J]. Federated Learning: Privacy and Incentive, 2020
>
> [11]Song T, Tong Y, Wei S. Profit allocation for federated learning[C] 2019 IEEE International Conference on Big Data (Big Data)
> [12]Jia R, Dao D, Wang B, et al. Towards efficient data valuation based on the shapley value[C] AISTATS 2019.
>
> [13]Wang G, Dang C X, Zhou Z. Measure contribution of participants in federated learning[C] 2019 IEEE international conference on big data (Big Data).
>
> [14]Du W, Xu D, Wu X, et al. Fairness-aware agnostic federated learning[C] SIAM on Data Mining (SDM) 2021.
>
> [15]Dwork C, Hardt M, Pitassi T, et al. Fairness through awareness[C] ITCS 2012.
>
> [16]Hardt M, Price E, Srebro N. Equality of opportunity in supervised learning[J]. NeurIPS 2016.
>
> [17]Chiu D M. A quantitative measure of fairness and discrimination for resource allocation in shared computer systems[R]. Digital Equipment Corporation, 1984.
>
> [18]Liu H, Zhang X, Shen X, et al. A fair and efficient hybrid federated learning framework based on XGBoost for distributed power prediction[J]. arXiv:2201.02783, 2022.
>
> [19]Cho Y J, Gupta S, Joshi G, et al. Bandit-based communication-efficient client selection strategies for federated learning[C] ACSSC 2020.

---

> ### Author Response · Authors · 2023-11-16
> **Response to Reviewer 4hVe (3/3)**
>
> > **Did you try FedEBA+ on other tasks beyond regression?**
> >
>
> Thank you for your valuable suggestion.
>
> **In theoretical analysis, beyond regression, we have tried to analyze FedEBA+ on smooth and strongly convex loss functions.** We have added this in Section 12.3 of the revised manuscript and we would like to state the result and idea here:
>
> - We show the performance variance of FedEBA+ is smaller than that of FedAvg:
> - We start by considering two client cases with smooth and strongly convex functions:
>     - one of them is considered to be the outlier, which means the loss value is much larger than others in the parameter space and the optimal model parameter is far away from others.
>     - For ease of computing and expression, let current $x_t=0$. Consider each client’s model after local update lies on its optimal parameter, w.l.o.g, assume Client 1 with optimal value $x_1^*$  and $F_1(x_1^*)=0$. Similarly, Client 2 (outlier) with optimal value $x_2^*$ that is far away from $x_1^*$ (w.l.o.g, assume $x_2^*>x_1^*$) and $F_2(x_2^*)=a>0$ (relative position, which does not affect the analysis).
>     - By consider the entropy-based aggregation probability (FedEBA+) and average aggregation (FedAvg), we obtain $\tilde{x}=p_1x_1^* +p_2x_2^*$ and $\overline{x}=\frac{1}{2}(x_1^*+x_2^*)$.
>     - By some derivation, we show that comparing variance is equal to comparing $\|\|F_1(\tilde{x})-F_2(\tilde{x})\|\|^2$ and $\|\|F_1(\overline{x})-F_2(\overline{x})\|\|^2$.
>     - Due to the property of strongly convex and smooth, the outlier $F_2(x)$ monotonically decreases on $(x_1^*,x_2^*)$. Combining the condition that $F_2(x)$ is an outlier, we can conclude that the smaller $\|\|x-x_2\|\|^2$ is, the smaller variance.
>     - By the defination of entropy-based aggregation, we know $p_1 = \frac{1}{1+e^a},p_2 = 1-p_1$, leading to $\|\|\tilde{x}-x_2\|\|^2 <\|\|\overline{x}-x_2\|\|^2$, and $\|\|F_1(\tilde{x})-F_2(\tilde{x})\|\|^2<\|\|F_1(\overline{x})-F_2(\overline{x})\|\|^2$. Thus in this case, we prove FedEBA+ achieves a smaller variance than FedAvg.
> - Extending the conclusion to any number of clients, we use mathematical induction to show that：
>     - Assume for $N$ client, the variance of FedBEA+ is smaller than that of FedAvg
>     - For $N+1$  client, we derive the variance as $\mathbf{Var}^{N+1}=\frac{N}{N+1}\mathbf{Var}^{N}+\frac{\sum_{i=1}^N(F_i-F_{N+1})^2}{(N+1)^2}$.
>     - Thus, to prove $\mathbf{Var_{EBA+}}^{N+1}<\mathbf{Var_{Avg}}^{N+1}$, it is equal to prove $\sum_{i=1}^N(F_i-F_{N+1})^2$. Since Client $N+1$ is an outlier, by the same analysis as we used in two clients case, we can prove for each client $i\in [N]$, $\|\|F_i(\tilde{x})-F_{N+1}(\tilde{x})\|\|^2<\|\|F_i(\overline{x})-F_{N+1}(\overline{x})\|\|^2$.
> - Thus, by combining the above two cases, we prove in strongly convex and smooth cases, the variance of FedEBA+ is smaller than FedAvg.
>
> In addition, the general analysis of performance fairness in FL remains an open challenge in this field. We would like to point out that the majority of existing fairness FL research employs simplified regression models to analyze fairness [1,2] or even lacks a fairness analysis altogether [3,4,5,6,7,8]. Only very limited work analyzes performance fairness (variance) [9] due to the natural property of the designed objective function $f(x)=(\frac{1}{2}\sum_{i=1}^mF_i(x)^2)^{\frac{1}{2}}$.
>
> We have applied FedEBA+ to many different models, including MLP, CNN, Resnet-18, and Mobilenet-v2, beyond regression.
>
> [1]Lin S, Han Y, Li X, et al. Personalized Federated Learning towards Communication Efficiency, Robustness and Fairness[J]. NeurIPS, 2022.
>
> [2]Chu W, Xie C, Wang B, et al. FOCUS: Fairness via Agent-Awareness for Federated Learning on Heterogeneous Data[J]. arXiv preprint arXiv:2207.10265, 2022.
>
> [3]Wang Z, Fan X, Qi J, et al. Federated learning with fair averaging[J]. arXiv preprint arXiv:2104.14937, 2021.
>
> [4]Hu Z, Shaloudegi K, Zhang G, et al. Federated learning meets multi-objective optimization[J]. IEEE Transactions on Network Science and Engineering, 2022, 9(4): 2039-2051.
>
> [5]Li T, Hu S, Beirami A, et al. Ditto: Fair and robust federated learning through personalization[C]//International Conference on Machine Learning. PMLR, 2021: 6357-6368.
>
> [6]Du W, Xu D, Wu X, et al. Fairness-aware agnostic federated learning[C]//Proceedings of the 2021 SIAM International Conference on Data Mining (SDM). Society for Industrial and Applied Mathematics, 2021: 181-189.
>
> [7]Kanaparthy S, Padala M, Damle S, et al. Fair Federated Learning for Heterogeneous Data[C]//5th Joint International Conference on Data Science & Management of Data (9th ACM IKDD CODS and 27th COMAD). 2022: 298-299.
>
> [8]Zhao Z, Joshi G. A dynamic reweighting strategy for fair federated learning[C]//ICASSP 2022-2022 IEEE International Conference on Acoustics, Speech and Signal Processing (ICASSP). IEEE, 2022: 8772-8776.
>
> [9]Li T, Sanjabi M, Beirami A, et al. Fair resource allocation in federated learning[J]. ICLR 2020.

---

> ### Author Response · Authors · 2023-11-20
> **Looking forward to your feedback**
>
> Dear reviewer 4hVe,
>
> Thank you very much for your valuable review, which significantly enhanced our manuscript's quality.
>
> As the discussion period is ending soon, we would like to know if our responses have addressed your concerns. Please let us know if there are any additional clarifications or experimental evaluations that you believe would contribute to the improvement of our manuscript.
>
> Your expertise and the time you've dedicated to reviewing our manuscript is deeply appreciated.
>
> Warmest regards,
>
> Authors

---

> > ### Comment · Reviewer_4hVe · 2023-11-23
> > **Response to the authors**
> >
> > I appreciate the authors' feedback. My concern for Q1 (choice of fairness metrics) has been well-addressed. However, the additional analysis on cases beyond regression is still fairly limited, since strong convexity must be assumed. Therefore, I will keep my original assessment.

---

> > > ### Author Response · Authors · 2023-11-23
> > > **To Reviewer 4hVe**
> > >
> > > Dear Reviewer 4hVe,
> > >
> > > Thank you for your response.
> > > We would like to highlight that the general analysis of performance fairness in FL remains an open challenge in this field.
> > >
> > > - The majority of existing fairness FL research employs simplified regression models to analyze fairness [1,2] or even lacks a fairness analysis altogether [3,4,5,6,7,8].
> > > - Limited research analyzes general performance fairness (variance) [9,10] due to the natural property of the designed objective function, like square function.
> > > - Notably, our convergence analysis is based on a nonconvex setting, while the new fairness analysis relies on a strongly convex setting.
> > >
> > > [1]Lin S, Han Y, Li X, et al. Personalized Federated Learning towards Communication Efficiency, Robustness and Fairness[J]. Advances in Neural Information Processing Systems, 2022.
> > >
> > > [2]Chu W, Xie C, Wang B, et al. FOCUS: Fairness via Agent-Awareness for Federated Learning on Heterogeneous Data[J]. arXiv preprint arXiv:2207.10265, 2022.
> > >
> > > [3]Wang Z, Fan X, Qi J, et al. Federated learning with fair averaging[J]. arXiv preprint arXiv:2104.14937, 2021.
> > >
> > > [4]Hu Z, Shaloudegi K, Zhang G, et al. Federated learning meets multi-objective optimization[J]. IEEE Transactions on Network Science and Engineering, 2022, 9(4): 2039-2051.
> > >
> > > [5]Li T, Hu S, Beirami A, et al. Ditto: Fair and robust federated learning through personalization[C]//International Conference on Machine Learning. PMLR, 2021: 6357-6368.
> > >
> > > [6]Du W, Xu D, Wu X, et al. Fairness-aware agnostic federated learning[C]//Proceedings of the 2021 SIAM International Conference on Data Mining (SDM). Society for Industrial and Applied Mathematics, 2021: 181-189.
> > >
> > > [7]Kanaparthy S, Padala M, Damle S, et al. Fair Federated Learning for Heterogeneous Data[C]//5th Joint International Conference on Data Science & Management of Data (9th ACM IKDD CODS and 27th COMAD). 2022: 298-299.
> > >
> > > [8]Zhao Z, Joshi G. A dynamic reweighting strategy for fair federated learning[C]//ICASSP 2022-2022 IEEE International Conference on Acoustics, Speech and Signal Processing (ICASSP). IEEE, 2022: 8772-8776.
> > >
> > > [9]Li T, Sanjabi M, Beirami A, et al. Fair resource allocation in federated learning[J]. ICLR 2020.
> > >
> > > [10]Sultana A, Haque M M, Chen L, et al. Eiffel: Efficient and fair scheduling in adaptive federated learning[J]. IEEE Transactions on Parallel and Distributed Systems, 2022, 33(12): 4282-4294.

---

### Official Review · Reviewer_ABP1 · 2023-10-31

**Soundness:** 2 fair
**Presentation:** 2 fair
**Contribution:** 3 good
**Rating:** 5
**Confidence:** 3

**Summary:**

The paper introduces FedEBA+, an algorithm designed to increase fairness without sacrificing overall performance. It begins by formulating a constrained optimization problem to optimize the entropy of the aggregation probability while constraining the model to achieve ideal performance.

The algorithm includes convergence analysis, variance analysis, and pareto-optimality analysis. Additionally, the authors evaluate FedEBA+ on several datasets against various baseline algorithms, demonstrating its superior performance in terms of global accuracy and fairness.

**Strengths:**

1. The paper addresses an important question in federated learning systems: how to achieve fairness without sacrificing accuracy.
2. The proposed approach is supported by convergence analysis and fairness guarantees.
3. The proposed approach's effectiveness is demonstrated by comparing it with a significant number of baselines and presenting experimental results.

**Weaknesses:**

1. [Major] It is unclear which contributes to the improved performance, the optimization algorithm, or the entropy-based fairness regularization.
2. [Medium] The performance of the proposed FedEBA+ doesn’t seem to be significant, especially in terms of worst 5% accuracy.

**Questions:**

See the weakness section.

---

> ### Author Response · Authors · 2023-11-16
> **Response to Reviewer ABP1 (1/2)**
>
> Dear reviewer ABP1, thanks for your time in reviewing our paper, please find our responses to your raised questions:
>
> > [Major] It is unclear which contributes to the improved performance, the optimization algorithm, or the entropy-based fairness regularization.
> >
>
> Thank you for your comment.
>
> We would like to clarify that
>
> - FedEBA+ is an optimization algorithm comprising two parts: the proposed entropy-based aggregation (derived from Eq 2) and the proposed alignment update (derived from Eq 6). The alignment update includes both model alignment (Sec 4.2.1, Line 45 of Algorithm 1) and gradient alignment (Sec 4.2.1, Line 45 of Algorithm 1). Although these are distinct components, they collectively form one part of our optimization algorithm. Thus, the optimization algorithm, integrating a new aggregation and a new alignment update process, contributes to improved performance.
> - The ablation results of FedEBA+ demonstrate that each step, i.e., aggregation and alignment update, is beneficial to improved performance, as shown in Table 1 of the original submission. We present the ablation study here to illustrate that both aggregation and alignment updates contribute to the performance:
>     - FedEBA corresponds to the algorithm that only uses entropy-based aggregation compared with FedAvg; FedEBA+ indicates that the algorithm uses both entropy-based aggregation and alignment update.
>     - Comparing FedEBA with FedAvg, FedEBA improves accuracy by about 1.6 and fairness by about 14.3 on CIFAR-10, highlighting the proposed aggregation approach’s advantages.
>     - comparing FedEBA+ and FedEBA, FedEBA+ improves accuracy by about 3.4 and fairness by about 20.8 on CIFAR-10, showing the effectiveness of the proposed alignment update approach.
>     - Similar comparisons can be observed in FashionMNIST, CIFAR-100, and Tiny-ImageNet.
>
> |  | CIFAR-10 |  |  |  | FashionMNIST |  |  |  |
> | --- | --- | --- | --- | --- | --- | --- | --- | --- |
> | Algorithm | Global Acc. | Var. | Worst 5% | Best 5% | Global Acc. | Var. | Worst 5% | Best 5% |
> | FedAvg | 67.79±0.35 | 103.83±10.46 | 45.00±2.83 | 85.13±0.82 | 86.49 ± 0.09 | 62.44±4.55 | 71.27±1.14 | 95.84± 0.35 |
> | FedEBA | 69.38±0.52   | 89.49±10.95 | 50.40±1.72  | 86.07±0.90 | 86.70±0.11 | 50.27±5.60 | 71.13±0.69 | 95.47±0.27 |
> | FedEBA+ | 72.75±0.25  | 68.71±4.39  | 55.80±1.28  | 86.93±0.52 | 87.50±0.19 | 43.41±4.34 | 72.07±1.47 | 95.91±0.19 |
> |  | **CIFAR-100** |  |  |  | **Tiny-ImageNet** |  |  |  |
> | Algorithm | Global Acc. | Var. | Worst 5% | Best 5% | Global Acc. | Var. | Worst 5% | Best 5% |
> | FedAvg | 30.94±0.04 | 17.24±0.08 | 0.20±0.00 | 65.90±1.48 | 61.99±0.17 | 19.62±1.12 | 53.60±0.06 | 71.18±0.13 |
> | FedEBA | 32.38±0.13 | 17.09±0.06 | 0.75±0.22 | 66.40±0.47 | 63.34±0.25 | 15.29±1.36 | 54.17±0.04 | 70.98±0.10 |
> | FedEBA+ | 33.39±0.22 | 16.92±0.04 | 0.95±0.15 | 68.51±0.21 | 64.05±0.09 | 14.91±1.85 | 54.32±0.09 | 71.27±0.04 |
>
> The above Table has been added as Table 11 of the revised version.

---

> ### Author Response · Authors · 2023-11-16
> **Response to Reviewer ABP1 (2/2)**
>
> > [Medium] The performance of the proposed FedEBA+ doesn’t seem to be significant, especially in terms of worst 5% accuracy.
> >
>
> Thank you for your suggestion.
>
> We would like to clarify that FedEBA+ aims to solve the challenge of improving fairness and the global model’s accuracy simultaneously. Thus, the interpretation of experimental results should consider both.
>
> - Figure 3 visually demonstrates that when considering both fairness and accuracy, FedEBA+ outperforms others significantly, positioning itself in the lower right corner while other algorithms cluster away.
> - To better understand the simultaneous improvement in fairness and accuracy, we introduce the *coefficient of variation (CV)* as a metric to measure the relative fairness level, as used in [1,2]. This relative statistic, defined as the ratio of the standard deviation to the mean ($C_v=\frac{std}{acc}$), captures both fairness and accuracy simultaneously. The table below presents the $C_v$ of algorithms on FashionMNIST and CIFAR-10 (Table 1 of our original submission), showing a substantial improvement with FedEBA+.
>     - From the table below, we can observe that when considering both fairness and accuracy simultaneously, FedBEA+ improves the performance to 21% in FashionMNIST, while the others achieve at most 4.8% (4.4x), and in CIFAR-10, FedEBA+ improves by 27.8%, while others achieve at most 13.3% (2.1x).
>
> |  | FashionMNIST |  | CIFAR-10 |  |
> | --- | --- | --- | --- | --- |
> | Algorithm | $C_v=\frac{std}{acc}$ | $C_v$ improvement over FedAvg | $C_v=\frac{std}{acc}$ | $C_v$ improvement over FedAvg |
> | FedAvg | 0.09136199 | 0% | 0.150312741 | 0% |
> | qffl | 0.112432356 | -23% | 0.144026806 | 4.2% |
> | FedMGDA+ | 0.089893051 | 1.3% | 0.146896915 | 2.4% |
> | AFL | 0.088978374 | 2.6% | 0.134878199 | 10.1% |
> | PropFair | 0.101459812 | -11.3% | 0.135671155 | 10.9% |
> | TERM | 0.101659126 | -10.1% | 0.146631123 | 2.7% |
> | FedFV | 0.086517483 | 4.8% | 0.130809249 | 13.3% |
> | FedEBA+ | 0.072539115 | 21.8% | 0.1139402 | 27.8% |
> - Even when looking at the performance boosts in accuracy and fairness alone, from the results in Table 1, we can conclude that:
>     - In FashionMNIST, the accuracy improvement of our algorithm is twice as large (**2×**) as the state-of-the-art (SOTA) algorithm FedFV, while the fairness improvement is **3×** larger than the SOTA algorithm FedFV.
>     - In CIFAR-10, the accuracy improvement of our algorithm is about **1.5×** than the SOTA algorithm FedFV, while the fairness improvement is **1.45×** larger than SOTA PropFair.
>
> Regarding the worst 5% accuracy:
>
> - While our primary focus is not solely on improving the worst clients' performance, the enhancement of fairness inherently improves the under-performing clients. FedEBA+ consistently delivers the best worst 5% performance in most cases, including CIFAR-10 and Tiny-Imagenet, as demonstrated in Tables 1 and 2 of our original submission.
> - AFL outperforms FedEBA+ in CIFAR-100 by focusing on the worst clients. However, this approach sacrifices overall performance, resulting in poor global accuracy, variance, and best 5% accuracy. A similar outcome is seen with FedMGDA+ on FashionMNIST.
>
> The above results have been added in Table 13 of Appendix 14 of the revised manuscript.
>
> [1]Jain R K, Chiu D M W, Hawe W R. A quantitative measure of fairness and discrimination[J]. Eastern Research Laboratory, Digital Equipment Corporation, Hudson, MA, 1984, 21.
>
> [2]Pitoura T, Triantafillou P. Distribution fairness in Internet-scale networks[J]. ACM Transactions on Internet Technology (TOIT), 2009, 9(4): 1-36.

---

> ### Author Response · Authors · 2023-11-20
> **Looking forward to your feedback**
>
> Dear reviewer ABP1,
>
> We greatly appreciate your insightful review, which plays a crucial role in enhancing the quality of our manuscript. Furthermore, we've incorporated comments from other reviewers to further enhance our paper. We summarize the revisions in the "****Summary of Revision****" global comment.
>
> As the discussion period is ending soon, we want to confirm that our responses, including the clarification of the algorithm’s contribution and showcasing the improvement of FedEBA+, have adequately addressed your concerns. We are happy to take any follow-up questions and look forward to discussing with you.
>
> We sincerely appreciate your expertise and dedicated time in reviewing our manuscript.
>
> Warmest regards,
>
> Authors

---

> > ### Comment · Reviewer_ABP1 · 2023-11-22
> > **Thanks for your response and new experiments**
> >
> > Q1: Thanks for the new experiments. However, if the key idea of this paper is to use the proposed two components together to improve the results, then I think a more thorough empirical study should be conducted to understand each component.
> >
> > Q2: The response cleared my question.
> >
> > Since my concern regarding Q1 is not fully addressed, I will keep my original evaluation.

---

> ### Author Response · Authors · 2023-11-22
> **Thank you for your response**
>
> Dear Reviewer ABP1,
>
> Thank you for your response and patience.
>
> - As motivated by your suggestion, we provide a thorough ablation study of FedEBA+ in terms of two components, i.e., the proposed entropy-based **aggregation** and the proposed **alignment** **update**. In particular:
>     1. **FedAvg** serves as the backbone.
>     2. **FedAvg + aggregation** is employed to demonstrate the individual effectiveness of our proposed aggregation step, which is effective in reducing variance and has the ability to increase accuracy compared with FedAvg.
>     3. `[New results]` **FedAvg + alignment** is utilized to showcase the individual effectiveness of our proposed alignment step, which is effective in improving accuracy and has the ability to reduce variance compared with FedAvg.
>     4. **FedAvg + aggregation + alignment** is used to show the effectiveness of our proposed algorithm, FedEBA+. FedEBA+ incorporates these two steps to effectively enhance fairness and accuracy.
>
> |  | CIFAR-10 |  |  |  | FashionMNIST |  |  |  |
> | --- | --- | --- | --- | --- | --- | --- | --- | --- |
> | Algorithm | Global Acc. | Var. | Worst 5% | Best 5% | Global Acc. | Var. | Worst 5% | Best 5% |
> | FedAvg | 67.79±0.35 | 103.83±10.46 | 45.00±2.83 | 85.13±0.82 | 86.49 ± 0.09 | 62.44±4.55 | 71.27±1.14 | 95.84± 0.35 |
> | FedAvg+aggregation | 69.38±0.52 | 89.49±10.95 | 50.40±1.72 | 86.07±0.90 | 86.70±0.11 | 50.27±5.60 | 71.13±0.69 | 95.47±0.27 |
> | FedAvg+alignment  | 72.04±0.51 | 75.73±4.27 | 53.45±1.25 | 87.33±0.23 | 87.42± 0.09 | 60.08±7.30 | 69.12±1.23 | 97.8±0.19 |
> | FedAvg+aggregation+alignment | 72.75±0.25 | 68.71±4.39 | 55.80±1.28 | 86.93±0.52 | 87.50±0.19 | 43.41±4.34 | 72.07±1.47 | 95.91±0.19 |
> - In addition, we conducted an ablation study to demonstrate the effectiveness of each update method within the proposed alignment step, including fair gradient alignment ($\theta=0$) and global model alignment ($\theta=90^\circ)$, as shown in [the ablation study of $\theta$](https://openreview.net/forum?id=UJeIujVxMn&noteId=oQd18SUjPP).
>
> The new ablation results for FedEBA+ have been incorporated into Table 11 of our revised manuscript. The ablation study for $\theta$, which reveals the results of the alignment step (i.e., fair gradient alignment and global model alignment), is available in Table 5 of both the original and revised versions.
>
> We sincerely appreciate your thoughtful consideration of the comprehensive ablation studies, which were conducted with the aim of addressing your concerns. It would be highly appreciated if you could reconsider your score in light of our responses to the issues raised.
>
> Thank you once again for your valuable comments, which have significantly contributed to the improvement of our paper.

---

> > ### Author Response · Authors · 2023-11-23
> > **To Reviewer ABP1: Thank you for your patience**
> >
> > Dear Reviewer ABP1,
> >
> > Thank you for your patience.
> >
> > In addition to FashionMNST and CIFAR-10, we have conducted a comprehensive ablation study of FedEBA+ on CIFAR-100 and Tiny-ImageNet.
> > |  | CIFAR-100 |  |  |  | Tiny-ImageNet |  |  |  |
> > | --- | --- | --- | --- | --- | --- | --- | --- | --- |
> > | Algorithm | Global Acc. | Var. | Worst 5% | Best 5% | Global Acc. | Var. | Worst 5% | Best 5% |
> > | FedAvg | 30.94±0.04 | 17.24±0.08 | 0.20±0.00 | 65.90±1.48 | 61.99±0.17 | 19.62±1.12 | 53.60±0.06 | 71.18±0.13 |
> > | FedAvg+aggregation | 32.38±0.13 | 17.09±0.06 | 0.75±0.22 | 66.40±0.47 | 63.34±0.25 | 15.29±1.36 | 54.17±0.04 | 70.98±0.10 |
> > | FedAvg+alignment  | 31.93±0.39 | 17.15±0.05 | 0.39±0.01 | 66.04±0.16 | 63.46±0.04 | 14.52±0.21 | 54.36±0.03 | 71.13±0.03 |
> > | FedAvg+aggregation+alignment | 33.39±0.22 | 16.92±0.04 | 0.95±0.15 | 68.51±0.21 | 64.05±0.09 | 14.91±1.85 | 54.32±0.09 | 71.27±0.04 |
> >
> > The new results consistently demonstrate the effectiveness of FedEBA+.

---

### Official Review · Reviewer_yc23 · 2023-11-09

**Soundness:** 2 fair
**Presentation:** 2 fair
**Contribution:** 2 fair
**Rating:** 5
**Confidence:** 3

**Summary:**

The paper addresses the challenge of achieving fairness in FL without compromising the performance of the global model. The proposed FedEBA+ algorithm employs an entropy-based aggregation strategy to give higher weights to underperforming clients and an alignment update method to improve fairness and global model performance. The paper provides convergence analysis and fairness proof for FedEBA+, and shows that it outperforms existing state-of-the-art methods on several datasets in terms of both fairness and global model performance. The paper also conducts ablation studies to evaluate the impact of hyperparameters.

**Strengths:**

- The FedEBA+ algorithm addresses the accuracy-fairness trade-off issue. The extensive results on four datasets show that FedEBA+ reduces variance without compromising the performance of clients.
- An ablation study highlights the advantages of the proposed aggregation approach of FedEBA+. Specifically, it shows that the aggregation strategy alone can differ from FedAvg, while the addition of the aligned update ($\alpha$ >0) further improves performance, demonstrating the effectiveness of each component of the algorithm.
- Theoretically, the authors provide convergence analysis and fairness analysis.

**Weaknesses:**

The algorithm may have flaws:
- The communication overhead appears to be much higher than FedAvg in Algorithm 1.  According to Line 3  and Line 9,  clients need to communicate with the server twice at each round t. This contradicts the traditional FedAvg that only communicates once at each round.
- In Line 9, $\tilde{g}^t$ is the fair gradient of the selected clients obtained using one local update. However, there is no step before line 9 that collects the one local update from clients. The authors should clarify this process.
-  In algorithm 1, Line 10  indicates that Eq 3 will use the local model $x_{t,K}^j$ as $x$ to calculate the $p_j$. However, using local model $x_{t,K}^j$ for Eq 3 leads to different denominators $\sum_{i=1}^N \exp[F_i(x_{t,K}^j)/\tau]$ for different client $j$. Then $p_j$ for different clients is not normalized by the same denominator.  I am not sure how it satisfies the constraint  $\sum_j p_j=1$ in Eq 2. An explanation of how this approach satisfies the constraint would be helpful.



Writing:
- Although the high-level idea is communicated well, the mathematical presentation could be improved for better accessibility. The paper's current use of a complex and inconsistent notation system hampers understanding.   For example, it seems that the notations, $\nabla \tilde{f}(x)$, $\tilde{\Delta}_t$,  and $\tilde{g}^t$ are interchangeably used to represent the FedSGD global gradient,  (e.g., the aggregation of one-step local updates) in different sections, causing confusion.
- It would be clearer if the authors could connect Eq 8 and Eq 10 to Eq 6, given that they share a similar form and intuition. For example, Eq 6 is not referenced in Section 4.2.2, thus making the purpose of Equation 10 unclear.

- The motivation of maximum entropy for fairness may need further clarification.  Though the authors discussed fairness in the introduction and mentioned the Shannon entropy in Section 4.1, the connection between entropy and fairness is not intuitively clear. More explicit reasoning or examples illustrating why higher entropy equates to greater fairness would be beneficial, especially in federated learning settings.

- From equation (2),  the term “ideal loss” $\tilde{f}(x)$ is introduced without a clear definition or expression, leading to confusion that is not resolved until later sections (4.2.1 and 4.2.2).  It would be beneficial if the authors could provide more discussion or examples for  “ideal loss” here.


Clarification:

- The fairness of FedEBA+ is partly achieved based on the assumption that the FedSGD global gradient is an ideal fair gradient. Specifically, according to the end of Section 4.2, the authors view a FedSGD global gradient as the ideal global gradient and the ideal fair gradient. Clarity is needed on why this assumption is made and whether the FedSGD gradient truly encapsulates ideal fairness.  Also, it would be helpful to consider FedSGD as one fairness baseline in the experiments, to demonstrate the effect of another component in FedEBA+, maximum entropy aggregation, for fairness.


- The following statement is not clear: “The specific challenge lies in finding a way to introduce entropy into the FL training process while incorporating FL’s uniform performance as a constraint within the framework.”  Isn’t maximum entropy encouraging uniform performance? (Intuitively,  the outcomes with uniform probability give the highest entropy, i.e., uncertainty, based on the definition of entropy)  Does maximum entropy go against the goal of uniform performance? The reason why these concepts are treated distinctly in this statement requires clarification.

Theory:
- The claimed convergence rate is not appropriate because there is a non-vanishing constant error term, which means that the gradient of the algorithm will never approach zero. The convergence rate is O(1) instead. This finding is inconsistent with previous studies, such as FedAvg convergence under the non-iid setting [1], which does not have the constant error term.

- The Theorem 5.3 claims that FedEBA+ achieves a smaller variance than FedAvg. However, the effect of FL data distributions is not reflected in Theorem 5.3. It would be clearer if the authors could justify how non-iid degree will affect the fairness results. Moreover, could the authors provide definitions for T() and A() in the variance analysis?

Experiments:
- The algorithms introduce a few additional hyperparameters: temperature $\tau$, angle threshold $\theta$, and $\alpha$. From Figure 4, it appears that the performance of the algorithm is quite sensitive to the hyperparameters, and different datasets have different optimal hyperparameters. It indicates a need for extensive hyperparameter tuning across different datasets. This requirement potentially reduces the algorithm's practicality compared to more straightforward federated learning algorithms like FedAvg.
- The influence of the angle threshold $\theta$ on FedEBA+'s performance is not demonstrated. It would be beneficial to provide an ablation study on  $\theta$.


Reference:

[1] On the Convergence of FedAvg on Non-IID Data, ICLR 2020

**Questions:**

Please see my questions in Weaknesses.

---

> ### Author Response · Authors · 2023-11-16
> **Response to Reviewer yc23 (1/7)**
>
> Dear reviewer yc23, we would like to thank you for your time spent reviewing our paper and for providing constructive comments. Please kindly find our responses to your raised questions below:
>
> ### For algorithm:
>
> > According to Line 3 and Line 9, clients need to communicate with the server twice at each round t. This contradicts the traditional FedAvg that only communicates once at each round.
> >
>
> Thank you for your suggestions. We'd like to clarify the communication issue in Algorithm 1:
>
> - **Communication time is less than twice that of FedAvg.** Apologies for the unclear description of Algorithm 1 in our original submission, which caused confusion that clients need to communicate with the server twice. We have carefully revised descriptions of Algorithm 1 in the revised manuscript. Actually, as shown in Algorithm 1 of the revised version, only the communication capacity is sufficient (Line 3) and the arccos between clients performance and $\mathbf{1}$ is larger than the threshold (Line 5), the server needs one additional communication (Line 6) compared with FedAvg, otherwise the communication cost is not doubled.
>     - **Communication cost is limited and acceptable:** (1) In Line 3, the client updates only the loss, with negligible cost (for example, in our CIFAR-10 experiment, the communication cost of a loss is $e^{-6}$MB) compared to the model communication. (2) In Line 11, additional cost involves uploading one-step gradients, but, as per our $\theta$ ablation study, this cost is minimal and leads to satisfactory results. For instance, as shown in Appendix 11, Table 5, in the original submission, on FashionMNIST, when the additional communication ratio (additional communication as a percentage of total communications) is 0 ($\theta=90^\circ$) or 2.1% ($\theta=45^\circ$), FedEBA+ achieves the accuracy 87% comparable with best baselines, and improves the variance to minimal 48.91, far outperforming others.
> - **FedEBA+ differs from FedAvg for a reason:** FedEBA+ aims to improve accuracy and ensure fairness simultaneously. FedEBA+ can be reverted to the FedAvg framework by setting $\theta$ large enough, making Line 3 unnecessary. Even so, FedEBA+ outperforms FedAvg in both accuracy and fairness, as demonstrated in our $\theta$ ablation study in Table 5 of Appendix 15, in the revised manuscript.
>
> We've updated Algorithm 1 in our revised version.
>
> > In Line 9,  $\tilde{g}$ is the fair gradient of the selected clients obtained using one local update. However, there is no step before line 9 that collects the one local update from clients. The authors should clarify this process.
> >
>
> Thank you for your thoughtful reminder. Sorry for the confusion caused by our abbreviated description of Algorithm 1 in the original submission. In the revised manuscript, we have amended Line 6
> “Calculate $\tilde{g}^t$ by (9)”
> into
>
> “Sever collects the gradient $\nabla F_i(x_t)$, calculates the ideal fair gradient $\tilde{g}^{b,t}$ by (9) and sends $\tilde{g}^{b,t}$ to selected clients.”
>
> > In algorithm 1, Line 10 indicates that Eq 3 will use the local model  $x_{t,K}^i$ as $x$ to calculate the $p_j$ . However, using local model $x_{t,K}^i$ for Eq 3 leads to different denominators $\sum_{i=1}^N\exp({F_i(x_{t,K}^i)/\tau})$ for different client . Then $p_j$ for different clients is not normalized by the same denominator. I am not sure how it satisfies the constraint $\sum_jp_j=1$ in Eq 2. An explanation of how this approach satisfies the constraint would be helpful.
> >
>
> Thank you for your suggestions. We would like to explain how this approach satisfies the constraint:
>
> 1. In Line 10 of Algorithm 1, Eq 3 corresponds to $p_i = \frac{\exp({F_i(x_{t,K}^i)/\tau})}{\sum_{j=1}^N\exp({F_j(x_{t,K}^i)/\tau})}$. Despite different clients having different $F_i(x_{t,K}^i)$, the denominators among clients are the same, i.e., $\sum_{j=1}^N\exp({F_j(x_{t,K}^i)/\tau})$.
> 2. We verify $\sum_i p_i = 1$ by $\sum_{i}p_i=p_1+p_2+\dots+p_N = \frac{\exp({F_1(x_{t,K}^1)/\tau})}{\exp({F_1(x_{t,K}^1)/\tau})+\exp({F_2(x_{t,K}^2)/\tau})+\dots+\exp({F_N(x_{t,K}^N)/\tau})}+\frac{\exp({F_2(x_{t,K}^2)/\tau})}{\exp({F_1(x_{t,K}^1)/\tau})+\exp({F_2(x_{t,K}^2)/\tau})+\dots+\exp({F_N(x_{t,K}^N)/\tau})}+\cdots+\frac{\exp({F_N(x_{t,K}^N)/\tau})}{\exp({F_1(x_{t,K}^1)/\tau})+\exp({F_2(x_{t,K}^2)/\tau})+\dots+\exp({F_N(x_{t,K}^N)/\tau})} $
> $= \frac{\exp({F_1(x_{t,K}^1)/\tau})+\exp({F_2(x_{t,K}^2)/\tau})+\dots+\exp({F_N(x_{t,K}^N)/\tau})}{\exp({F_1(x_{t,K}^1)/\tau})+\exp({F_2(x_{t,K}^2)/\tau})+\dots+\exp({F_N(x_{t,K}^N)/\tau})}=1$.
> 3. Moreover, using the same reasoning for $i\in S_t$ instead of $i \in [N]$, the result remains unchanged, i.e., $\sum_{i\in S_t}p_i = 1$.

---

> ### Author Response · Authors · 2023-11-16
> **Response to Reviewer yc23 (2/7)**
>
> ### Writing:
>
> > The paper's current use of a complex and inconsistent notation system hampers understanding. For example, it seems that the notations, $\nabla\tilde{f}$, $\tilde{\Delta}_t$, and $\tilde{g}^t$ are interchangeably used to represent the FedSGD global gradient, (e.g., the aggregation of one-step local updates) in different sections, causing confusion.
> >
>
> Apologies for any confusion caused by the notations. We'd like to clarify:
>
> - Only $\tilde{\Delta}_t$ represents the average aggregation of one-step local updates, i.e., FedSGD. In contrast, $\tilde{g}_t$ represents the reweighted aggregation of the current global model’s gradient on each local dataset.
> - To prevent confusion with the interchangeability of $\nabla \tilde{f}(x_t),\tilde{\Delta}_t,\tilde{g}_t$ in different sections, we will use:
>     - $\nabla \tilde{f}^{a}(x_t)=\tilde{\Delta}t^a=\frac{1}{|S_t|}\sum_{i\in S_t}(x_{t,1}^i-x_{t,0}^i)$ for Section 4.2.1 to represent the ideal global gradient (the FedSGD global gradient).
>     - $\nabla \tilde{f}^{b}(x_t)=\tilde{g}^{b,t}=\sum_{i\in S_t}\tilde{p}_i g_i^t$ for Section 4.2.2 to represent the ideal fair gradient.
>
> The above description has been added to our revised manuscript, indicated by a red line in Section 4.2.
>
> > It would be clearer if the authors could connect Eq 8 and Eq 10 to Eq 6, given that they share a similar form and intuition. For example, Eq 6 is not referenced in Section 4.2.2, thus making the purpose of Equation 10 unclear.
> >
>
> Thank you for your suggestions. We would like to connect (1) Eq 8 to Eq 6 and (2) Eq 10 to Eq 6.
>
> - Eq 6 is  $\frac{\partial L\left(x,p_i, \lambda_0, \lambda_1\right)}{\partial x} = (1-\alpha)\sum_{i=1}^m p_i \nabla F_i(x) + \alpha \nabla \tilde{f}(x)$.
>
>     According to model update and local SGD, $x_{t+1}= x_t-\eta\frac{\partial L\left(x,p_i, \lambda_0, \lambda_1\right)}{\partial x}$ and $\eta\Delta_t=x_{t+1}-x_{t}$, the final global model update can be written as:
>
>     $\Delta_t=-\frac{\partial L\left(x,p_i, \lambda_0, \lambda_1\right)}{\partial x}=-(1-\alpha)\sum_{i=1}^m p_i \nabla F_i(x) - \alpha \nabla \tilde{f}(x)$.
>
> - Eq 8 is  $\tilde{\Delta}^a_t = \frac{1}{|S_t|}\sum_{i\in S_t} \tilde{\Delta_{t}}^{a,i}= \frac{1}{|S_t|}\sum_{i\in S_t}(x_{t,1}^i - x_{t,0}^i)$ .
> Here, the ideal gradient $\nabla \tilde{f}(x)$ of Eq 6 is expressed by $\nabla \tilde{f}^a(x_t)=\tilde{\Delta}t^a$*,* and thus the final model update is expressed as \
> $\Delta_t = (1-\alpha) \sum{i\in S_t}p_i\Delta_t^i +\alpha \tilde{\Delta}^a_t$, \
> completing the connection to Eq 6.
> - Eq 10 is $h_{t,k}^i \gets (1-\alpha)g_{t,k}^i + \alpha \tilde{g}^{b,t}$.
> Here, the ideal gradient $\nabla \tilde{f}(x)$ of Eq 6 is expressed by $\nabla \tilde{f}^b(x_t)=\tilde{g}^{b,t}=\sum\nolimits_{i\in S_t}\tilde{p_i} g_i^t$, and thus the final model update is expressed as \
>  $\Delta_t=-(1-\alpha)\sum_{i\in S_t}p_i\eta_L\sum_{k=0}^{K-1}g_{t,k}^i-\alpha K \eta_L\sum_{i\in S_t}\tilde{p}ig_i^t$ \
> by $\Delta_t=\sum{i\in S_t}p_i\Delta_t^i,\Delta_t^i=x_{t,K}^i-x_{t,0}^i=-\eta_L\sum_{k=0}^{K-1} h_{t,k}^i$, finishing the connection to Eq 6.
>
> We have added these connections in the revised manuscript of Sec 4.2.1 and Sec 4.2.2, indicated by red text.
>
> > The motivation of maximum entropy for fairness may need further clarification. Though the authors discussed fairness in the introduction and mentioned the Shannon entropy in Section 4.1, the connection between entropy and fairness is not intuitively clear. More explicit reasoning or examples illustrating why higher entropy equates to greater fairness would be beneficial, especially in federated learning settings.
> >
>
> Thank you for your suggestion, we would like to elaborate on why higher entropy equates to greater fairness would be beneficial in FL setting and give example to illustrate it:
>
> - **Elaboration:** In FL, without constraints on the aggregation probability, maximum entropy results in a uniform distribution of aggregation probabilities, treating each client equally. However, when constraints are introduced, such as ensuring the aggregated loss is close to the ideal loss, maximum entropy is achieved by not making overly confident predictions while adhering to the constraints. Consequently, the aggregation becomes proportional to each client’s loss. The rationale is that a larger loss indicates poor model performance on a client, prompting the updated model to pay more attention to that client, resulting in equal performance across all clients.

---

> ### Author Response · Authors · 2023-11-16
> **Response to Reviewer yc23 (3/7)**
>
> - **Example:** We present a toy case in our original submission of Appendix 11.1, comparing the fairness behavior between FedAvg, q-FedAvg, and FedEBA+. The example illustrates that our aggregation probability, with maximum entropy, leads to better fairness than FedAvg and q-FedAvg[1]. We would like to restate it here to illustrate the benefit that higher entropy leads to greater fairness:
>
> In particular, we consider two clients participating in training, each with a regression model:
> $f_1(x_t) = 2(x-2)^2,
> f_2(x_t) = \frac{1}{2}(x+4)^2.$
> Corresponding,$\nabla f_1(x_t) = 4(x-2),\nabla f_2(x_t) = (x+4).$
>
> When the global model parameter $x_t = 0$ is sent to each client, each client will update the model by running gradient decent, here w.l.o.g, we consider one single-step gradient decent, and stepsize $\lambda = \frac{1}{4}$:
> $x_1^{t+1} = x_t - \lambda \nabla f_1(x_t) = 2,
> x_2^{t+1} = x_t - \lambda \nabla f_2(x_t) = -1.$
>
> Thus, for uniform aggregation: $x_{uniform}^{t+1} = \frac{1}{2}(x_1^{t+1}+x_2^{t+1})=\frac{1}{2}.$
>
> While for FedEBA+: $x_{EBA+}^{t+1}=\frac{e^{f_1(x_1^{t+1})}}{e^{f_1(x_1^{t+1})}+e^{f_2(x_2^{t+1})}}x_1^{t+1} +\frac{e^{f_2(x_2^{t+1})}}{e^{f_1(x_1^{t+1})}+e^{f_2(x_2^{t+1})}}x_2^{t+1} \approx -0.1$
>
> Therefore,
> $\mathbb{Var_{uniform}} = \frac{1}{2}\sum_{i=1}^2\left(f_i(x_{uniform}^{t+1})-\frac{1}{2}\sum_{i=1}^2(f_i(x_{uniform}^{t+1})\right)^2 = 2*(2.81)^2 ,
> \mathbb{Var_{EBA+}} = \frac{1}{2}\sum_{i=1}^2\left(f_i(x_{EBA+}^{t+1})-\frac{1}{2}\sum_{i=1}^2(f_i(x_{EBA+}^{t+1})\right)^2 = 2*(0.6)^2 .$
>
> Thus, we prove that FedEBA+ achieves a much smaller variance than uniform aggregation.
>
> Furthermore, for q-FedAvg, we consider $q=2$ which is also used in the proof of [1]:
> $\nabla x_1^t = L(x^t - x_1^{t+1})=-2, \nabla x_2^t = L(x^t - x_2^{t+1})=1 .$
>
> Thus, we have:
>
> $\Delta_1^t = f_1^q(x_t)\nabla x_1^t = 8*(-2) = -16,
> h_1^t = qf_1^{q-1}(x_t)\|\nabla x_1^t\|^2 + Lf_1^q(x_t) = 1\times 1\times 2^2+8 = 12.$
>
> $\Delta_2^t = f_2^q(x_t)\nabla x_2^t = 8*(1) = 8,
> h_2^t = qf_2^{q-1}(x_t)\|\nabla x_2^t\|^2 + Lf_2^q(x_t) = 1\times 1\times 1^2+8 = 9.$
>
> Finally, we can update the global parameter as:
> $x^{t+1}_{qfedavg} = x^t - \frac{\sum_i \Delta_i^t}{\sum_i h_i^t} \approx -0.4.$
>
> Then we can easily get:
> $\mathbb{Var_{qfedavg}} =\frac{1}{2}\sum_{i=1}^2\left(f_i(x_{qfedavg}^{t+1})-\frac{1}{2}\sum_{i=1}^m(f_i(x_{qfedavg}^{t+1})\right)^2 = 2*(2.52)^2$
>
> In conclusion, we prove that $\mathbb{Var_{EBA+}} \leq \mathbb{Var_{qfedavg} }\leq \mathbb{Var_{uniform}}.$
>
> In this case, the normalized performance’s entropy, after maxing the constrained entropy of aggregation probability, exhibits a relationship akin to variance (greater entropy corresponds to improved fairness).
>
> $Entropy(f(x_{EBA+}^{t+1}))=-\sum_{i=1}^2\frac{f_i(x_{EBA+}^{t+1})}{\sum_{j=1}^2f_j(x_{EBA+}^{t+1})}\log(\frac{f_j(x_{EBA+}^{t+1})}{\sum_{i=j}^2f_i(x_{EBA+}^{t+1})})\approx  0.996$, where $f_1(x_{EBA+}^{t+1})=2*(2.1)^2,f_2(x_{EBA+}^{t+1})=\frac{1}{2}*(3.9)^2.$
>
> $Entropy(f(x_{qfedavg}^{t+1}))=-\sum_{i=1}^2\frac{f_i(x_{qfedavg}^{t+1})}{\sum_{j=1}^2f_j(x_{qfedavg}^{t+1})}\log(\frac{f_j(x_{qfedavg}^{t+1})}{\sum_{i=j}^2f_i(x_{qfedavg}^{t+1})})\approx 0.942$, where$f_1(x_{qfedavg}^{t+1})=2*(2.4)^2,f_2(x_{qfedavg}^{t+1})=\frac{1}{2}*(3.6)^2.$
>
>   $Entropy(f(x_{uniform}^{t+1}))=-\sum_{i=1}^2\frac{f_i(x_{uniform}^{t+1})}{\sum_{j=1}^2f_j(x_{uniform}^{t+1})}\log(\frac{f_j(x_{uniform}^{t+1})}{\sum_{i=j}^2f_i(x_{uniform}^{t+1})})\approx 0.890$, where $f_1(x_{uniform}^{t+1})=2*(1.5)^2,f_2(x_{uniform}^{t+1})=\frac{1}{2}*(4.5)^2$.
>
> Therefore,
>  $Entropy(f(x_{EBA+}^{t+1}))\geq  Entropy(f(x_{qfedavg}^{t+1}))\geq Entropy(f(x_{uniform}^{t+1}))$, and $\mathbb{Var_{EBA+}} \leq \mathbb{Var_{qfedavg}} \leq \mathbb{Var_{uniform}} .$
>
> We have mentioned this in Section 4.1 of the revised manuscript.
>
> [1]Li T, Sanjabi M, Beirami A, et al. Fair resource allocation in federated learning[J]. ICLR 2020.
>
>
> > From equation (2), the term “ideal loss” $\tilde{f}(x)$ is introduced without a clear definition or expression, leading to confusion that is not resolved until later sections (4.2.1 and 4.2.2). It would be beneficial if the authors could provide more discussion or examples for “ideal loss” here.
> >
>
> Thank you for your suggestion. We would like to add an explanation of the ideal loss:
>
> The ideal loss $\tilde{f}(x)$ serves as the target or objective for the aggregated losses. Its specific expression depends on the goal of the FL system. In Eq (2), as a robust constraint, $\tilde{f}(x)$ aims to ensure that the aggregated clients' losses are as close to the desired target loss as possible.
>
> In this paper, for example, when the ideal loss acts as the ideal fair loss, the gradient of the ideal loss should be a reweighted aggregation of clients’ unbiased local updates, $\nabla \tilde{f_{fair}}(x)=\sum_{i\in S_t}p_i \nabla F_i(x)$, where $\nabla F_i(x)$ is the unbiased local update of the model $x$.
>
> Motivated by your comments, we have added this discussion in our revised manuscript.

---

> ### Author Response · Authors · 2023-11-16
> **Response to Reviewer yc23 (4/7)**
>
> ### Clarification
>
> > The fairness of FedEBA+ is partly achieved based on the assumption that the FedSGD global gradient is an ideal fair gradient. Specifically, according to the end of Section 4.2, the authors view a FedSGD global gradient as the ideal global gradient and the ideal fair gradient.
> >
>
> We would like to clarify that we employ the one-step gradient to obtain the ideal local gradient.
>
> For the ideal global gradient, we use the average of the ideal local update as the ideal global gradient, which aligns with the FedSGD global gradient. However, it's crucial to note that even for the ideal global gradient, the FedSGD gradient is only utilized to correct the model update rather than being used as the updated model itself.
>
> In contrast, for the ideal fair gradient, it differs from the FedSGD gradient. The fair gradient is the reweighted average of the ideal local update based on our fair aggregation probability $\sum_{i\in S_t}\tilde{p_i }\nabla F_i(x_t)$, instead of a direct average $\frac{1}{|S_t|}\sum_{i\in S_t}\nabla F_i(x_t)$.
>
> > Clarity is needed on why this assumption is made and whether the FedSGD gradient truly encapsulates ideal fairness.
> >
>
> Clarification: **It is not FedSGD that encapsulates the ideal fairness, but the reweighted aggregated one-step local updates represent the ideal fairness.**
>
> - [Why reweight] Based on the local loss with the global model on each client ($F_i(x_t)$), the server knows the global model’s performance on each client, allowing it to adjust the attention it pays to different clients: assigning higher weights to clients with larger loss (poor performance) for fairness. According to the analysis of entropy, to make the aggregated loss to be fairer, the aggregation should be reweighted based on Eq 3.
> - [Why one-step update] The choice of using one-step local updates stems from the observation that multiple local updates can introduce local bias in FL[1,2,3]. Therefore, using the one-step gradient can better reflect the true global model's update on the clients without local bias. Combining this with the fair reweight aggregation strategy ensures that the model represents a more accurate fair gradient.
>
> [1]Karimireddy S P, Kale S, Mohri M, et al. Scaffold: Stochastic controlled averaging for federated learning[C]//International conference on machine learning. PMLR, 2020: 5132-5143.
>
> [2]Wang J, Liu Q, Liang H, et al. Tackling the objective inconsistency problem in heterogeneous federated optimization[J]. Advances in neural information processing systems, 2020, 33: 7611-7623.
>
> [3]Mendieta M, Yang T, Wang P, et al. Local learning matters: Rethinking data heterogeneity in federated learning[C]//Proceedings of the IEEE/CVF Conference on Computer Vision and Pattern Recognition. 2022: 8397-8406.
>
> > Also, it would be helpful to consider FedSGD as one fairness baseline in the experiments, to demonstrate the effect of another component in FedEBA+, maximum entropy aggregation, for fairness.
> >
>
> Thank you for your suggestion.
>
> - We present the results of FedSGD in the table below, demonstrating that FedSGD [1] performs even worse than FedAvg. This aligns with findings in other works [2], and the suboptimal performance of FedSGD can be attributed to the inefficiency introduced by the one-step local update. While the one-step update reduces local bias in a non-iid setting, other FL algorithms, including FedEBA+, with multiple updates achieve higher accuracy in significantly fewer communication rounds than FedSGD. Our approach leverages the advantages of both multiple and one-step updates by using FedSGD gradient solely for alignment in model updates.
>
> |  | CIFAR-10 |  |  |  | FashionMNIST |  |  |  |
> | --- | --- | --- | --- | --- | --- | --- | --- | --- |
> | Algorithm | Global Acc. | Var. | Worst 5% | Best 5% | Global Acc. | Var. | Worst 5% | Best 5% |
> | FedAvg | 67.79±0.35 | 103.83±10.46 | 45.00±2.83 | 85.13±0.82 | 86.49 ± 0.09 | 62.44±4.55 | 71.27±1.14 | 95.84± 0.35 |
> | FedSGD | 67.48±0.37 | 95.79±4.03 | 48.70±0.9 | 84.20±0.40 | 83.79±0.28 | 81.72±0.26 | 61.19±0.30 | 96.60±0.20 |
> | FedEBA+ | 72.75±0.25  | 68.71±4.39  | 55.80±1.28  | 86.93±0.52 | 87.50±0.19 | 43.41±4.34 | 72.07±1.47 | 95.91±0.19 |
>
> - Additionally, we want to emphasize that we have conducted an ablation study for each step of FedEBA+, demonstrating the effectiveness of the maximal entropy-based aggregation. The detailed results are provided in the restated table here:
>     - FedEBA corresponds to the algorithm that only uses entropy-based aggregation compared with FedAvg; FedEBA+ indicates the algorithm uses both entropy-based aggregation and alignment update
>     - Comparing FedEBA with FedAvg, FedEBA improves accuracy by about 1.6 and fairness by about 14.3 on CIFAR-10, highlighting the proposed entropy-based aggregation approach’s effectiveness. Similar observations can be found in FashionMNIST, CIFAR-100, and Tiny-ImageNet.

---

> ### Author Response · Authors · 2023-11-16
> **Response to Reviewer yc23 (5/7)**
>
> |  | CIFAR-10 |  |  |  | FashionMNIST |  |  |  |
> | --- | --- | --- | --- | --- | --- | --- | --- | --- |
> | Algorithm | Global Acc. | Var. | Worst 5% | Best 5% | Global Acc. | Var. | Worst 5% | Best 5% |
> | FedAvg | 67.79±0.35 | 103.83±10.46 | 45.00±2.83 | 85.13±0.82 | 86.49 ± 0.09 | 62.44±4.55 | 71.27±1.14 | 95.84± 0.35 |
> | FedEBA | 69.38±0.52   | 89.49±10.95 | 50.40±1.72  | 86.07±0.90 | 86.70±0.11 | 50.27±5.60 | 71.13±0.69 | 95.47±0.27 |
> | FedEBA+ | 72.75±0.25  | 68.71±4.39  | 55.80±1.28  | 86.93±0.52 | 87.50±0.19 | 43.41±4.34 | 72.07±1.47 | 95.91±0.19 |
> |  | **CIFAR-100** |  |  |  | **Tiny-ImageNet** |  |  |  |
> | Algorithm | Global Acc. | Var. | Worst 5% | Best 5% | Global Acc. | Var. | Worst 5% | Best 5% |
> | FedAvg | 30.94±0.04 | 17.24±0.08 | 0.20±0.00 | 65.90±1.48 | 61.99±0.17 | 19.62±1.12 | 53.60±0.06 | 71.18±0.13 |
> | FedEBA | 32.38±0.13 | 17.09±0.06 | 0.75±0.22 | 66.40±0.47 | 63.34±0.25 | 15.29±1.36 | 54.17±0.04 | 70.98±0.10 |
> | FedEBA+ | 33.39±0.22 | 16.92±0.04 | 0.95±0.15 | 68.51±0.21 | 64.05±0.09 | 14.91±1.85 | 54.32±0.09 | 71.27±0.04 |
>
> We have included the new experimental results in Table 11 of the revised manuscript.
>
> [1]McMahan H B, Moore E, Ramage D, et al. Federated learning of deep networks using model averaging[J]. CoRR, 2016.
>
> [2]McMahan B, Moore E, Ramage D, et al. Communication-efficient learning of deep networks from decentralized data[C]//Artificial intelligence and statistics. PMLR, 2017
>
> > The following statement is not clear: “The specific challenge lies in finding a way to introduce entropy into the FL training process while incorporating FL’s uniform performance as a constraint within the framework.” Isn’t maximum entropy encouraging uniform performance? (Intuitively, the outcomes with uniform probability give the highest entropy, i.e., uncertainty, based on the definition of entropy) Does maximum entropy go against the goal of uniform performance? The reason why these concepts are treated distinctly in this statement requires clarification.
> >
>
> Sorry for the confusion by our previous expression. Thank you for pointing out the inaccurate statement, we have revised it into  “The challenge lies in modeling FL aggregation as entropy and incorporating that using aggregation method to make the performance distribution more uniform.”
>
> Indeed, maximizing the entropy of performance aligns with the goal of achieving uniform performance. However, in our paper, the entropy is not over performance but aggregation probability $\mathbb{H}(p) := -\sum_{i=1}^m p_i \log(p_i)$. Simply maximizing the entropy $\mathbb{H}(p)$ corresponds to making each client have a uniform aggregation distribution, this is FedAvg.
>
> Recall that our aim is to make the distribution of performance uniform, thus it’s necessary to link the aggregation probability and model performance, and link them as a constraint to make the performance uniform. Thus, we introduce the ideal fair loss $\tilde{f}(x)$ as a constraint $\sum_{i}p_iF_i=\tilde{f}(x)$ of the constrained entropy as shown in Eq (2), this forces the loss of aggregation to be close to the loss under truly fair model performance to ensure that the aggregated model (global model) obtained with such an aggregation probability is fair. Consequently, the aggregation probability $p_i$ becomes proportional to each client’s loss $F_i(x)$. The rationale is that a larger loss indicates poor model performance on a client, prompting the updated model to pay more attention to that client, resulting in equal performance across all clients.
>
> The constrained entropy also yields the highest entropy after satisfying the condition, ensuring that the aggregation does not take other confidential conditions into consideration and giving the highest entropy.
>
> ### Theory:
>
> > The claimed convergence rate is not appropriate because there is a non-vanishing constant error term, which means that the gradient of the algorithm will never approach zero. The convergence rate is O(1) instead. This finding is inconsistent with previous studies, such as FedAvg convergence under the non-iid setting [1], which does not have the constant error term.
> >
>
> Thank you for your feedback. We'd like to clarify:
>
> - **No-vanishing term is common in biased sampling/aggregation FL:** The inclusion of a no-vanishing term is a common requirement in biased sampling/aggregation methods, as demonstrated in various works[1,2,3,4,5]. The proposed entropy-based aggregation is a biased method, aiming to improve fairness, thus there is a no-vanishing term. Besides FL, this phenomenon is also commonly observed in optimization[6,7].
> - **Biased sampling/aggregation has advantages in FL:** Previous research has shown that biased sampling can expedite convergence rates and enhance accuracy in FL [1], as well as mitigate local bias issues [2]. In our work, we employ biased aggregation methods to enhance the fairness of FL.

---

> ### Author Response · Authors · 2023-11-16
> **Response to Reviewer yc23 (6/7)**
>
> - **Our convergence can degrade to FedAvg:** Our approach can revert to the state-of-the-art FedAvg's convergence rate, particularly with partial client participation [8], by setting the aggregation probability to a uniform distribution $f(x_t)=\sum_{i\in S_t}p_iF_i(x_t)=\sum_{i\in S_t}\frac{1}{|S_t|}F_i(x_t)$, causing the no-vanishing terms to vanish.
>
> [1]Cho Y J, Wang J, Joshi G. Towards understanding biased client selection in federated learning[C] AISTATS 2022.
> [2]Wang J, Liu Q, Liang H, et al. Tackling the objective inconsistency problem in heterogeneous federated optimization[J]. NeurIPS 2021.
>
> [3]Balakrishnan R, Li T, Zhou T, et al. Diverse client selection for federated learning via submodular maximization[C] ICLR. 2022.
>
> [4]Cho Y J, Wang J, Joshi G. Client selection in federated learning: Convergence analysis and power-of-choice selection strategies[J]. arXiv preprint arXiv:2010.01243, 2020.
>
> [5]Guo Y, Lin T, Tang X. Towards federated learning on time-evolving heterogeneous data[J]. ICLR federated learning workshop 2021.
>
> [6]Generalized-Smooth Nonconvex Optimization is As Efficient As Smooth Nonconvex Optimization, ICML 2023
>
> [7]Loizou N, Vaswani S, Laradji I H, et al. Stochastic polyak step-size for sgd: An adaptive learning rate for fast convergence[C] AISTATS 2021.
>
> [8]Yang H, Fang M, Liu J. Achieving linear speedup with partial worker participation in non-iid federated learning[J]. ICLR 2021.
>
> > The Theorem 5.3 claims that FedEBA+ achieves a smaller variance than FedAvg. However, the effect of FL data distributions is not reflected in Theorem 5.3. It would be clearer if the authors could justify how non-iid degree will affect the fairness results.
> >
>
> Thank you for your suggestions. We'd like to justify the non-iid degree in Theorem 5.3:
>
> As discussed in Appendix 11.2, specifically in the *Test Loss* paragraph, “the dataset on client $i$ is denoted as $\left(\boldsymbol{\Xi}_i, \mathbf{y}_i\right)$, where $\boldsymbol{\Xi}_i$ is fixed, and $\mathbf{y}_i$ follows a Gaussian distribution $\mathcal{N}\left(\boldsymbol{\Xi}_i \mathbf{w}_i, \sigma_2^2 \boldsymbol{I}_n\right)$. The heterogeneity across clients arises solely from the variability in $\mathbf{w}_i$“, where $\textbf{w}_i$ is the true parameter on client $i$.
>
> In Theorem 5.3, we have demonstrated that $V^{\text{Avg}}=\operatorname{var}\left(f_i^{t e}\left(\mathbf{w}^{A v g}\right)\right)=\frac{b^2}{4} \operatorname{var}\left(\left\|\overline{\mathbf{w}}-\mathbf{w}_i\right\|_2^2\right)$ and $V^{\text{EBA+}}=\operatorname{var}\left(f_i^{t e}\left(\mathbf{w}^{EBA+}\right)\right)=\frac{b^2}{4} \operatorname{var}\left(\left\|\tilde{\mathbf{w}}-\mathbf{w}_i\right\|_2^2\right)$. This implies that a larger non-iid degree, i.e., greater data heterogeneity across clients, corresponds to a larger heterogeneity of $\mathbf{w}_i$. Consequently, larger heterogeneity of $\textbf{w}_i$ results in larger variance for both FedAvg and FedEBA+.
>
> Moreover, Theorem 5.3 establishes that under the same heterogeneity degree, FedEBA+ consistently achieves smaller variance than FedAvg.
>
> Motivated by your comments, we have mentioned this explicitly in Theorem 5.3 of the revised manuscript.
>
> > Moreover, could the authors provide definitions for T() and A() in the variance analysis?
> >
>
> Sorry for the confusion by the notations. We would like to clarify that
>
> - $T(\xi_{i,k})$ is the generalized regression coefficient, representing a generalized expression of the variable $\xi_{i,k}$.
> - $A(\xi_{i,k})$ denotes the noise term distributed by $N(\mu_{\xi},\sigma)$.
>
> These clarifications have been incorporated into the notations section before Theorem 5.3 in our revised manuscript.
>
> ### Experiments:
>
> > The algorithms introduce a few additional hyperparameters: temperature , angle threshold , and . From Figure 4, it appears that the performance of the algorithm is quite sensitive to the hyperparameters, and different datasets have different optimal hyperparameters. It indicates a need for extensive hyperparameter tuning across different datasets. This requirement potentially reduces the algorithm's practicality compared to more straightforward federated learning algorithms like FedAvg.
> >
>
> Thank you for your comments. We would like to clarify:
>
> - **No extensive hyperparameter tuning is needed:** Our algorithm consistently **achieves the best performance across all four datasets (FashionMNIST, CIFAR-10, CIFAR-100, and Tiny-ImageNet) under the same set of parameters** (temperature $\tau=0.1$, $\alpha=0.9$), as reported in Table 1 and Table 2 of our original submission.
> - **Fine-tuning hyperparameters can further improve algorithms, but common choices suffice:** Figure 4 demonstrates that different hyperparameter choices can enhance performance, even surpassing the best result in Table 1 and Table 2. However, no specific finetuned hyperparameter choice outperforms baselines, as indicated in the table below.

---

> ### Author Response · Authors · 2023-11-16
> **Response to Reviewer yc23 (7/7)**
>
> |  | FashionMNIST |  | CIFAR-10 |  |
> | --- | --- | --- | --- | --- |
> | Algorithm | Global Acc. | Var. | Global Acc. | Var. |
> | FedAvg | 86.49 ± 0.09 | 62.44±4.55 | 67.79±0.35  | 103.83±10.46 |
> | q-FFL$_{q=0.001}$ | 87.05± 0.25   | 66.67± 1.39  | 68.53± 0.18  | 97.42± 0.79 |
> | q-FFL$_{q=0.5}$ | 86.57± 0.19  | 54.91± 2.82 | 68.76± 0.22  | 97.81± 2.18 |
> | q-FFL$_{q=10.0}$ | 77.29± 0.20  | 47.20± 0.82 | 40.78± 0.06  | 85.93± 1.48 |
> | PropFair$_{M=0.2, thres=0.2}$ | 85.51±0.28  | 75.27±5.38  | 65.79±0.53  | 79.67±5.71 |
> | PropFair$_{M=5.0, thres=0.2}$ | 84.59±1.01 | 85.31±8.62 | 66.91±1.43  | 78.90±6.48 |
> | FedFV$_{\alpha=0.1, \tau{fv}=10} $ | 86.98±0.45  | 56.63±1.85  | 71.10±0.44  | 86.50±7.36 |
> | FedFV$_{\alpha=0.2, \tau{fv}=0}$ | 86.42±0.38   | 52.41±5.94 | 68.89±0.15 |  82.99±3.10 |
> | FedEBA+$_{\alpha=0.1, \tau=0.1}$ | 86.98±0.10   | 53.26±1.00 | 71.82±0.54 | 83.18±3.44 |
> | FedEBA+$_{\alpha=0.3, \tau=0.1}$ | 87.01±0.06 | 51.878±1.56 | 71.79±0.35 | 77.74±6.54 |
> | FedEBA+$_{\alpha=0.5, \tau=0.1}$ | 87.21±0.06 | 40.02±1.58 | 72.44±0.33 | 69.71±3.09 |
> | FedEBA+$_{\alpha=0.7, \tau=0.1}$ | 87.23±0.07 | 40.456±1.45 | 72.36±0.15 | 77.61±6.31 |
> | FedEBA+$_{\alpha=0.9, \tau=0.1}$ | 87.50±0.19 | 43.41±4.34 | 72.57±0.33 | 69.88±5.86 |
> | FedEBA+$_{\alpha=0.9, \tau=0.05}$ | 87.42±0.10 | 50.46±2.37 | 72.19±0.16 | 71.79±6.37 |
> | FedEBA+$_{\alpha=0.9, \tau=0.5}$ | 87.26±0.06 | 52.65±4.03 | 71.89±0.39 | 75.29±9.01 |
> | FedEBA+$_{\alpha=0.9, \tau=1.0}$ | 87.14±0.07 | 52.71±1.45 | 72.30±0.26 | 73.79±9.11 |
> | FedEBA+$_{\alpha=0.9, \tau=5.0}$ | 87.10±0.14 | 55.52±2.15 | 72.43±0.11 | 82.08±8.31 |
>
> - **Angle threshold** $\theta$ **requires no tuning as it is a choice, not a fine-tuned hyperparameter:** For $\theta$, a smaller value corresponds to better performance, as shown in the ablation study. The choice of $\theta$ depends solely on the user's needs based on communication capabilities and does not necessitate fine-tuning. Even with a large $\theta$ and no additional communication, FedEBA+ outperforms baselines, notably FedAvg. The ablation study for $\theta$ is provided in the reply to the next answer.
> - **Hyperparameter finetuning is common in fair FL algorithms:** Table 3 in our submission illustrates that all fair algorithms require some hyperparameter tuning. These algorithms aim to improve fairness, not just the basic communication function between the server and clients. Unlike FedAvg, which requires no hyperparameter tuning, fair algorithms can significantly enhance fairness and global accuracy through fine-tuning.
>
> The results for different parameter choices of ($\tau$ and $\alpha$) have been added to Table 12 in the revised manuscript.
>
> > The influence of the angle threshold  on FedEBA+'s performance is not demonstrated. It would be beneficial to provide an ablation study on .
> >
>
> Thank you for your suggestion.
>
> - We would like to clarify that due to page limits, we provided the ablation study of $\theta$ in Appendix 14 (Table 5 and Figures 9-12) on FashionMNIST in our original submission.
> - Furthermore, we have presented new ablation results for $\theta$ on CIFAR-10. The table below illustrates that:
>     - With sparse additional communication, like $\theta=40^\circ$, FedEBA+ achieves significantly better performance than baselines (Accuracy improves by 2.5, fairness improves by 24.3 compared to FedAvg on CIFAR-10).
>     - Even with no additional communication ($\theta=90^\circ$), FedEBA+ promises the best fairness and comparable accuracy on both datasets.
>
> |  | FashionMNIST (MLP) |  | Additional communication as a percentage of total communications (7.8MB/round) | CIFAR-10 (CNN) |  | Additional communication as a percentage of total communications (30.4MB/round) |
> | --- | --- | --- | --- | --- | --- | --- |
> | Algorithm | Global Acc. | Var. |  | Global Acc. | Var. |  |
> | FedAvg | 86.49 ± 0.09 | 62.44±4.55 | - | 67.79±0.35  | 103.83±10.46 | - |
> | q-FFL | 87.05± 0.25   | 66.67± 1.39  | - | 68.53± 0.18  | 97.42± 0.79 | - |
> | FedMGDA+ | 84.64±0.25  | 57.89±6.21 | - | 67.16±0.33  | 97.33±1.68 | - |
> | AFL | 85.14±0.18  | 57.39±6.13 | - | 66.21±1.21  | 79.75±1.25 | - |
> | PropFair | 85.51±0.28  | 75.27±5.38 | - | 65.79±0.53  | 79.67±5.71 | - |
> | TERM | 84.31±0.38  | 73.46±2.06 | - | 65.41±0.37  | 91.99±2.69 | - |
> | FedFV | 86.98±0.45  | 56.63±1.85  | - | 71.10±0.44  | 86.50±7.36 | - |
> | FedEBA+$_{\tau=0.1, \alpha=0.9}$ |  |  |  |  |  |  |
> |     $\theta=0°$ | 87.50±0.19 | 43.41±4.34 | 50.0% | 72.75±0.25 | 68.71±4.39 | 50.0% |
> |     $\theta=15°$ | 87.14±0.12 | 43.95±5.12 | 48.6% | 71.92±0.33 | 75.95±4.72 | 26.2% |
> |     $\theta=30°$ | 86.96±0.06 | 46.82±1.21 | 37.7% | 70.91±0.46 | 70.97±4.88 | 12.7% |
> |     $\theta=45°$ | 86.94±0.26 | 46.63±4.38 | 4.2% | 70.24±0.08 | 79.51±2.88 | 0.2% |
> |     $\theta=90°$ | 86.78±0.47 | 48.91±3.62 | 0 | 70.14±0.27 | 79.43±1.45 | 0 |
>
> We add the results of CIFAR-10 to Table 5 of the original paper as a richer Table 5 in the revised manuscript.

---

> ### Author Response · Authors · 2023-11-20
> **Looking forward to your feedback**
>
> Dear Reviewer yc23,
>
> I hope this message finds you well.
>
> We would like to first express our profound gratitude for the time and expertise you've dedicated to the assessment of our submission. We appreciate your insights and have found your comments to be extremely beneficial in refining our work.
>
> Regarding your concerns, we hope that our rebuttal was able to shed light on them. To summarize briefly, we have clearly described the algorithm by adding explanations, improved the writing based on your suggestions, clarified the misunderstanding regarding relationships between FedSGD and FedEBA+, and between maximum entropy and performance uniformity. We have also explained and justified the soundness of the theory and provided additional experimental results of hyperparameters.
>
> Your constructive feedback has recommended that we refine our expression and pinpoint the limitations of our work. We deeply appreciate your expertise and we have refined our work based on your feedback. Thank you once again for your time and consideration.
>
> Warmest regards,
>
> Authors

---

> ### Comment · Reviewer_yc23 · 2023-11-22
>
> Thank you for the responses. After reading your responses, I have some comments on it:
>
> > 1: High communication costs remain.
>
> Thanks for fixing the algorithm. However, the algorithm still requires two communication exchanges between the server and clients in each round $t$, as suggested by  Line 6 and Line 11. This seems inherent to the proposed method since each client needs the global gradient to regularize local training. In contrast, other fairness-focused algorithms like q-FFL and AFL manage with a single communication per round.
>
> > 2: The notations are not consistent in Algorithm 1.
>
> $\tilde{g}^t$ in line 9 is not defined.  How does this differ from the $\tilde{g}^{b,t}$ in line 6?
>
> > 3:  Eq 3 is still wrong.
>
> In your response, step 1 and step 2 are not consistent: In step 1, the same $x^i_{t,K}$ is used for both denominator and numerator, aligning with what Equation 3 suggests. However, in the step 2, take $p_1$ as an example, a different approach is used: $x^1_{t,K}$ for numerator, but different $x^1_{t,K}, x^2_{t,K}, … $ for denominator. This inconsistency needs to be addressed for a clearer understanding.
>
> > 4: Interchangeability of $\tilde{f}(x)$, $\tilde{\Delta}_t$,  and $\tilde{g}^t$.
>
> Thank the author for providing the justifications for introducing the unweighted aggregation (i.e., ideal global gradient) and reweighted aggregation for FedSGD update  (ideal fair gradient). However, considering the heterogeneity of local data, it seems that weighted aggregation could also enhance global model performance and should be used as the ideal global gradient. The original FedSGD algorithm [1] uses a weighted average based on the number of local samples. The design choice for unweighted aggregation needs further clarification.
>
> > 5: Theory.
>
> Thank you for clarifying the issue of biased sampling. However, the paper's claim about a convergence rate of O(1/T) as mentioned above Remark 5.2, appears problematic due to the constant error term O(1). This suggests that the accumulated gradient will not converge to zero.
>
>
> The authors well answer other questions. I would like to keep my original score.
>
>
> [1] Communication-Efficient Learning of Deep Networks from Decentralized Data, AISTATS 2017.

---

> > ### Author Response · Authors · 2023-11-23
> > **To Reviewer yc23 (1/2)**
> >
> > Dear reviewer yc23,
> >
> > Thank you for your response.
> >
> > > 1: High communication costs remain. Thanks for fixing the algorithm. However, the algorithm still requires two communication exchanges between the server and clients in each round t, as suggested by Line 6 and Line 11.
> > >
> >
> > Thank you for your valuable comments. We would like to clarify that the execution of Line 6 and Line 11 is contingent upon the simultaneous satisfaction of two conditions: (1) the alignment process and (2) the arccos being greater than the specified threshold, denoted as $\theta$. However, our ablation study indicates that the loop involving Line 6 and Line 11 does not consistently occur, thereby ensuring that the communication cost does not double in comparison to FedAvg.
> >
> > Moreover, as illustrated in the accompanying table:
> >
> > 1. Setting the threshold sufficiently high ($\theta=90^\circ$) ensures that the algorithm systematically bypasses Line 6 and Line 11. This strategic choice maintains the algorithm's superiority over alternatives such as FedAvg, q-FFL, and AFL. Thus, with an equivalent communication cost to FedAvg, FedEBA+ emerges as the preferred choice among fair algorithms.
> > 2. Given a manageable communication capacity (Line 3 in Algorithm 1), FedEBA+ demonstrates an enhanced performance in terms of accuracy and variance with only a marginal increase in communication cost, as exemplified by $\theta=45^\circ$.
> >
> > |  | FedAvg | q-FFL | AFL | FedEBA+$\|_{\theta=90^\circ}$ | FedEBA+$\|_{\theta=45^\circ}$ |
> > | --- | --- | --- | --- | --- | --- |
> > | Parameter transmitted in each round (single model $\times$ 10 clients) on FashionMNIST (MLP) | 7.8M | 7.8M | 7.8M | 7.8M | 8.1M |
> > | Accuracy on FashionMNIST | 86.49 ± 0.09 | 87.05± 0.25   | 85.14±0.18  | 86.78±0.47 | 86.94±0.26 |
> > | Variance (fairness) on FashionMNIST | 62.44±4.55 | 66.67± 1.39  | 57.39±6.13 | 48.91±3.62 | 46.63±4.38 |
> >
> > |  | FedAvg | q-FFL | AFL | FedEBA+$\|_{\theta=90^\circ}$ | FedEBA+$\|_{\theta=45^\circ}$ |
> > | --- | --- | --- | --- | --- | --- |
> > | Parameter transmitted in each round (single model $\times$ 10 clients)  on CIFAR-10 (CNN) | 30.4M | 30.4M | 30.4M | 30.4M | 30.5M |
> > | Accuracy on CIFAR-10 | 67.79±0.35  | 68.53± 0.18  | 66.21±1.21  | 70.14±0.27 | 70.24±0.08 |
> > | Variance (fairness) on CIFAR-10 | 103.83±10.46 | 97.42± 0.79 | 79.75±1.25 | 79.43±1.45 | 79.51±2.88 |
> >
> > > 2: The notations are not consistent in Algorithm 1.
> > >
> >
> > Thank you for your suggestion.
> >
> > We have addressed the typo by replacing "$\tilde{g}^t$" with "$\tilde{g}^{b,t}$" in the revised manuscript.
> >
> > > 3: Eq 3 is still wrong. In your response, step 1 and step 2 are not consistent.
> > >
> >
> > We would like to clarify that in our response, steps 1 and 2 are consistent. In both steps, the denominator is always $\exp({F_1(x_{t,K}^1)/\tau})+\dots+\exp({F_N(x_{t,K}^N)/\tau})$, which is the sum of the exponentials of different clients' losses.
> >
> > The underlying concept of Equation 3 is that the sampling probability is proportional to each client's loss, and different clients have different losses. Although in our expression this is represented by $F_i(x)$, where different values of $i$ distinguish the losses, it should be noted that as long as $F_i$ is normalized, the equation remains correct. Therefore, in Federated Learning (FL), it is common to use $F_i(x_i)$ to represent different losses for different clients.
> >
> > As we clarify, $\sum_i p_i = p_1+\cdots+p_n = \frac{\exp({F_1(x_{t,K}^1)/\tau})}{\sum_j\exp({F_j(x_{t,K}^j)/\tau})}+\cdots+\frac{\exp({F_n(x_{t,K}^n)/\tau})}{\sum_j\exp({F_j(x_{t,K}^j)/\tau})}=1$.
> >
> > Thus, the equation consistently sums to 1, ensuring its correctness.
> >
> > This type of probability expression is commonly found in client sampling of FL [1,2] and energy-based models [3,4].
> >
> > [1]Cho Y J, Wang J, Joshi G. Towards understanding biased client selection in federated learning[C]//International Conference on Artificial Intelligence and Statistics. PMLR, 2022: 10351-10375.
> >
> > [2]Chen W, Horváth S, Richtárik P. Optimal Client Sampling for Federated Learning[J]. Transactions on Machine Learning Research, 2022.
> >
> > [3]Ou Z, Xu T, Su Q, et al. Learning Neural Set Functions Under the Optimal Subset Oracle[J]. Advances in Neural Information Processing Systems, 2022, 35: 35021-35034.
> >
> > [4]LeCun Y, Chopra S, Hadsell R, et al. A tutorial on energy-based learning[J]. Predicting structured data, 2006, 1(0).

---

> > ### Author Response · Authors · 2023-11-23
> > **To Reviewer yc23 (2/2)**
> >
> > > 4. Interchangeablity of $\nabla \tilde{f},\tilde{\Delta},\tilde{g}^t.$ Considering the heterogeneity of local data, it seems that weighted aggregation could also enhance global model performance and should be used as the ideal global gradient.
> > >
> >
> > We would like to clarify that we employ FedSGD as the unweighted gradient to approximate the ideal global gradient in both our paper and response, rather than the ideal fair gradient.
> >
> > The weighted gradient is defined as $\nabla \tilde{f}_{fair}=p_i \nabla F_i(x)$, where $p_i$ represents the entropy-based aggregation probability (Eq 3).
> >
> > Taking into account heterogeneity, the expression for FedSGD is $\sum_i \frac{n_i}{n}\nabla F_i(x)$, with the aggregation probability differing from that in Eq 3. Nevertheless, FedSGD is still used as the ideal global gradient, replacing $\frac{1}{|S_t|}$ with $\frac{n_i}{n}$. It's important to note that we consistently use $\frac{n_i}{n}$ in all Federated Learning (FL) algorithms when dealing with varying amounts of data.
> >
> > We appreciate your suggestion, and we have incorporated this clarification into Section 4.2.1 of our revised version.
> >
> > > 5: Theory. The paper's claim about a convergence rate of O(1/T) as mentioned above Remark 5.2, appears problematic due to the constant error term O(1). This suggests that the accumulated gradient will not converge to zero.
> > >
> >
> > Thank you for your suggestion. We would like to clarify the statement above Remark 5.2 is: "FedEBA+ will converge to a nearby neighborhood of optimality at a rate of $\mathcal{O}\left(\frac{1}{T}+\frac{1}{nKT} \right)$, the same order as that of the SOTA FedAvg."
> >
> > This type of statement is commonly used in convergence rates, indicating that the algorithm converges to a neighborhood [1,2,3,4]. It signifies the speed of the algorithm converging to a neighborhood of optimality, rather than the optimality itself. Therefore, it does not imply the speed of the gradient converging to zero.
> >
> > [1]Cho Y J, Wang J, Joshi G. Towards understanding biased client selection in federated learning[C]//International Conference on Artificial Intelligence and Statistics. PMLR, 2022: 10351-10375.
> >
> > [2]Wang J, Liu Q, Liang H, et al. Tackling the objective inconsistency problem in heterogeneous federated optimization[J]. Advances in neural information processing systems, 2020, 33: 7611-7623.
> >
> > [3]Balakrishnan R, Li T, Zhou T, et al. Diverse client selection for federated learning via submodular maximization[C]//International Conference on Learning Representations. 2022.
> >
> > [4]Cho Y J, Wang J, Joshi G. Client selection in federated learning: Convergence analysis and power-of-choice selection strategies[J]. arXiv preprint arXiv:2010.01243, 2020.

---

> ### Comment · Reviewer_yc23 · 2023-11-23
>
> Q3: Eq 3 defines $p_i = \frac{\exp (F_i(x)/\tau)}{\sum_j \exp (F_j (x)/\tau)}$.  The denominator and numerator use the *same* $x$, which I believe is incorrect. Your response in step 3 presents $p_i = \frac{\exp (F_i(x)/\tau)}{\sum_j \exp (F_j (x^j)/\tau)}$, differing from Equation 3. You are using different $x$ for the denominator and numerator.
>
> Q4:  Why do the authors use weighted average $n_i/n$ for all FL algorithms, but use unweighted average $1/|S_i|$ for the proposed ideal global gradient? In the current design, "the ideal global gradient" does not take the heterogeneity into account, which is not ideal.
>
> Q5: [1] proves that  $F (w^T) - F (w*) \leq O(1/T) $ + constant error. This indicates that the algorithm in [1] converges to a neighborhood of optimality (e.g., the achieved loss is not optimal).  [3] proves a similar bound but for $w^T - w^*$.
> However, the authors try to bound the gradient, i.e.,  $\nabla F( w^T) \leq O(1/T) $ + constant error.  The constant error term for gradient indicates that the algorithm does not converge to a stationary point.  Theorem 1 in [2] and  [A] actually do not involve the constant error term.
>
> [A] Adaptive Federated Optimization, ICLR 2021

---

> > ### Author Response · Authors · 2023-11-23
> > **To Reviewer yc23**
> >
> > Dear Reviewer yc23,
> >
> > > Q3. Eq 3 defines $p_i = \frac{\exp (F_i(x)/\tau)}{\sum_j \exp (F_j (x)/\tau)}$. The denominator and numerator use the *same* $x$, which I believe is incorrect.
> > Your response in step 3 presents $p_i = \frac{\exp (F_i(x)/\tau)}{\sum_j \exp (F_j (x^j)/\tau)}$, differing from Equation 3. You are using different $x$ for the denominator and numerator.
> > >
> >
> > Thank you for your further question. We understand you think in Eq 3, $x$ is the same for the denominator and numerator. We want to clarify that the idea in Eq 3 is the client aggregation probability is proportional to client loss, not $x$.
> >
> > After local update, in FL, the clients' losses vary with different parameters $x_i$, making it unreasonable to use the same $x$ for different clients. This is why we employ $x_i$ in the expression for $p_i=\frac{\exp({F_i(x_{t,K}^i)/\tau})}{\sum_j\exp({F_j(x_{t,K}^j)/\tau})}$, but we as we proved, it still satisfies  $\sum_i p_i=1$.
> >
> > Thank you for bringing this to our attention. We appreciate your feedback and have added an explanation of why we use $x_{t,K}^i$ in Eq 3 in our revised manuscript.
> >
> > > Q4. Why do the authors use weighted average $\frac{n_i}{n}$ for all FL algorithms, but use unweighted average $\frac{1}{|S_t|}$ for the proposed ideal global gradient? In the current design, "the ideal global gradient" does not take the heterogeneity into account, which is not ideal.
> > >
> >
> > We apologize for the confusion caused by the expression. To clarify, we consistently use $\frac{n_i}{n}$ for all algorithms, including our own. The term $\frac{1}{|S_t|}$ is simply an equivalent representation of $\frac{n_i}{n}$ when the client’s dataset size is the same. We mentioned using $\frac{n_i}{n}$ for all algorithms because, in our implementation, we account for heterogeneity scenarios, where naturally $\frac{1}{|S_t|}$ expands into $\frac{n_i}{n}$.
> >
> > We omitted the explanation that $\frac{1}{|S_t|}$ is equivalent to $\frac{n_i}{n}$ when clients have different dataset sizes. We apologize for any confusion caused, and we have added this explanation in Section 4.2.1 of our revised manuscript.
> >
> > > Q5. [1,3] converges by function or variable, [2] and [A] converges with no constant term.
> > >
> >
> > In [1] and [3], the algorithm converges to a neighborhood, considering either function value or variable value, owing to their treatment in a strongly convex setting.
> >
> > In [2], Theorem 1 provides a convergence rate for the surrogate objective, while Theorem 2 demonstrates the convergence of the global objective, serving as the final result of the algorithm:
> >
> > $\min_{t \in[T]}\left\|\nabla F\left(\boldsymbol{x}^{(t, 0)}\right)\right\|^2 \leq
> > \underbrace{2\left[\chi_{\boldsymbol{p} \|\| \boldsymbol{w}}^2\left(\beta^2-1\right)+1\right] \epsilon_{\text{opt}}}_{\text{vanishing error term }} + $
> >
> > $\underbrace{2 \chi_{\boldsymbol{p} \|\| \boldsymbol{w}}^2 \kappa^2}_{\text {non-vanishing error due to obj. inconsistency }}$
> >
> > Upon examination, in Theorem 2, due to biased aggregation, there is a constant term in the convergence rate, **same as ours.**
> >
> > In [A], where there is no biased aggregation or sampling, it is natural not to have a constant term.
> >
> > In the revised manuscript, we have revised the statement into "converge to a neighborhood of the stationary" to better reflect our consideration of a nonconvex setting. Thank you for your suggestion.

---

### Author Response · Authors · 2023-11-16
**Summary of Revision**

First of all, we sincerely appreciate the reviewers for their valuable time and constructive comments. Especially, thanks to the reviewers for acknowledging our focus is pretty impressive and interesting, and that our theories and experiments are clear and comprehensive.
We have thoroughly considered their suggestions and diligently incorporated them into our revised manuscript, clearly marking the changes with the red lines in the revised manuscript.

For the convenience of the reviewers, we outline the key modifications of the manuscript in the following summary.

- New experimental results:
    - [Reviewer yc23] Added FedSGD as a baseline in Table 1 to highlight the differences and superiority of FedEBA+.
    - [Reviewer yc23] Presented complete hyper-parameter ($\tau$ and $\alpha$) tuning results for FedEBA+, demonstrating its relative robustness and continued advantages over other fairness baselines in Table 12.
    - [Reviewer yc23] Introduced new ablation experiments of $\theta$ on CIFAR-10 in addition to the existing ablation study on FashionMNIST, as shown in Table 5.
    - [Reviewer yc23 and ABP1] Conducted an ablation study of FedEBA+ on four datasets (FashionMNIST, CIFAR-10, CIFAR-100, and Tiny-ImageNet) to demonstrate the benefits of each step of our proposed methods, i.e., proposed aggregation method and alignment update method, presented in Table 11.
    - [Reviewer ABP1] Evaluated the performance of algorithms using the coefficient of variation $C_v=\frac{std}{acc}$, simultaneously assessing fairness (variance) and accuracy, with Table 13 showing the significant improvement of FedEBA+.
    - [Reviewer 4hVe] Demonstrated the advantage of FedEBA+ in other fairness metrics, i.e., cosine similarity (angle) and entropy (KL-divergence) in Table 14.
- New analysis result:
    - [Reviewer 4hVe] Provided new fairness analysis using smooth and strongly convex loss functions beyond the regression model, as detailed in Section 12.3.
- Clarifications:
    - [Reviewer 4hVe] Summarized commonly used definitions of fairness metrics, discussed their advantages and disadvantages, and justified the use of variance in Section 16.
    - [Reviewer yc23] Improved and clarified the description of Algorithm 1, incorporating comments on key steps.
    - [Reviewer yc23] Enhanced the writing based on yc23’s suggestions, [illustrating the benefits of entropy-based aggregation with a toy example](https://openreview.net/forum?id=UJeIujVxMn&noteId=o0lyMoZTXY) in Appendix 12.1.
    - [Reviewer yc23] Clarified misunderstandings about the relationship between FedSGD and FedEBA+ and the relationship between entropy and uniform distribution.
    - [Reviewer yc23] Elaborated on the reasonability of the non-vanishing term of convergence, justified the non-iid-ness in our fairness result, and explained the notation of $T()$ and $A()$, as presented in Section 5.
    - [Reviewer yc23, ABP1, and 4hVe] Addressed other concerns in conjunction with experiments.


All experiments and clarifications are referenced in the main paper and provided in the individual responses.

[1]Li T, Sanjabi M, Beirami A, et al. Fair resource allocation in federated learning[J]. ICLR 2020.

---

### Meta-Review · Area_Chair_tMTc · 2023-12-08

**Metareview:**

The paper has received in-depth discussion from the reviewers. After taking a read of the paper myself, I agree with Reviewer yc23 that the paper can be significantly improved for better presentation, especially for the notation used throughout the paper -- the authors should be more consistent with the notation and make it easier for the readers to follow. Technically, although the idea of assigning larger weights to underperforming clients is intuitive, it is not clear how this could lead to the desired fairness measure, i.e., smaller variance across the agents in the network. There must be some assumptions on the data distribution and the model class for this to work. For example, consider the extremal case where there are only two clients, one with separable data and the other with pure noise. In this case, the proposed algorithm would give the client with pure noise a larger weight, and yet this does not help to reduce the variance across the clients since the second client's accuracy cannot be improved due to the noisy underlying distribution. As also pointed out by Reviewer yc23, this may also explain the non-vanishing constant term in the convergence analysis as well.

I would encourage the authors to take a look at the following paper, which also addresses the issue of fairness but with a definition that is more tailored for FL.

[1].    FOCUS: Fairness via Agent-Awareness for Federated Learning on Heterogeneous Data, https://arxiv.org/abs/2207.10265

In the rebuttal, the authors have clarified some of the questions raised by the reviewers. However, significant concerns still remain, including both the high communication complexity of the algorithm, and the issue in the convergence analysis (i.e., the non-vanishing constant term). The authors are encouraged to address these concerns and further improve the presentation of the paper.

**Justification For Why Not Higher Score:**

Despite the active discussion between the authors and reviewers during the rebuttal period, significant technical concerns remain (see the comments from Reviewer yc23 and the above summary for more details). The paper cannot be published due to the technical issues.

**Justification For Why Not Lower Score:**

N/A

---

### Decision · Program_Chairs · 2024-01-16

Reject